# WHEN BIAS MEETS TRAINABILITY:
# CONNECTING THEORIES OF INITIALIZATION

**Alberto Bassi**[†]
ETH Zurich

**Marco Baity-Jesi**
Eawag (ETH)

**Aurelien Lucchi**
University of Basel

**Carlo Albert**
Eawag (ETH)

**Emanuele Francazi**[†]
EPFL

## ABSTRACT

The statistical properties of deep neural networks (DNNs) at initialization play an important role to comprehend their trainability and the intrinsic architectural biases they possess before data exposure. Well established mean-field (MF) theories have uncovered that the distribution of parameters of randomly initialized networks strongly influences the behavior of the gradients, dictating whether they explode or vanish. Recent work has showed that untrained DNNs also manifest an initial-guessing bias (IGB), in which large regions of the input space are assigned to a single class. In this work, we provide a theoretical proof that links IGB to previous MF theories for a vast class of DNNs, showing that efficient learning is tightly connected to a network's prejudice towards a specific class. This connection leads to a counterintuitive conclusion: the initialization that optimizes trainability is systematically biased rather than neutral.

## 1 INTRODUCTION

In recent years, deep neural networks have achieved remarkable empirical success across diverse domains (Jumper et al., 2021; Brown et al., 2020; Ramesh et al., 2021). However, understanding their properties theoretically, especially regarding their trainability, remains challenging. A central difficulty consists in explaining how the choice of hyper-parameters — such as weights and biases variances — governs the network's ability to propagate signals and gradients through depth. Improper initialization typically leads to gradient-related issues: vanishing gradients, causing persistent initial conditions and learning stagnation; or exploding gradients, causing instability in the early stages of training.

Mean-field (MF) theories of wide networks have provided a systematic framework to analyze how these initial parameters shape trainability (Schoenholz et al., 2016; Gilboa et al., 2019; Chen et al., 2018; Ángel Poc-López & Aguilera, 2024; Yang et al., 2019; Yang & Schoenholz, 2017; Xiao et al., 2018; Lee et al., 2018; Hayou et al., 2019; Jacot et al., 2022; Poole et al., 2016; Yang & Hu, 2021). Depending on the initialization state, a network exhibits either an ordered phase, where gradients vanish, or a chaotic phase, where gradients explode. The optimal boundary — the so-called *edge of chaos* (EOC) — is characterized by an infinite depth scale in both the forward and backward pass, making the network effectively trainable. This highlights the initial state's crucial role in determining a network's subsequent learning dynamics.

Beyond their seemingly simplistic theoretical assumption – that is of infinite width – MF theories have led to surprising results valid for real-world models. For the first time, they allowed training of very deep CNN from scratch (Xiao et al., 2018) and hyper-parameters transferability across depth and width through the Tensor Program approach (Yang, 2019; 2020; Yang & Hu, 2021). In particular, Yang & Hu (2021) showed that MF networks are capable of feature learning, effectively exiting the lazy learning regime predicted by the Neural Tangent Kernel (NTK) parametrization. These results underscore the importance of MF theories in effectively describing production architectures, which are in many cases sufficiently large to lay in the MF description.

---

[†]Corresponding authors. Contact: `abassi@ethz.ch` and `emanuele.francazi@gmail.com`

While previous MF theories specialized on the trainability conditions, recent insights – also derived in the infinite-width limit – showed that architectural choices also impact the initial predictive states (Francazi et al., 2024), significantly changing the classification behaviour of neural networks even before being exposed to data. Specifically, an untrained network may exhibit a *prejudice* toward certain classes — referred to as initial guessing bias (IGB) — or it may remain *neutral*, assigning equal frequency to all classes. Surprisingly, IGB solely depends on factors such as network architecture and the initialization of weights. The impact these initial predictive states have on learnability, however, remains unclear. Since IGB is itself a theory of wide-untrained neural networks, it begs the question: how does it connect to previous MF theories of initialization?

**Main Contributions**    In this work, we bridge this gap between MF-based trainability insights and IGB-based predictive state characterizations. Specifically, our contributions are:

- We elucidate the **link between predictive initial behaviours (IGB states) and trainability conditions (MF phases)**, clarifying how initialization hyperparameters and architectural choices jointly determine initial predictive behaviours and shape subsequent training dynamics.
- We show that **trainability in wide architectures coincides with a state of transient deep prejudice at initialization**, challenging the intuitive assumption (Francazi et al., 2024) that the optimal trainability state must be unbiased.
- We verify empirically our conclusions on a wealth of architectures trained on binary and multi-classification tasks, showing that at the EOC models exhibit strong bias, which is absorbed during the initial phase of the learning dynamics.
- We generalize the IGB framework to accommodate non-zero bias terms and multi-node activation functions (such as maxpool layers), further expanding its applicability, and correct existing MF phase diagram inaccuracies (*e.g.* for ReLU).
- We emphasize the role of per-class gradients, showing that the vanishing/exploding gradient behaviour is class-dependent.

Our results are valid for a general wide architecture and only assume its output to be normally distributed, a simpler and more general setting than the original, full IGB analytical framework.

**Practical Consequences of Our Theory**    Aside from the theoretical value of linking bias with trainability, there are some practical takeaways which become obvious in light of our theory:

- Gradient exploding. Due to initialization biases, gradient exploding only involves a subset of classes (Fig. 4). This causes an imbalance in per-class gradients, which can drastically slow down learning (Francazi et al., 2023).
- Hyperparameter tuning. Assessing model performances based on short runs is at risk of privileging specific classes due to residual bias (Fig. 5).
- Knowledge transfer. By linking IGB to previous MF theories, which do not impose any prior data distribution, we relax the assumptions of IGB theory, thus extending its validity. On the converse, the IGB formalism complements the MF phase diagrams, revealing novel phases and unveiling unexpected biases (Fig. 3).

## 2    BACKGROUND

### 2.1    SETUP

We consider a generic neural network architecture $\mathcal{A}$ composed by $L$ layers (depth) of width $N_l$, for $l = 1, \ldots, L$. We define $y_i^{(l)}(a)$ the pre-activation (signal) of each layer $l$ and each neuron $i = 1, \ldots, N_l$, and $\mathcal{W}^{(l)}$ the set of all network's weights at layer $l$. The architecture $\mathcal{A}$ processes sequentially the pre-activations according to the recursive rule:

$$\mathbf{y}^{(l)} = \mathcal{F}\big(\phi^{(l)}\big(\mathbf{y}^{(l-1)}\big); \mathcal{W}^{(l)}\big) \,, \tag{1}$$

where $\phi^{(l)}(\cdot) : \mathbb{R}^{N_l} \to \mathbb{R}^{N_l}$ is the $l$-th layer generic non-linear activation function and $\mathcal{F}$ is a linear function. We consider the problem of processing $N_D$ samples belonging to a dataset $\mathcal{D} = \{\boldsymbol{\xi}(a)\}_{a=1,\ldots,N_D}$ through $\mathcal{A}$, and we denote with $y_i^{(l)}(a)$ the pre-activation computed with the $a$-th data sample as input. Thus, in the input layer we have by definition that $y_i^{(0)}(a) = \xi_i(a)$ and $N_0$ denotes the input dimension. We consider the large-width limit taken before that of the dataset samples and depth. Formally, our results are valid when the limiting order is: for any function $f$, we take $\lim_{L \to \infty} \lim_{N_D \to \infty} \lim_{\min_l(N_l) \to \infty} f$.

For the sake of simplicity, we will specialize the discussion to multi-layer perceptrons (MLPs), which still play an important role in modern machine learning as they are the building blocks of most complex architectures. For instance, Transformers (Vaswani et al., 2017) are composed by several stacked MLPs and attention layers. For MLPs, Eq. (1) reads $\mathcal{W}^{(l)} = \{\boldsymbol{W}^{(l)}, \mathbf{b}^{(l)}\}$ and $\mathcal{F}$ is the affine transformation: $\mathcal{F}(\mathbf{x}; \boldsymbol{W}^{(l)}, \mathbf{b}^{(l)}) = \boldsymbol{W}^{(l)}\mathbf{x} + \mathbf{b}^{(l)}$, where $\boldsymbol{W}^{((l))}$ is the weight matrix and $\mathbf{b}^{(l)}$ the bias vector.

## 2.2 ORDER/CHAOS PHASE TRANSITION: INITIALIZATION CONDITIONS FOR TRAINABILITY

**One kind of average**  When the datapoints are fixed, the pre-activations of an untrained MLP are random functions of one source of randomness, coming from the joint set of all weights and biases, shortly denoted with $\mathcal{W}$. One usually considers the random ensemble corresponding to initializing weights and biases i.i.d. with Gaussian distributions (Schoenholz et al., 2016; Poole et al., 2016; Hayou et al., 2019), i.e. $W_{i,j}^{(l)} \sim \mathcal{N}\left(0, \sigma_{w^{(l)}}^2/N_l\right)$ and $b_i^{(l)} \sim \mathcal{N}\left(0, \sigma_{b^{(l)}}^2\right)$ for every neurons $i, j = 1, \ldots, N_l$ and for every layer $l = 1, \ldots, L$. The scaling of the weight variance by the square root of the network's width is necessary in order maintain the signal $O(1)$ as it transverses the network.

In this setup, only one kind of average naturally arises: the average over $\mathcal{W}$ at fixed dataset $\mathcal{D}$, which is denoted with an overbar, $\bar{x} \equiv \mathbb{E}_{\mathcal{W}}(x|\mathcal{D})$. For fixed inputs, when performing the limit of infinite width before that of depth, the pre-activations become i.i.d. Gaussian variables with mean $\mu^{(l)} = 0$ and signal variance $\sigma_{y^{(l)}}^2 = q_{aa}^{(l)}$, with $q_{aa}^{(l)}\delta_{ij} = \overline{y_i^{(l)}(a)y_j^{(l)}(a)}$ (Lee et al., 2018). The infinite-width limit is commonly referred to as mean-field regime, since correlations among neurons vanish and the pre-activation distributions are fully characterized by the signal variance. The pre-activation distributions defined in MF theory permit the study of the propagation of the signal through the network; however, they do not yield any insight into interactions among distinct data samples. To such end, one has to define a correlation coefficient between inputs as

$$c_{ab}^{(l)} = q_{ab}^{(l)}/\sqrt{q_{aa}^{(l)}q_{bb}^{(l)}}, \tag{2}$$

where $\overline{y_i^{(l)}(a)y_j^{(l)}(b)} = q_{ab}^{(l)}\delta_{ij}$, and $q_{ab}^{(l)}$ is the signal covariance between inputs $a$ and $b$. The reader is referred to App. C, where we report the recursive relations for the signal variance [Eq. (7)] and covariance [Eq. (8)], first derived by Poole et al. (2016).

**Phase transition in bounded activation functions**  The authors of Schoenholz et al. (2016) extensively analyzed bounded activation functions, such as Tanh. By defining $\chi_1 \equiv \partial c_{ab}^{(l+1)}/\partial c_{ab}^{(l)}|_{c=1}$, $\chi_1 = 1$ separates an ordered phase ($\chi_1 < 1$) where the correlation coefficient converges to one, *i.e.* $c \equiv \lim_{l \to \infty} c_{ab}^{(l)} = 1$, and a chaotic phase ($\chi_1 > 1$) where the correlation coefficient converges to a lower value. Additionally, the value of $\chi_1$ determines the transition from vanishing gradients ($\chi_1 < 1$), to exploding gradients ($\chi_1 > 1$). These two phases have well-known consequences for training: vanishing gradients hinder learning by causing a long persistence of the initial conditions, while exploding gradients lead to instability in the training dynamics (Bengio et al., 1994; Pascanu et al., 2013). At the transition point, both the gradients are stable and the depth-scale of signal propagation diverges exponentially. This is the optimal setting for training, as it allows all layers in the network to be trained from the start.

**Phase diagram**  The MF theory constructs a phase diagram by looking at the convergence behaviour of the correlation coefficient, in terms of the weight and bias variances at initialization — that is, the $(\sigma_b^2, \sigma_w^2)$ plane. $\sigma_b^2$ and $\sigma_w^2$ are referred to as *control parameters*, whereas the quantities identifying different phases are termed *order parameters*. Consequently, for bounded activation

functions, the correlation coefficient constitutes an order parameter, as its asymptotic value alone suffices to distinguish between phases.

**Unbounded activation functions** In this case, the signal variance is not guaranteed to converge, and one has to account for unbounded signals when defining the order/chaos phase transition. For example, it is possible for the correlation coefficient to converge always to one in the whole phase diagram, as we will later demonstrate for ReLU; hence, $c$ does not always serve as an effective order parameter for discriminating between phases. In particular, in App. C we prove that the quantity $\tilde{\chi}_1 \equiv \partial q_{ab}^{(l+1)}/\partial q_{ab}^{(l)}|_{c=1}$ can discriminate the ordered from the chaotic phase and it is equal to $\chi_1$ in the domain of convergence of the variance. Thus, for unbounded activation functions, $\tilde{\chi}_1 = 1$ separates the region in the phase diagram with exploding gradients from the one where gradients vanish, acting as a discriminative order parameter. Following Hayou et al. (2019), it is therefore appropriate to define the *edge of chaos* (EOC) as the set of points in the phase diagram where 1) the signal variance converges and 2) $\chi_1 = \tilde{\chi}_1 = 1$. Additionally, Hayou et al. (2019) provided a simple algorithm (Algorithm 1 of the main paper) to compute the EOC for a generic single-node activation function. This way, it is also possible to analyze unbounded activation functions in regions of the phase diagram characterized by convergent signals.

### 2.3 INITIAL GUESSING BIAS: PREDICTIVE BEHAVIOUR AT INITIALIZATION

**Two kinds of averages** For randomly initialized deep neural networks processing inputs drawn from a dataset distribution, two distinct sources of randomness naturally arise: randomness from network $\mathcal{W}$ and randomness from $\mathcal{D}$. The MF approaches typically fix the input and average over the ensemble of random weights to analyze signal propagation. In contrast, recent studies (Francazi et al., 2024; 2025) introduced an alternative approach — the IGB framework — where, for a fixed initialization, the entire input distribution is propagated through the network. Coherently with these works, here we suppose each data component to be i.i.d. according to a standard Gaussian distribution, *i.e.* $\xi_j(a) \sim \mathcal{N}(0,1)$, $\forall a \in \mathcal{D}$. Interestingly, when averages over the dataset are performed first, the pre-activation distributions change, being not centred around zero (Francazi et al., 2024): $y_i^{(l)}(a) \sim \mathcal{N}\left(\mu_i^{(l)}, \sigma_{y^{(l)}}^2\right)$, with $\mu_i^{(l)} \sim \mathcal{N}\left(0, \sigma_{\mu^{(l)}}^2\right)$.

**Neutrality vs prejudice** Within the IGB framework, a predictive bias arises as a consequence of a systematic drift in signal activations. In the presence of IGB, the pre-activation signals at each node are still Gaussian distributed in the infinite-width limit, with variance $\sigma_{y^{(l)}}^2$, but each node $i$ is centred around a different point, $\mu_i^{(l)}$, which is generally different from zero. The centers only vary with initialization, and are Gaussian-distributed too, with zero mean and variance $\sigma_{\mu^{(l)}}^2$. Now the signals related to different nodes in the same layer are distributed differently. This causes a misalignment between the decision boundary — initially positioned near the origin — and the data distribution (Francazi et al., 2024). Consequently, most input points are assigned to a single class, defining a predictive state which we term as *prejudice*, to emphasize that this is a statistically skewed assignment of classes *before any data is observed or evaluated.*

Conversely, when the drift is negligible and activations remain symmetrically distributed around zero, predictions remain balanced across classes; we call it *neutral* state. Prejudice can manifest at different levels, depending on the strength of the classification bias. The extent of activation drift and corresponding predictive bias can be quantified by the activation drift ratio $\gamma^{(l)}$.

**Definition 2.1** (Activation Drift Ratio). We define the activation drift ratio at layer $l$ as:

$$\gamma^{(l)} \equiv \sigma_{\mu^{(l)}}^2 / \sigma_{y^{(l)}}^2 \ , \tag{3}$$

where $\sigma_{\mu^{(l)}}^2$ is the variance across random initializations of node activation centers (averaged over the dataset), and $\sigma_{y^{(l)}}^2$ is the variance of activations due to input data variability (at fixed initialization).

If the variances around the node centers are much larger than the variances of the centers themselves, the signals of different nodes become indistinguishable. Thus, large (diverging) values of

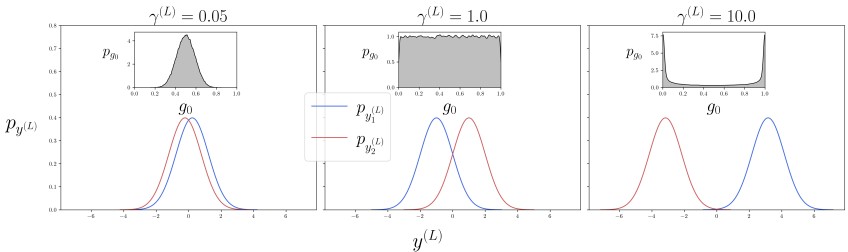

Figure 1: Example of pre-activation distributions for neutrality (**left**) and moderate prejudice (**right**) computed by sampling Gaussian variables with synthetic data. The inset plots show the distribution of the dataset elements classified into the reference class $G_0$. In the neutral phase, $G_0$ is centred around $0.5$, while with moderate prejudice, $G_0$ concentrates at the extremes. At the transition between these two phases, $G_0$ is uniformly distributed (**middle**).

$\gamma^{(l)}$ indicate significant drift (strong prejudice), whereas small (vanishing) values reflect minimal drift (neutrality).

**Classification fractions** In classification tasks, predictive bias can be quantified by measuring the fraction of inputs, $G_c$, classified into each class $c$ at initialization. For illustrative clarity, we consider the binary setting, where the predictive imbalance is fully captured by the fraction of inputs assigned to one reference class. Here, we derive an implicit formula to compute the distribution over $\mathcal{W}$ of the fraction of points classified as a reference class in the infinite-width regime, where all pre-activations are normally distributed: $G_0 \equiv \mathbb{P}\left(y_1^{(L)} > y_2^{(L)} \mid \delta(\mathcal{W})\right) = \Phi\left(\sqrt{\frac{\gamma^{(L)}}{2}}\delta\right)$, where $\Phi\left(\cdot\right)$ is the Gaussian cumulative function, and $\delta$ is a standard Gaussian variable. The proof is straightforward upon utilizing the IGB pre-activation distributions (see App. D).

The value of $\gamma^{(L)}$ permits to distinguish between three phases and connects them to the prejudice/neutrality framework provided before. When $\gamma^{(L)} \ll 1$, the distribution of $G_0$ converges, in the distribution sense, to a Dirac-delta centred in $0.5$, whereas for $\gamma^{(L)} \gg 1$ its distribution converges to a mixture of two Dirac-deltas centred in $0$ and $1$, respectively. Remarkably, for $\gamma^{(L)} = 1$ (Fig. 1 - middle), $G_0$ is uniformly distributed in $(0,1)$; this critical threshold delineates a phase where the fraction of points classified to a reference class exhibits a Gaussian-like shape centred in $0.5$ (Fig. 1 - left), from one where the distribution deviates markedly from Gaussian behaviour and it is bi-modal (Fig. 1 - right). Consequently, neutrality emerges for $\gamma^{(L)} < 1$, whereas for $\gamma^{(L)} > 1$ the network exhibits prejudice. Moreover, prejudice can compound with depth — we call this *deep prejudice* — when $\gamma \equiv \lim_{L \to \infty} \gamma^{(L)} = \infty$, resulting in a network with strongly-biased predictions.

The IGB framework offers a rigorous theoretical characterization specifically in settings involving random, unstructured data identically distributed across classes. This demonstrates how predictive imbalances at initialization can emerge purely from architectural choices, independently of any intrinsic data structure. Moreover, it provides quantitative tools to distinguish unbiased (*neutrality*) from biased (*prejudice*) initial states.

## 3 CONNECTING CLASSIFICATION BIAS TO THE ORDERED PHASE

In this section, we establish a direct link between the IGB framework and the standard MF theory. The phase diagram in MF is typically expressed in terms of the initialization parameters $(\sigma_b^2, \sigma_w^2)$, whereas the original formulation of IGB was limited to the case $\sigma_b^2 = 0$. To bridge this gap and lay the groundwork for a unified understanding, we extend the IGB framework to include non-zero bias variances (App. D). This extension allows us to reinterpret the MF phase diagram in terms of the IGB phases, revealing the connection between initial predictive behaviour and trainability. Here, we show that all the quantities of interest in MF have an equivalent counterpart in the IGB framework.

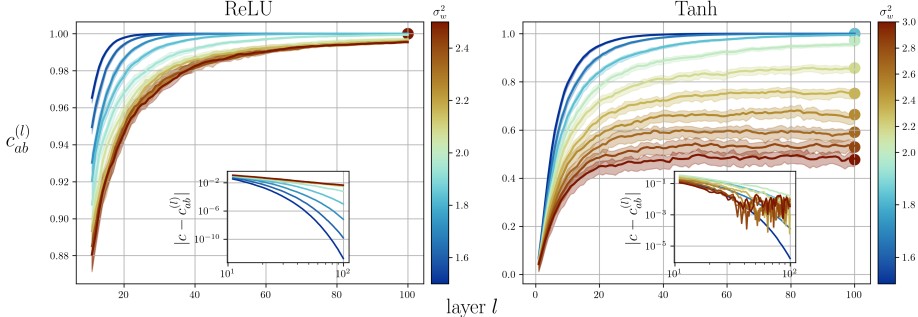

Figure 2: Convergence behaviour of the correlation coefficient of ReLU and Tanh for a single MLP with width equal to 10 000 and depth 100. $\sigma_b^2 = 0.1$ and $\sigma_w^2$ varies uniformly from the ordered phase (blue) to the chaotic phase (red). The transition point is $\sigma_w^2 = 2.0$ for ReLU and close to it for Tanh. Scatter points indicate the asymptotic values. The inset plots show the convergence rate for the correlation coefficient to ity asymptotic value $c$, always exponential for Tanh and power law for ReLU in the chaotic phase. Solid lines are computed using the IGB approach [Eq. (6)], while shaded areas represent the 90 % central confidence interval computed using the MF approach [Eq. (2)].

---

**Theorem 3.1** (Equivalence between MF and IGB, informal). *Consider a generic architecture $\mathcal{A}$ (Eq. (1)) in the mean-field regime. Let us suppose that $q_{aa}^{(0)} = 1, \forall a \in \mathcal{D}$ and $q_{ab}^{(0)} = 0, \forall a, b \in \mathcal{D}, a \neq b$. Then in the infinite data limit, $\forall a \in \mathcal{D}$ and $\forall l > 0$, the total variance in the IGB approach is equal to the signal variance in the MF approach:*

$$q_{aa}^{(l)} = \sigma_{\mu^{(l)}}^2 + \sigma_{y^{(l)}}^2 . \tag{4}$$

*Moreover, $\forall a, b \in \mathcal{D}$ with $a \neq b$, the centers variance in the IGB approach is equal to the input covariance in the MF approach:*

$$q_{ab}^{(l)} = \sigma_{\mu^{(l)}}^2 , \tag{5}$$

*and the correlation coefficient is related to $\gamma$ through*

$$c_{ab}^{(l)} = \frac{\gamma^{(l)}}{1 + \gamma^{(l)}} . \tag{6}$$

---

We report the proof of this theorem in App. E. The key result of Thm. 3.1 establishes a *correspondence* between the MF formulation and IGB, enabling signal propagation in wide networks to be described interchangeably using either framework. Intriguingly, this result makes no further assumption than the MF regime, and it remains valid for *every* architecture defined by Eq. (1). In MF theory, $q_{aa}^{(l)}$ and $q_{ab}^{(l)}$ are generally functions of the dataset $\mathcal{D}$, and therefore become random variables upon imposing a distribution over $\mathcal{D}$, as done in the IGB approach. Remarkably, these two quantities concentrate around their mean in the infinite-width and then -data limit, allowing their treatment as deterministic variables (equivalently through $\sigma_{y^{(l)}}^2$ and $\sigma_{\mu^{(l)}}^2$). On the one hand, this formal connection between MF and IGB frameworks enables a unified view, where predictive behaviour at initialization and trainability conditions are jointly entangled, enriching the classical MF picture, as will be discussed in Secs. 4 and 5. On the other, it extends the IGB framework from Francazi et al. (2024) to settings not previously analyzed (e.g., identifying prejudice and neutrality phases for Tanh activations directly from the MF phase diagram (Schoenholz et al., 2016)).

In Fig. 2, we test the validity of Thm. 3.1 by plotting the correlation coefficient in function of the depth for ReLU and Tanh MLPs with $\sigma_b^2 = 0.1$ and $\sigma_w^2$ uniformly varying from the ordered to the chaotic phase analyzed in MF theory. We show similar results on realistic architectures and datasets in App. E.1. In all cases we observe a good agreement between the curves obtained with the IGB approach (solid lines - computed via Eq. (6)) and the 90 % central confidence interval computed using the MF approach (shaded areas - computed via Eq. (2)). The distribution of MF

is very narrow around the IGB-computed values, corroborating the treatment of the signal variance and covariance as deterministic variables. As network's width increases, the MF distributions of $q_{aa}^{(l)}$ and $q_{ab}^{(l)}$ become progressively more concentrated (see App. E). For ReLU, we observe that the correlation coefficient always converges to one, but the convergence rate is exponential in the ordered phase ($\sigma_w^2 < 2$) and follows a power-law in the chaotic phase ($\sigma_w^2 > 2$). For Tanh, the correlation coefficient converges to one in the ordered phase and to a lower value in the chaotic phase; in this case, we always observe an exponential convergence behaviour.

## 4 BEST TRAINABILITY CONDITIONS

In Sec. 3, we proved the connection between IGB and MF frameworks. We will now see how this association allows us to connect predictive behaviour at initialization with dynamic behaviour, specifically in terms of the network's trainability conditions, rooted in gradient stability.

Gradients at initialization have extensively been analyzed in the MF literature, so the reader is referred to, for example, Schoenholz et al. (2016) or Hayou et al. (2019) for an extensive discussion. For our purposes, it is sufficient to note that $\tilde{\chi}_1 \equiv \partial q_{ab}^{(l+1)}/\partial q_{ab}^{(l)}|_{c=1}$ is a key quantity separating the ordered phase, where gradients vanish, from the chaotic phase, where gradients explode. We show examples of this transition from gradient vanishing to exploding in Fig. 4–left and in App. G). When the signal variance is non-divergent, $\tilde{\chi}_1 = \chi_1$ measures the stability of the fixed point $c = 1$, where its dynamic counterpart $c_{ab}^{(l)}$ is connected to $\gamma^{(l)}$ through Eq. (6). In both the ordered phase and at the EOC, the state $c = 1$ is a stable fixed point. Consequently, in both cases, we observe an asymptotic $\gamma = \infty$, indicating a state of deep prejudice at initialization.

However, the dynamical behaviour differs significantly between ordered and chaotic phases. In the ordered phase, gradients vanish exponentially, resulting in a state of *persistency* of the initial conditions characterized by $c = 1$ and $\tilde{\chi}_1 < 1$ (see Tab. 1). At the EOC ($\tilde{\chi}_1 = 1$), by contrast, gradients remain stable, enabling trainability and facilitating the gradual absorption of the initial bias, resulting in a condition of *transiency* of deep prejudice. Conversely, the chaotic phase — in which training is precluded by gradient instability ($\tilde{\chi}_1 > 1$) — is generally characterized either by prejudice (*i.e.*, $1 > c > 0.5$), or neutrality (*i.e.*, $c < 0.5$).

In Francazi et al. (2024), one of the main open questions concerned the distinction in dynamical behaviour between neutral and prejudiced phases. The present results not only clarify this distinction by linking it to gradient stability properties, but also reveal a finer structure within the prejudiced phase, identifying conditions that govern the persistence of predictive bias. Moreover, these findings lead to the following conclusion (see proof in App. E), which counters the suggestion of Francazi et al. (2024), that neutral initializations lead to the fastest dynamics.

> **Proposition 4.1.** *From a trainability perspective, the optimal initial condition (EOC) is not one of neutrality, but rather a state of transient deep prejudice.*

We validate this proposition by performing training experiments of different architectures across multiple datasets (see Sec. 6).

**How to access the EOC in practice** In real scenarios, when the EOC is not known analytically, one can compute the initialization gradients for different values of $\sigma_w^2$ and $\sigma_b^2$ and choose the combination that shows stable gradients across the depth. In App. G, we report initialization gradients for an array of architectures and datasets, showing how the EOC is reached. We also compare the propagation of signals in pretrained vs untrained networks, showing that also in pretrained models one has a similar phenomenology (App. G.4.1).

## 5 DETAILED PHASE DIAGRAMS

Due to the equivalence between IGB and MF, all MF results remain valid in the IGB framework. Therefore, for a comprehensive analysis of generic single-node activation functions, we refer to the work of Hayou et al. (2019). Beyond that, in App. F we extend the MF/IGB theory to multi-node activation functions, broadening the range of applicability of these theories to e.g. max- and average-pool layers.

Here, as an example, we compare the differences between two widely utilized activation functions:

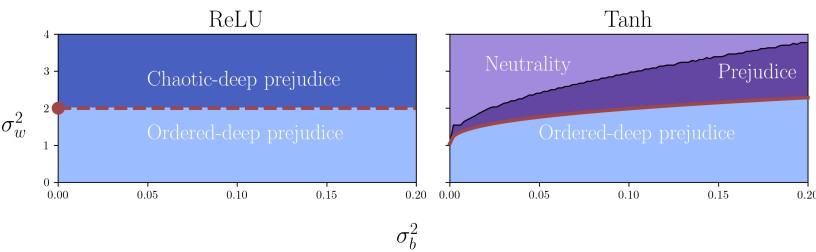

Figure 3: Extensive phase diagrams of infinitely wide MLPs, where we can observe some phases described in Tab. 1. The EOC is indicated with a continuous red line and it becomes a single point for ReLU (unbounded). In general, red lines indicate the transition between vanishing/exploding gradients.

Table 1: Phase descriptions with IGB and MF order parameters.

| IGB | MF | | Phase |
|---|---|---|---|
| | | $\tilde{\chi}_1 < 1$ | **Ordered-deep prejudice** |
| $\gamma = \infty$ | $c = 1$ | $\tilde{\chi}_1 = 1$ | **Transient-deep prejudice (EOC)** |
| | | $\tilde{\chi}_1 > 1$ | **Chaotic-deep prejudice** |
| $1 < \gamma < \infty$ | $0.5 < c < 1$ | $\tilde{\chi}_1 > 1$ | **(chaotic) Prejudice** |
| $\gamma < 1$ | $c < 0.5$ | $\tilde{\chi}_1 > 1$ | **(chaotic) Neutrality** |

Tanh (bounded) and ReLU (unbounded). This analysis enables the construction of a comprehensive phase diagram for these two illustrative cases, thereby broadening the range of phases examined in the preceding sections. A summary of these phases is reported in Tab. 1.

**Bounded activation functions** In this case, the signal variance is also bounded and the value of $\chi_1$ fully delineates the ordered and the chaotic phases. As analyzed in Schoenholz et al. (2016), the chaotic phase of Tanh — characterized by gradient explosion and training instability — induces a shift of the correlation fixed point to $c < 1$, which is shown in Fig. 2 (right plot). Remarkably, the EOC exists for every $\sigma_b^2 \in \mathbb{R}^+$ with non-trivial shape (Hayou et al., 2019) (right plot of Fig. 3).

**Unbounded activation functions** This case has been extensively analyzed by Hayou et al. (2019). However, prior work has overlooked that, for ReLU networks, the correlation coefficient $c^{(l)}$ converges to $c = 1$ across the entire phase diagram (see Fig. 2 - left plot), revealing a persistent deep prejudice at initialization. In App. H, we derive explicit recursive relations for the IGB metrics in ReLU networks, demonstrating that $\lim_{l \to \infty} c^{(l)} = 1$, while $\gamma^{(l)}$ diverges. Nevertheless, the two MF phases remain distinct, as in the bounded activation case: in the ordered phase, gradients vanish; in the chaotic phase, gradients explode. Crucially, these two phases differ in their depth-scaling behaviour: in the ordered phase, the total signal variance converges and $\gamma^{(l)}$ diverges exponentially with depth, whereas in the chaotic phase, the signal variance diverges and $\gamma^{(l)}$ follows a power-law divergence (Lemma H.1). Hence, persistent deep prejudice may arise via two distinct mechanisms. In the first, the total signal variance ($\sigma_{y^{(l)}}^2 + \sigma_{\mu^{(l)}}^2$) remains bounded while $\sigma_{y^{(l)}}^2$ tends toward zero; we denote this phase *ordered-deep prejudice*, owing to its link with vanishing gradients. In the second mechanism, at least one of $\sigma_{y^{(l)}}^2$ or $\sigma_{\mu^{(l)}}^2$ diverges, causing network outputs to blow up and gradients to explode. We refer to this as *chaotic–deep prejudice*. Two independent order parameters are required to distinguish between these regimes.

**Per-class gradient exploding.** The behaviour of the signal in the chaotic phase of unbounded activation functions has direct consequences for training with cross entropy loss. Indeed, large separation of output distributions makes the softmax to concentrate entirely to a single class (which depends on the weights initialization), therefore causing a strong *per-class* dependence of gradients vanishing/exploding (see Fig. 4 and App. G).

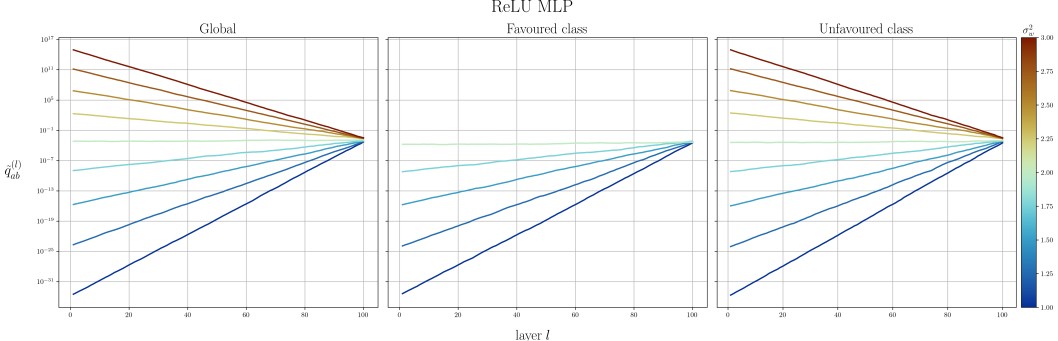

Figure 4: Layerwise gradients $\tilde{q}_{ab}^{(l)}$ as a function of the network layer, of a ReLU MLP in the MF regime computed on a batch of CIFAR10. **Left:** we show the total gradients. **Center:** we show the most favored class (the one towards which there is most prejudice). **Right:** the least favored class. In the chaotic phase, the gradients of the favored class are numerically zero. In App. G we explain the notation $\tilde{q}_{ab}^{(l)}$ for the gradients, and show similar figures on different datasets and architectures.

Therefore, depending on the MLP architecture design and the initialization hyper-parameters (such as weight and bias variances), different behaviours can emerge at initialization. Specifically, a network can become untrainable either by entering a persistent-deep prejudice phase (either characterized by vanishing or exploding gradients) or a purely chaotic phase, in which the signal variance is finite and gradients explode. The nature of the chaotic phase itself depends critically on the activation function. For bounded activations, the chaotic phase can give rise to either a prejudiced phase ($0.5 < c < 1$) or a neutral phase ($c < 0.5$), depending on the initialization parameters. In contrast, for ReLU, the chaotic phase leads exclusively to chaotic-deep prejudice, where biased initial predictions are coupled with dynamical instability due to gradient explosion.

Therefore, successful training requires finely tuning the initialization to precisely sit at the transition between these phases — the transient-deeep prejudice phase (equivalent to EOC) — where gradients are stable and both persistence and instability are avoided.

## 6 TRAINING DYNAMICS

In order to experimentally verify Prop. 4.1, we train several vanilla architectures (MLP, residual MLP, Vision Transformer) on binary (binarized Fashion MNIST, binarized CIFAR10) and multi-class classification tasks (CIFAR10) across the phases described in Tab. 1. Extensive details on the training and the models employed are reported in App. B. In order to evaluate the evolution of bias and overall performance during training, we report the global accuracies, and the maximum classification frequency at each training step. For a Tanh MLP, we can observe the dynamical behaviour described in this work (Fig. 5 - top). Indeed, the EOC corresponds to the maximally biased state, but this bias is rapidly absorbed at the beginning of the dynamics. As expected, the unbiased state (neutrality) performs poorly and it cannot retrieve high accuracy.

As an example of more complex architecture, we study empirically a vanilla Vision Transformer (Dosovitskiy et al., 2020), where we remove all batch- and layer-norm and skip connections, which may destroy the distinction between phases (indeed, it is easy to show that e.g. a residual MLP has only a critical phase of optimal learning (Yang & Schoenholz, 2017)). Our simplified Vision Transformer posseses only convolutional, linear and attention layers, and it shows the same transition behaviour of the gradients as the MLP (Fig. 27).

Finally, we validate our assumption by fine-tuning on CIFAR100 a large Vision Transformer (without modifications) pre-trained on ImageNet (Deng et al., 2009) (Fig. 5 - bottom). By multiplying all linear and convolutional layers' weights by a factor $\sigma_w^2$, we trigger the phase behaviour which we observe in vanilla models. The optimal training state, corresponding to the original non-scaled model, exhibits weak IGB (i.e. maximum classification frequency is significantly greater than $1/n_c$, where $n_c$ is the number of classes). However, slightly reducing all weights by a factor of $\sigma_w^2 = 0.5$ triggers a strong IGB phase, while a larger scaling factor ($\sigma_w^2 = 1.5$) reduces IGB, yet hindering the

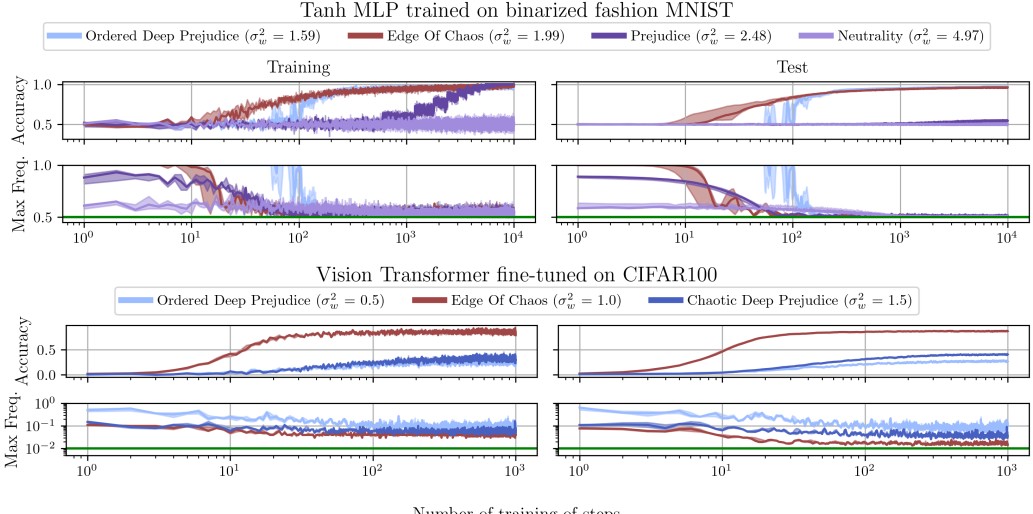

Figure 5: Accuracies and max classification frequency for a Tanh MLPs trained on binarized fashion MNIST, and the large Vision Transformer fine-tuned on CIFAR100. The EOC corresponds to both the initial state with the fastest learning dynamics and maximally biased state. The green line corresponds to $1/n_c$, where $n_c$ is the number of classes.

training dynamics. In both cases, the dynamical instabilities encountered either due to vanishing or exploding gradients cannot be absorbed in the early phase of learning.

## 7    CONCLUSIONS AND OUTLOOK

We established an equivalence between two independent frameworks for the analysis of wide networks at initializaton: mean field theory (MF) and initial guessing bias (IGB). We showed that the fundamental quantities in the two approaches can be mapped onto one another. This connection offered us the possibility to reinterpret the order/chaos phase transition in light of classification bias; the ordered phase is characterized by persistent-deep prejudice, while the chaotic phase is either characterized by persistent-deep prejudice, prejudice or neutrality. Furthermore, our categorization of the edge of chaos (EOC) as a state with deep prejudice reveals that the best trainable model necessarily exhibits bias, which, however, rapidly disappears in the learning dynamics.

Our findings have important implications for understanding the role of architectural choices and hyper-parameter choices in shaping the onset behaviour of deep networks. They suggest that even before training begins, design decisions can inject systematic biases that impact signal propagation, gradient stability, and ultimately trainability.

Overall, our work provides a new lens to understand how dataset randomness at initialization, coupled with architectural design, shapes the phase diagram of large MLPs at initialization. We hope this connection between MF and IGB inspires further exploration of the subtle interplay between structure, randomness, and learning dynamics in modern neural networks, as well as of the conditions for reabsorption of initialization biases.

**Practical Takewaways** Finally, we highlight two practical takeaways of our work. First: neutral initializations are inefficient, so typical initializations are prejudiced. This means that, in order to be meaningful, hyperparameter tuning runs need to be long enough to reabsorb IGB. As our runs show, the EOC is not only optimal from the point of view of stability of the gradients, but also in terms of how fast the biases are reabsorbed. A consequence is that, if one tunes the weights' variances in order to be at the edge of chaos (as explained in Sec. 4), shorter HP tuning runs are needed. Second: since initializations are biased, the gradients of the favored classes will typically be smaller than those of the unfavored. This is especially true with gradient exploding in the chaotic deep prejudice phase, where one class has exploding, while the other(s) have zero gradients (Fig. 4). Such disparities in the per-class gradients can have drastical impacts on the learning speed and quality, as demonstrated in Francazi et al. (2023).

## REPRODUCIBILITY STATEMENT

We provide the full code used to replicate all the simulations and training experiments at the link: `https://github.com/abassi98/igb_and_trainability.git`. We declare the usage of Large Language Models for finding related work.

## ACKNOWLEDGEMENTS

This work was supported by the Swiss National Science Foundation, SNSF grants # 208249 and # 196902.

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

SUPPLEMENTARY MATERIALS

## A    EXTENSIVE NOTATION

- $p_X(x')$: Given a random variable $X$, $p_X(x')$ denotes the probability density function (p.d.f.) evaluated at $X = x'$. Formally, $p_X(x') = \frac{d}{dx}F_X(x)\big|_{x=x'}$. For variables with multiple sources of randomness, if some of these sources are either fixed or marginalized, we will specify the active sources of randomness as subscripts.

- $\mathrm{erf}(\cdot)$: Error function. $\mathrm{erf}(\cdot) \equiv \frac{2}{\sqrt{\pi}} \int_0^x e^{-t^2} dt$

- $F_X(x)$: Given a r.v. $X$, we denote its cumulative distribution function (c.d.f.) as $F_X(x)$, i.e., $F_X(x) = \mathbb{P}(X < x)$. Considering the r.v. $X(\mathcal{D}, \mathcal{W})$, which is a function of two independent sets of random variables $\mathcal{D}$ and $\mathcal{W}$, when one of these sources of randomness is fixed, we explicitly indicate the active source in the notation. For example, $F_X^{(\mathcal{D})}(x) = \mathbb{P}(X < x \mid \mathcal{W})$.

- $\mathcal{N}(x; \mu, \sigma^2)$: Given a Gaussian r.v. $X$, we indicate with $\mathcal{N}(x; \mu, \sigma^2)$ the p.d.f. computed at $X = x$, i.e. $\mathcal{N}(x; \mu, \sigma^2) \equiv p_X(x) = \frac{e^{-\frac{1}{2\sigma^2}(x-\mu)^2}}{\sqrt{2\pi\sigma^2}}$

- $\delta_D$: the Dirac-delta function.

- $\theta_H$: the Heaviside-theta function.

- $\Phi(x) = \frac{1}{2}(1 + \mathrm{erf}(x))$: Gaussian cumulative function.

- $\mathcal{D}y = \frac{dy}{\sqrt{2\pi}} e^{-y^2/2}$ denotes the standard Gaussian measure.

- $\mathcal{A}$: generic neural network archiecture.

- $L$: number of layers.

- $N_l$: number of node of layer $l$.

- $N_0$: network's input dimension.

- $\xi(a) \in \mathbb{R}^{N_0}$: $a$-th data sample.

- $N_D$: number of dataset samples.

- $\mathcal{D}$: dataset, i.e. collection of $N_D$ data samples, i.e. $\{\xi(a)\}_{a=1,\dots,N_D}$.

- $\langle x \rangle$: average of $x$ w.r.t. the input dataset, i.e. $\langle x \rangle \equiv \mathbb{E}_{\mathcal{D}}(x|\mathcal{W})$.

- $\bar{x}$: average of $x$ w.r.t. the weights $\mathcal{W}$, i.e. $\bar{x} = \mathbb{E}_{\mathcal{W}}(x|\mathcal{D})$.

- $\mathrm{Var}_{\#}(\cdot)$: Indicate the variance of the argument. Since we have r.v.s with multiple sources of randomness where necessary we will specify in the subscript the source of randomness used to compute the expectation. For example $\mathrm{Var}_{\mathcal{D}}(\cdot) \equiv \langle \cdot - \langle \cdot \rangle \rangle^2$.

- $\mathrm{Cov}_{\#}(\cdot, \cdot)$: similar to Var () but for the covariance among different variables.

- $\boldsymbol{W}^{(l)}$: weight matrix at layer $l$, i.e. matrix of elements $W_{i,j}^{(l)}$ connecting neuron $j$ at layer $l$ with neuron $i$ at layer $l+1$.

- $\mathbf{b}^{(l)}$: bias vector at layer $l$, i.e. vector of elements $b_i^{(l)}$, corresponding to neuron $i$.

- $\sigma_w^2$: variance of the weights. Each weight $W_{i,j}^{(l)}$ is i.i.d. as a Normal with zero mean and variance $\sigma_w^2$.

- $\sigma_b^2$: variance of the biases. Each bias $b_i^{(l)}$ is i.i.d. as a Normal with zero mean and variance $\sigma_b^2$.

- $\mathcal{W}$: shorthand notation for the set of all network weights and biases, i.e. for the MLP: $\{\boldsymbol{W}^{((l))}, \mathbf{b}^{(l)}\}_{l=1,\dots,L}$.

- $N_l$: number of nodes in the $l$-th layer; $N_0 \equiv d$ indicates the dimension of the input data (number of input layer nodes) while $N_L$ the number of classes (number of output layer nodes).

- $G_c$: fraction of dataset elements classified as belonging to class $c$. The argument $M$ indicates the total number of output nodes for the variable definition, i.e. the number of classes considered. For binary problems we omit this argument ($G_0$) as there is only one non-trivial possibility, i.e. $M = 2$.

- $n_c$: number of classes.
- $y_i^{(l)}(a)$: pre-activation of neuron $i$ at layer $l$, computed by propagating the $a$-th data sample.
- $q_{ab}^{(l)} = \delta_{ij}\overline{y_i^{(l)}(a)y_j^{(l)}(b)}$: the covariance of the pre-activation signals at the $l$-th hidden layer nodes, corresponding to inputs $a$ and $b$, computed before applying the activation function.
- $q_{aa}^{(l)} \equiv \delta_{ij}\overline{y_i^{(l)}(a)y_j^{(l)}(a)}$: the variance of the pre-activation signals at the $l$-th hidden layer nodes, corresponding to inputs $a$, computed before applying the activation function.
- $c_{ab}^{(l)} \equiv \dfrac{q_{ab}^{(l)}}{\sqrt{q_{aa}^{(l)} q_{bb}^{(l)}}}$: correlation coefficient between the pre-activation signals at layer $l$ corresponding to inputs $a$ and $b$, computed before applying the activation function.
- $\chi_1 \equiv \partial c_{ab}^{(l+1)}/\partial c_{ab}^{(l)}|_{c=1}$: derivative of the correlation coefficient at layer $l+1$ w.r.t. layer $l$, at the fixed point $c = 1$. This value determines the gradient vanishing/explosion for bounded activation functions.
- $\tilde{\chi}_1 \equiv \partial q_{ab}^{(l+1)}/q_{ab}^{(l)}|_{c=1}$: derivative of the covariance among two inputs at layer $l+1$ w.r.t. layer $l$, at the fixed point $c = 1$. This value determines the gradient vanishing/explosion for unbounded activation functions.
- $\sigma_{\mu^{(l)}}^2 \equiv \mathrm{Var}_{\mathcal{D}}\left(y_i^{(l)}\right)$: signal (pre-activation) variance w.r.t. to the dataset at layer $l$. In the Supplementary Material, we show that this quantity does not depend on neuron $i$.
- $\sigma_{y^{(l)}}^2 \equiv \mathrm{Var}_{\mathcal{W}}\left(\left\langle y_i^{(l)}\right\rangle\right)$: signal (pre-activation) variance of the signal centers, i.e. the signal averaged w.r.t. dataset.
- $\gamma^{(l)} \equiv \sigma_{\mu^{(l)}}^2/\sigma_{y^{(l)}}^2$: the ratio between the signal variance w.r.t. the dataset and the variance of the signal centers w.r.t. the weights ensemble.
- $\delta_i^{(l)}(a) \equiv \frac{\partial \mathcal{L}}{\partial y_i^{(l)}}(a)$: derivative of the loss w.r.t. the pre-activation.
- $\tilde{q}_{ab}^{(l)} \equiv \overline{\delta_i^{(l)}(a)\delta_i^{(l)}(b)}$: correlation between the gradient computed with inputs $a$ and $b$.

# B    TRAINING EXPERIMENTS

In this section, we provide detailed information on the training experiments we performed, reported in Tab. 2. We employ the following datasets: binarized Fashion MNIST (BFMNIST), binarized CIFAR10 (BCIFAR), CIFAR10, CIFAR100. We binarized Fashion MNIST and CIFAR10 by classifying odd versus even classes.

| Architecture | Depth | Width | Act. Function | lr | Dataset |
|---|---|---|---|---|---|
| MLP | 100 | 1000 | Tanh, ReU | $10^{-7}$ | BFMNIST |
| MLP | 100 | 1000 | Tanh, ReLU | $10^{-5}$ | CIFAR10 |
| Residual MLP | 100 | 1000 | Tanh, ReLU | $10^{-7}$ | BFMNIST |
| Vanilla VIT | 100 | 300 | ReLU | $10^{-5}$ | CIFAR10 |
| Vanilla VIT | 100 | 300 | ReLU | $10^{-5}$ | CIFAR10 |
| Large VIT ($\sim 300$ M params.) | 97 | $\sim 5000$ | GeLU | $10^{-3}$ | CIFAR100 |

Table 2: Details on the training experiments performed in this work. The learning rates for each rows (lr) refer to different datasets.

We train all the models employing the Adam optimizer (Kingma & Ba, 2017) without any regularization. The training and test batch sizes are both set to 100. The learning rate is chosen according to the model and dataset, as reported in Tab. 2. In the next subsections, we report the global accuracy, the accuracy of the most favoured and the most unfavoured classes during training, and the maximum classification frequency, all computed at each training step. In particular, we observe the phase behaviour claimed in Prop. 4.1: phases that are the fastest to train correspond to a maximally biased state, with deep prejudice.

## B.1    MLP

We train a vanilla MLP (Eq. (11)) with ReLU and Tanh activation functions on binarized Fashion MNIST and CIFAR10. On BFMNIST, we can reach high accuracies (Fig. 6, Fig. 7), while the vanilla MLP reaches at most around $50\%$ test accuracy when trained on CIFAR10 (Fig. 8 and Fig. 9). We argue that this is due to the simplicity of the architecture, and training would require more adjustments to perform well even for more complex datasets. However, we can still observe the phase distinctions in the early phase of learning, with the EOC being the fastest to absorb the bias. Clearly, in the chaotic phases (chaotic-deep prejudice, prejudice and neutrality), learning is slowest where the initial ignorance of the class (i.e. neutrality in class assignment) hinders training. The ordered-deep prejudice phase seems to reach eventually the same performance as the EOC, but with a sudden performance jump. We argue that this is a combined effect of the optimizer employed and the convexity of the loss landscape, typical of over-parametrized models. In particular, when gradients are vanishing (as it happens in the ordered-deep prejudice phase), momentum-based optimizers like Adam can quickly navigate almost-flat loss landscapes, eventually reaching a (sub)-optimal plateau.

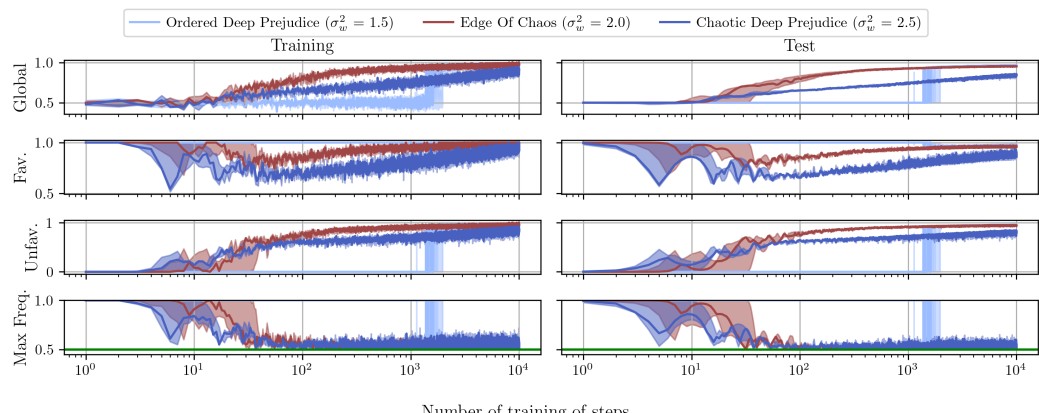

Figure 6: Global, favoured and unfavoured class train and test accuracies for a ReLU MLP trained on binarized fashion MNIST.

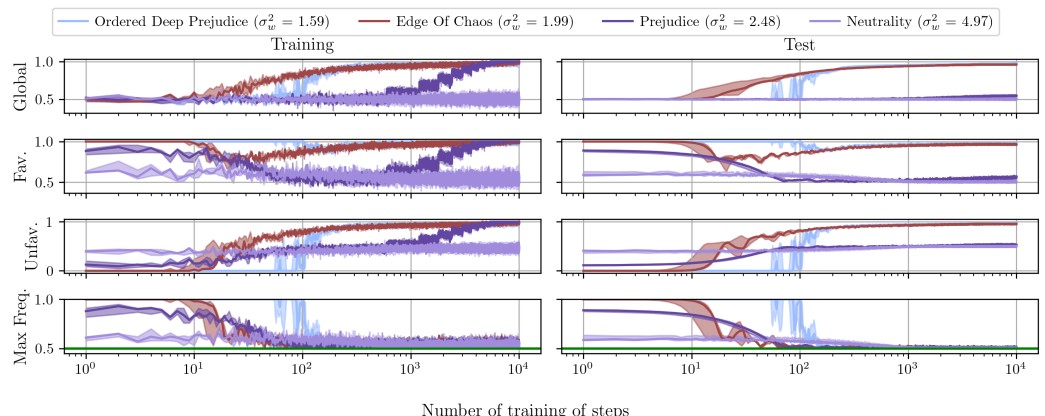

Figure 7: Global, favoured and unfavoured class train and test accuracies for a Tanh MLP trained on binarized fashion MNIST.

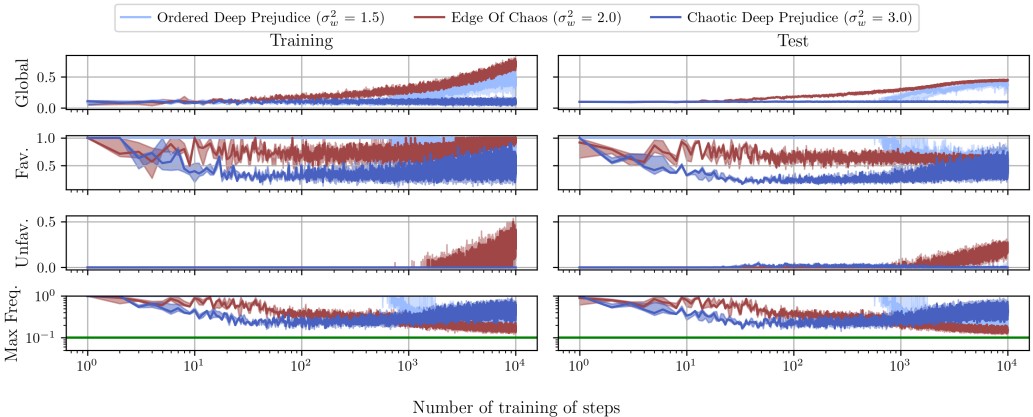

Figure 8: Global, favoured and unfavoured class train and test accuracies for a ReLU MLP trained on CIFAR10.

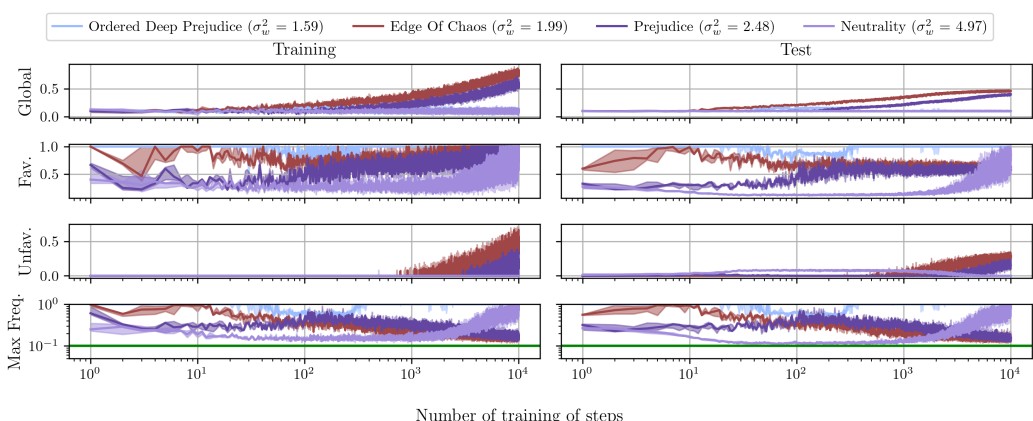

Figure 9: Global, favoured and unfavoured class train and test accuracies for a Tanh MLP trained on CIFAR10.

## B.2 RESIDUAL MLP

We employ a residual MLP studied in the MF regime by Yang & Schoenholz (2017), where the residual branch is rescaled by the total network's length in order to avoid trivial signal explosion. In particular, Yang & Schoenholz (2017) showed that residual MLP are *always* critical, i.e. there is no order/chaos phase transition; the correlation coefficient always converges to one at initialization and gradients are always stable (see Fig. 26). In this work, we corroborated this result by training Residual Tanh and ReLU MLPs on binarized Fashion MNIST (Fig. 10 and 11, respectively). Since these models are always critical, the experiments show deep prejudice at beginning of learning, which is absorbed at the same time in all the models, producing high classification accuracies.

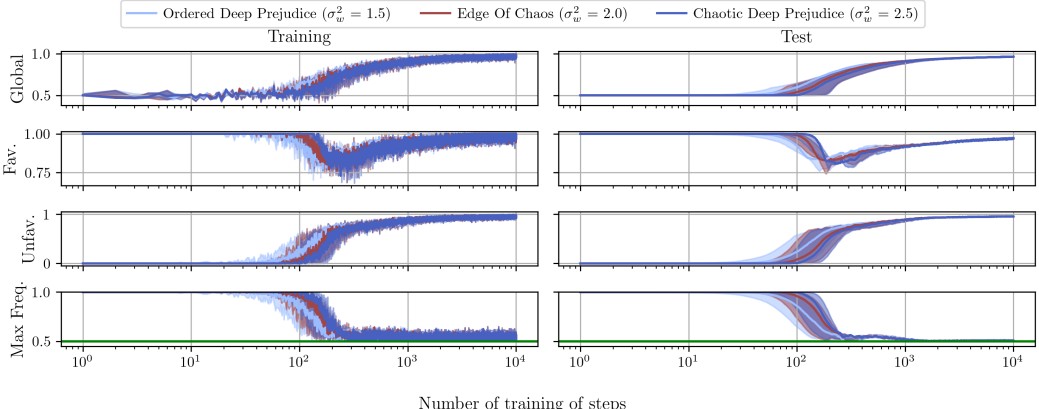

Figure 10: Global, favoured and unfavoured class train and test accuracies for a ReLU Residual MLP trained on binarized Fashion MNIST.

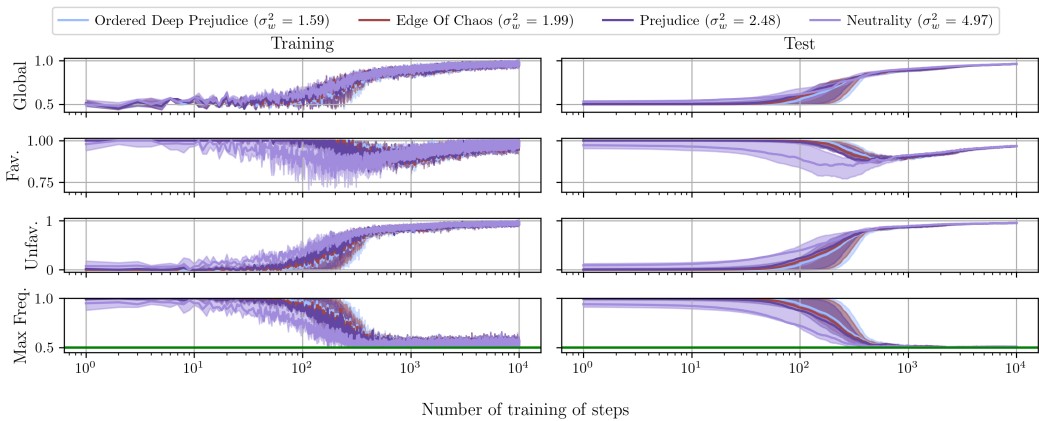

Figure 11: Global, favoured and unfavoured class train and test accuracies for a Tanh Residual MLP trained on binarized Fashion MNIST.

### B.3 VANILLA VISION TRANSFORMER

We employ a vanilla Vision Transformer (VIT) (Dosovitskiy et al., 2020) for classification trained on binarized CIFAR10 (Fig. 12) and CIFAR10 (Fig. 13). We eliminate all batch- and layer-norm layers, since they are not included in our theory and could potentially destroy the phase distinctions. We preserve only linear and convolutional layers, which we initialize with the prescription of this work. In particular, this assumption is justified since the MF theory of convolutional layer is identical to that of MLP (Xiao et al., 2018). We initialize the attention matrices as the linear layer, with the *gain* depending on the activation function placed before the activation. This way, we show that gradients manifest the same transition behaviour observed in vanilla models, where an exact theory is available (see for instance Fig. 27).

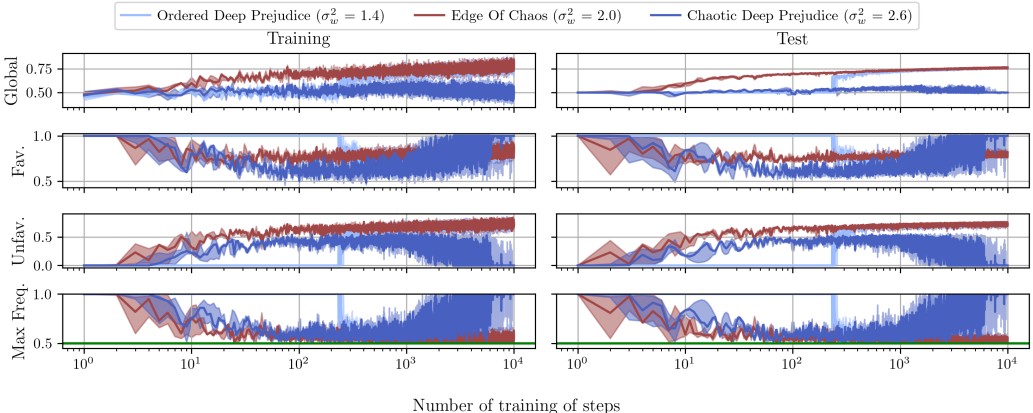

Figure 12: Global, favoured and unfavoured class train and test accuracies for a ReLU Vision Transformer trained on binarized CIFAR10.

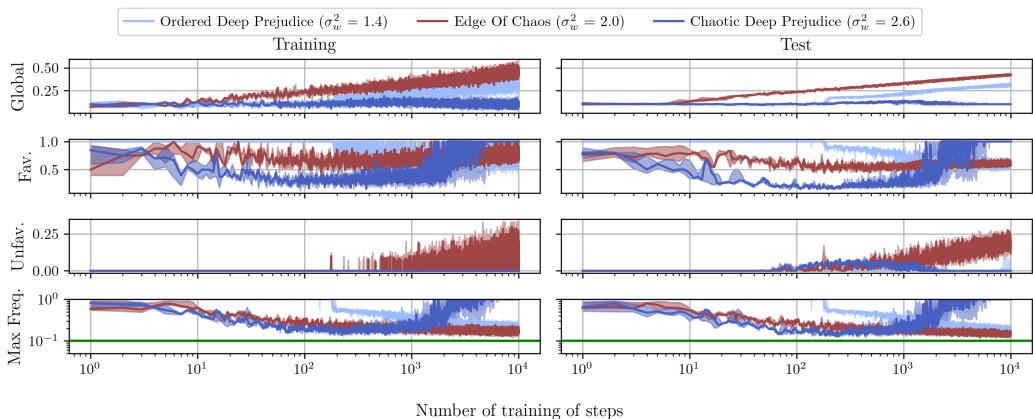

Figure 13: Global, favoured and unfavoured class train and test accuracies for a ReLU Vision Transformer trained on CIFAR10.

### B.4 LARGE VISION TRANSFORMER

Finally, we validate our assumption by fine-tuning a large model, without any architectural modification. We download a pre-train large Vision Transformer (vit_large_patch16_384) from the library Timm (Wightman, 2019). This model was pre-trained on ImagenNet-21k, and consists of about 300 million parameters. We fine tune this model on CIFAR100. We multiply the weights of all linear and convolutional layers by a factor $\sigma_w^2$, with the goal of triggering the order/chaos phase transition studied in this paper. This way, the pre-trained model behaves like an untrained model when evaluated on a statistically different dataset. Indeed, we show that gradients behave like at the phase boundary between order and chaos (Fig. 28). We report the training dynamics in Fig. 14, where we observe that the original unscaled state (at EOC - red line) manifests the fastest learning dynamics. In particular, it begins with moderate prejudice and it is the only state significantly improving the classification accuracy of the most unfavoured class. Instead, the state rescaled with $\sigma_w^2 = 1.5$, ideally in the chaotic phase, begins with a lower amount of prejudice, but it cannot recover. Finally, the state rescaled by $\sigma_w^2 = 0.5$ shows strong initialization prejudice, typical of the ordered phase, but smaller gradients (Fig. 28), which hinder training.

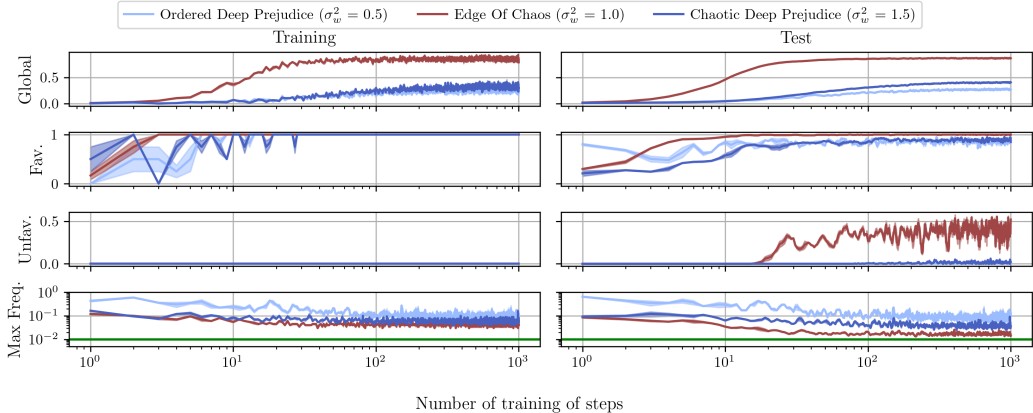

Figure 14: Global, favoured and unfavoured class train and test accuracies for a large pre-trained Vision Transformer fine-tuned on CIFAR100.

## C    Mean field results

In this section, we report some results of previous MF theory, starting from wide MLPs (Poole et al., 2016; Schoenholz et al., 2016; Hayou et al., 2019).

We report the derivation of the following recursive equations for the signal variance and covariance

$$q_{aa}^{(l)} = \sigma_w^2 \int \mathcal{D}y \, \phi\left(u^{(l-1)}\right)^2 + \sigma_b^2 \ , \tag{7}$$

$$q_{ab}^{(l)} = \sigma_w^2 \int \mathcal{D}y \, \mathcal{D}y' \, \phi\left(u^{(l-1)}\right) \phi\left(u'^{(l-1)}\right) + \sigma_b^2 \ . \tag{8}$$

where $\mathcal{D}y = \frac{dy}{\sqrt{2\pi}} e^{-y^2/2}$ denotes the standard Gaussian measure, and

$$u^{(l)} \equiv \sqrt{q_{aa}^{(l)}} y \ , \tag{9}$$

$$u'^{(l)} \equiv \sqrt{q_{bb}^{(l)}} \left(c_{ab}^{(l)} y + \sqrt{1 - (c_{ab}^{(l)})^2} y'\right) \ . \tag{10}$$

To prove them, let us consider the signal propagation through a MLP, which reads

$$y_i^{(l)}(a) = \sum_{j=1}^{N_l} W_{i,j}^{(l)} \phi\left(y_j^{(l-1)}(a)\right) + b_i^{(l)} \ , \tag{11}$$

where for the first layer we have

$$y_i^{(1)}(a) = \sum_{j=1}^{d} W_{i,j}^{(1)} \xi_j(a) + b_i^{(1)} \ , \tag{12}$$

where $\xi_j(a)$ is the $j$-th component of the $a$-th data instance. We consider the ensemble of MLPs over weights and biases ($\mathcal{W}$) initialized according to the following scheme:

$$p_{W_{i,j}^{(l)}}(x) = \mathcal{N}\left(x; 0, \frac{\sigma_{w^{(l)}}^2}{N_l}\right) \quad \forall i, j = 1, \ldots, N_l \ , \tag{13}$$

$$p_{b_i^{(l)}}(x) = \mathcal{N}\left(x; 0, \sigma_{b^{(l)}}^2\right) \quad \forall \, i = 1, \ldots, N \ . \tag{14}$$

The signal variance is defined as

$$q_{aa}^{(l)} \equiv \frac{1}{N} \sum_{i=1}^{N} \left(y_i^{(l)}(a)\right)^2 \ . \tag{15}$$

In the limit of large width ($N \to \infty$), the distributions of the pre-activations $y_i^{(l)}$ converge to i.i.d. zero mean Gaussians, since weights and biases are all independent across neurons and layers, and $y_i^{(l)}$ is a weighted sum of a large number of uncorrelated random variables. This treatment is valid as long as we do not impose any distribution on the input dataset $\mathcal{D}$, but consider only averages over $\mathcal{W}$. By applying Eq. (11) to Eq. (15) and using the definition of the weights and biases distribution (Eqs. (13) and 14), we easily get

$$q_{aa}^{(l)} = \sigma_w^2 \frac{1}{N} \sum_{i=1}^{N} \phi\left(y_i^{(l-1)}(a)\right)^2 + \sigma_b^2 \ . \tag{16}$$

Since the empirical distribution across the layer $l - 1$ is a zero mean Gaussian with variance given by $q_{aa}^{(l-1)}$, in the large-width limit, we can substitute the empirical distribution with an integral over a Gaussian variable. In this regime, the distribution of signals across neurons of a single MLP converges to the distribution of signals of a single neuron across the random ensemble; this is known as *self-averaging* assumption from statistical physics of disordered systems (which is formally true in the large-width limit). This Gaussian variable can be re-parametrized and finally we get an integral over a standard Gaussian variable $y$ as

$$q_{aa}^{(l)} = \sigma_w^2 \int \mathcal{D}y \, \phi\left(u^{(l-1)}\right)^2 + \sigma_b^2 \ . \tag{17}$$

The covariance among inputs is defined as

$$q_{ab}^{(l)} \equiv \frac{1}{N} \sum_{i=1}^{N} y_i^{(l)}(a) y_i^{(l)}(b) . \tag{18}$$

The joint empirical distribution of $y_i^{(l)}$ and $y_i^{(l)}$ converges at large $N$ to a 2-dimensional Gaussian with covariance $q_{ab}^{(l)}$. Similarly as for the signal variance, we can find a recursive equation for $q_{ab}^{(l)}$ as

$$q_{ab}^{(l)} = \sigma_w^2 \int \mathcal{D}y \, \mathcal{D}y' \, \phi\left(u^{(l-1)}\right) \phi\left(u'^{(l-1)}\right) + \sigma_b^2 . \tag{19}$$

Let us denote with $q$ the limiting variance in its domain of convergence (Hayou et al., 2019). We can show (Schoenholz et al., 2016) that:

$$\chi_1 \equiv \left. \frac{\partial c_{ab}^{(l+1)}}{c_{ab}^{(l+1)}} \right|_{c=1} = \sigma_w^2 \int \mathcal{D}y \, \phi'\left(\sqrt{q}y\right)^2 , \tag{20}$$

and

$$\alpha \equiv \frac{\partial q_{aa}^{(l+1)}}{\partial q_{aa}^{(l)}} = \chi_1 + \sigma_w^2 \int \mathcal{D}y \phi\left(\sqrt{q}y\right) \phi''\left(\sqrt{q}y\right) . \tag{21}$$

In previous MF works, it was not clear the role played by $\chi_1$, because it is usually derived by assuming the variance to converge faster than the correlation coefficient (Poole et al., 2016; Schoenholz et al., 2016). Here, we prove that when the variance is not assumed to be constant, the result slightly changes, suggesting that for some activation functions, the asymptotic correlation coefficient is not able to discriminate between phases. First, we prove the following Lemma.

**Lemma C.1.** *If the variance does not converge, we can compute*

$$\chi_1^{(l)} = \frac{q^{(l)}}{q^{(l+1)}} \int \mathcal{D}y \, \phi'\left(\sqrt{q^{(l)}}y\right)^2 , \tag{22}$$

*where $q_{aa}^{(l)} \equiv q^{(l)}$, $\forall a \in \mathcal{D}$.*

*Proof.* Let us compute $\frac{\partial c_{ab}^{(l+1)}}{\partial c_{ab}^{(l)}}$. For generic activation functions, $q_{aa}^{(l)}$ may diverge with depth and thus it cannot be safely kept constant as done in the MF literature for bounded activation functions (Schoenholz et al., 2016). Moreover, due to our main result (reported as Thm. E.1), for large $N$, $q_{aa}^{(l)} = q_{bb}^{(l)}$, $\forall a, b \in \mathcal{D}, \forall l$; thus, we can write $q_{aa}^{(l)} \equiv q^{(l)}$, $\forall a \in \mathcal{D}$. From Eq. (8) we can directly calculate:

$$\frac{\partial c_{ab}^{(l+1)}}{\partial c_{ab}^{(l)}} = \frac{\sigma_w^2}{q^{(l+1)}} \int \mathcal{D}y \, \mathcal{D}y' \, \phi(u) \, \phi'(u') \sqrt{q^{(l)}} \left[ y - \frac{c_{ab}^{(l)}}{\sqrt{1 - (c_{ab}^{(l)})^2}} y' \right] , \tag{23}$$

where $u^{(l)}, u'^{(l)}$ have been defined in Eqs. (9) and 10.

Next, we proceed using the following key identity known as Stein's lemma (Stein, 1956):

$$\int \mathcal{D}y \, F(y) \, y = \int \mathcal{D}y \, F'(y) , \tag{24}$$

which holds for any function $F(y)$, where $y$ is a standard Gaussian variable (zero mean and unit standard variance).

Using this key identity and the definition of $u^{(l)}$ and $u'^{(l)}$ in Eqs. (9) and (10), we get (omitting their $l$-dependency for simplicity)

$$\int \mathcal{D}y \, \mathcal{D}y' \, \phi(u) \, \phi'(u') y = \sqrt{q^{(l)}} \int \mathcal{D}y \, \mathcal{D}y' \left[ \phi'(u) \phi'(u') + c_{ab}^{(l)} \phi(u) \phi''(u') \right] , \tag{25}$$

$$\int \mathcal{D}y \, \mathcal{D}y' \, \phi(u) \, \phi'(u') y' = \sqrt{1 - (c_{ab}^{(l)})^2} \sqrt{q^{(l)}} \int \mathcal{D}y \, \mathcal{D}y' \, \phi(u) \phi''(u') . \tag{26}$$

Therefore, by combining Eqs. (25) and (26) with Eq. (23), we get

$$\frac{\partial c_{ab}^{(l+1)}}{\partial c_{ab}^{(l)}} = \frac{q^{(l)}}{q^{(l+1)}} \sigma_w^2 \int \mathcal{D}y \, \mathcal{D}y' \, \phi'(u) \, \phi'(u') \ . \tag{27}$$

At the critical point $c = 1$, the former expression further simplifies to

$$\chi_1^{(l)} \equiv \left. \frac{\partial c^{(l+1)}}{\partial c^{(l)}} \right|_{c=1} = \frac{q^{(l)}}{q^{(l+1)}} \tilde{\chi}_1^{(l)} \ , \tag{28}$$

where

$$\tilde{\chi}_1^{(l)} \equiv \left. \frac{\partial q_{ab}^{(l+1)}}{\partial q_{ab}^{(l)}} \right|_{c=1} = \sigma_w^2 \int \mathcal{D}y \, \phi'\left(u^{(l)}\right)^2 \ . \tag{29}$$

$\square$

Specifically, for ReLU activations we can prove the following Lemma.

**Lemma C.2.** *For ReLU, it holds that*

$$\chi_1^{(l)} = 1 - \frac{\sigma_b^2}{q^{(l+1)}} \ . \tag{30}$$

*Proof.* For ReLU we get $\int \mathcal{D}y \, [\phi'\left(u^{(l)}\right)]^2 = \frac{1}{2}$ as long as the variance $q^{(l)}$ is finite, and $q^{(l+1)} = \frac{\sigma_w^2}{2} q^{(l)} + \sigma_b^2$, which implies that

$$\chi_1^{(l)} = 1 - \frac{\sigma_b^2}{q^{(l+1)}} \le 1 \ , \tag{31}$$

and the fixed point $c = 1$ is never repelling. Only for bounded activation function, the variance $q^{(l)}$ always converges, therefore $\chi_1 = \sigma_w^2 \int \mathcal{D}y \, [\phi'(u)]^2$ and the definition agrees with MF theory. Hence

$$\lim_{l \to \infty} \chi_1^{(l)} = \begin{cases} 1 & \textit{if } \sigma_b^2 = 0 \textit{ or } \sigma_w^2 \ge 2, \forall \sigma_b^2 \ , \\ < 1 & \textit{else} \ . \end{cases} \tag{32}$$

By using the IGB framework, in App. H we show that this implies that for ReLU the correlation coefficient (resptc. $\gamma^{(l)}$) converges exponentially (respct. diverges) in the order phase, while the chaotic phase (and the line $\sigma_b^2 = 0$) is all critical and the correlation coefficient converges sub-exponentially. $\square$

In App. H we find explicit recursion relations for quantities of interest for ReLU by using the IGB approach, corroborating the divergence behaviour of $\gamma^{(l)}$ in different phases.

## D   IGB EXTENSION TO EXPLICIT INITIALIZATION BIASES

The standard MF approach takes into account only one source of randomness, coming from the network ensemble, whereas the input is fixed. Here, we extend the analysis to the case where the dataset $\mathcal{D}$ is randomly distributed. For simplicity, we suppose that each datapoint follows a standard Gaussian distribution, *i.e.* $\xi_j(a) \sim \mathcal{N}(0, 1)$. For random datasets, it is meaningful to define an averaging operator over fixed weights and biases.

---

**Definition D.1** (Averages over the dataset). The average over data $\mathcal{D}$ at fixed weights and biases $\mathcal{W}$ is denoted with $\langle x \rangle \equiv \mathbb{E}_{\mathcal{D}} (x | \mathcal{W})$.

---

Ref. Francazi et al. (2024) derived the pre-activation distributions of a MLP processing a random dataset in case of zero explicit initialization biases (Thm D.2, Appendix). Here, we generalize it to accomplish non-zero initialization biases.

---

**Theorem D.2** (IGB pre-activation distributions). *When averages over the dataset are taken first, in the limit of infinite width and then data, the pre-activation $y_i^{(l)}$ are independently Gaussian distributed as:*

$$p_{y_i^{(l)}}^{(\mathcal{D})} (x) = \mathcal{N} \left( x; \mu_i^{(l)}, \sigma_{y^{(l)}}^2 \right) , \ \forall i = 1, \dots, N ,\tag{33}$$

*where $\sigma_y^{(l)}$ is the node variance. $\{\mu_i^{(l)}\}_{i=1,\dots,N}$ are independent random variables which depend on $\mathcal{W}$ only and are distributed according to zero mean Gaussian distribution:*

$$p_{\mu_i^{(l)}}^{(\mathcal{W})} (x) = \mathcal{N} \left( x; 0, \sigma_{\mu^{(l)}}^2 \right) , \forall i = 1, \dots, N ,\tag{34}$$

*where $\sigma_{\mu^{(l)}}^2$ is the variance of the centers of the node signals.*

---

*Proof.* The result in Thm D.2 extends directly from prior work that considered the same setting but with zero bias terms. In particular, Ref. Francazi et al. (2024) proved that under the assumption of i.i.d. Gaussian data, fixed weights and large layer, the pre-activations

$$y_i^{(l)} = \sum_{j=1}^{N} W_{i,j}^{(l)} \phi \left( y_j^{(l-1)} \right)\tag{35}$$

are i.i.d. normally distributed with mean $\mu_i^{(l)}$ and variance $\sigma_{y^{(l)}}^2$. When considering the variability over the weights, $\mu_i^{(l)}$ is also normally distributed with mean zero and variance $\sigma_{\mu^{(l)}}^2$. Moreover, $\sigma_{y^{(l)}}^2$ is self-averaging with respect to the weights, *i.e.* $\sigma_{y^{(l)}}^2 = \overline{\sigma_{y^{(l)}}^2}$. To extend this to the setting with non-zero bias terms, observe that the bias enters as an additive random variable that is itself Gaussian and independent of the weighted sum in Eq. 35. Therefore, the resulting pre-activations remain Gaussian, as the sum of independent Gaussian variables is Gaussian. □

The key difference compared to the MF approach is that, in the IGB approach, the pre-activation distributions are not centred around zero. Moreover, the variability of the dataset can be captured solely by the variance of the nodes $\sigma_y^2$, whereas the ensemble variability is fully characterized by the variance of the centers $\sigma_\mu^2$.

**Lemma D.3** (IGB recursion formulas, informal). *For a general MLP (Eq. (11)), the signal variance and the centers variance satisfy the following recursive equations:*

$$\sigma^2_{y^{(l+1)}} = \sigma^2_w Var_{\mathcal{D}}\left(\phi\left(y^{(l)}\right)\right) , \tag{36}$$

$$\sigma^2_{\mu^{(l+1)}} = \sigma^2_w \overline{\left\langle\phi\left(y^{(l)}\right)\right\rangle^2} + \sigma^2_b , \tag{37}$$

*with initial values* $\sigma^2_{y^{(0)}} = 1$ $\sigma^2_{\mu^{(0)}} = 0$. *Moreover,* $Var_{\mathcal{D}}(\phi(y^{(l)}))$ *is self-averaging with respect to* $\mathcal{W}$*, that is* $Var_{\mathcal{D}}(\phi(y^{(l)})) = \overline{Var_{\mathcal{D}}(\phi(y^{(l)}))}$ .

*Proof.* We now want to prove the recursive relations for $\sigma^2_{y^{(l)}}$ and $\sigma^2_{\mu^{(l)}}$, *i.e.* Eq. (36) and Eq. (37), respectively. By defining $\phi_i^{(l)} \equiv \phi\left(y_i^{(l)}\right)$, we compute the covariance (with respect to the input data) of the generic layer as:

$$\text{Cov}_{\mathcal{D}}\left(y_i^{(l+1)}, y_j^{(l+1)}\right) = \sum_{k,p=1}^N W_{i,k}^{(l)} W_{j,p}^{(l)} \left\langle\phi_p^l \phi_k^{(l)}\right\rangle + \sum_{k=1}^N W_{i,k}^{(l)} \left\langle\phi_k^{(l)}\right\rangle b_j^{(l)} + \sum_{k=1}^N W_{j,k}^{(l)} \left\langle\phi_k^{(l)}\right\rangle b_i^{(l)} +$$

$$+(b_i^{(l)})^2 - \sum_{k,p=1}^N W_{i,k}^{(l)} W_{j,p}^{(l)} \left\langle\phi_p^{(l)}\right\rangle\left\langle\phi_k^{(l)}\right\rangle - \sum_{k=1}^N W_{i,k}^{(l)} \left\langle\phi_k^{(l)}\right\rangle b_j^{(l)} - \sum_{k=1}^N W_{j,k}^{(l)} \left\langle\phi_k^{(l)}\right\rangle b_i^{(l)} - (b_i^{(l)})^2$$

$$= \sum_{k,p=1}^N W_{i,k}^{(l)} W_{j,p}^{(l)} \text{Cov}_{\mathcal{D}}\left(\phi_k^{(l)}, \phi_p^{(l)}\right) . \tag{38}$$

Ref. Francazi et al. (2024) proved that in the large width limit $\text{Var}_{\mathcal{D}}\left(y_i^{(l)}\right)$ is self-averaging (as distribution of the weights) and does not depend on $i$, *i.e.*

$$\lim_{N\to\infty} \overline{\text{Var}_{\mathcal{D}}\left(y_i^{(l)}\right)} = \text{Var}_{\mathcal{D}}\left(y^{(l)}\right) , \tag{39}$$

and that $\text{Cov}_\chi\left(\phi_k^{(l)}, \phi_p^{(l)}\right) = \delta_{p,k}\text{Var}_{\mathcal{D}}\left(\phi^{(l)}\right)$. Self-averaging implies also that $\lim_{N\to\infty} \sum_{k,p=1}^N W_{i,k}^{(l)} W_{i,p}^{(l)} = \sigma^2_w$, which together with Eq. (38) yields Eq. (36).

Now let us consider the calculation for $\sigma^2_{\mu^{(l)}}$. From Eq. (11) we easily see that $\overline{\left\langle y_i^{(l+1)}\right\rangle} = 0$ and

$$\overline{\left\langle y_i^{(l+1)}\right\rangle^2} = \overline{\left(\sum_{j=1}^N W_{i,j}^{(l)} \left\langle\phi_j^{(l)}\right\rangle + b_i^{(l)}\right) \cdot \left(\sum_{k=1}^N W_{i,k}^{(l)} \left\langle\phi_j^{(l)}\right\rangle + b_i^{(l)}\right)} = \tag{40}$$

$$= \overline{\left[\sum_{j,k=1}^N W_{i,j}^{(l)} W_{i,k}^{(l)} \left\langle\phi_j^{(l)}\right\rangle\left\langle\phi_k^{(l)}\right\rangle + (b_i^{(l)})^2\right]} = \tag{41}$$

$$= \sigma^2_w \overline{\left\langle\phi^{(l)}\right\rangle^2} + \sigma^2_b , \tag{42}$$

which is Eq. (37). $\qquad\square$

**Lemma D.4** (Fraction of Inputs Classified to Reference Class). *Given a fixed initialization $\mathcal{W}$, the fraction of inputs classified into reference class $0$ is given by:*

$$G_0 \equiv \mathbb{P}\left(y_1^{(L)} > y_2^{(L)} \mid \delta(\mathcal{W})\right) = \Phi\left(\sqrt{\frac{\gamma^{(L)}}{2}}\delta\right) , \tag{43}$$

*where $y_1^{(L)}, y_2^{(L)}$ are the two output node pre-activations at layer L, $\Phi$ is the Gaussian cumulative function, and $\delta$ is a standard Gaussian variable.*

*Proof.* From Lemma D.2, $y_1^{(L)} - y_2^{(L)}$ follows a Normal distribution with mean $\mu_1^{(L)} - \mu_2^{(L)}$ and variance $2\sigma_{y^{(L)}}^2$. Moreover, $\mu_1^{(L)} - \mu_2^{(L)}$ follows a Normal distribution centred around zero and with variance $2\sigma_{\mu^{(L)}}^2$. Therefore

$$\mathbb{P}\left(y_1^{(L)} > y_2^{(L)} \mid \delta(\mathcal{W})\right) = 1 - \Phi\left(-\frac{\mu_1^{(L)} - \mu_2^{(L)}}{2\sigma_{y^{(L)}}}\right) , \tag{44}$$

where $\Phi$ is the Gaussian cumulative function. By reparametrization $\mu_1^{(L)} - \mu_2^{(L)} \equiv \sqrt{2}\sigma_{\mu^{(L)}}\delta$, where $\delta$ is a standard Gaussian variable. By definition $\Phi(x) = \frac{1}{2}\left[1 + \text{erf}(x)\right]$, where $\text{erf}$ is the error function. Since $\text{erf}(-x) = -\text{erf}(x)$ and $\gamma^{(L)} = \frac{\sigma_{\mu^{(L)}}^2}{\sigma_{y^{(L)}}^2}$, we finally get

$$G_0 = \frac{1}{2}\left[1 + \text{erf}\left(\sqrt{\frac{\gamma^{(L)}}{2}}\delta\right)\right] = \Phi\left(\sqrt{\frac{\gamma^{(L)}}{2}}\delta\right) . \tag{45}$$

$\square$

# E    THE EQUIVALENCE BETWEEN MF AND IGB IN THE LARGE-WIDTH LIMIT

**Theorem E.1.** *Consider a generic architecture $\mathcal{A}$ (Eq. (1)) in the mean-field regime. Let us suppose to fix the initial conditions of IGB and MF to be equal, i.e. $q_{aa}^{(0)} = 1, \forall a \in \mathcal{D}$ and $q_{ab}^{(0)} = 0, \forall a, b \in \mathcal{D}$, with $a \neq b$. Then in the infinite width and then data limit, and for every layer $l > 0$, the total variance in the IGB approach is equal to the signal variance in the MF approach:*

$$q_{aa}^{(l)} = \sigma_{\mu^{(l)}}^2 + \sigma_{y^{(l)}}^2 \, , \forall a \in \mathcal{D} \, . \tag{46}$$

*Moreover, the centers variance in the IGB approach is equal to the input covariance in the MF approach:*

$$q_{ab}^{(l)} = \sigma_{\mu^{(l)}}^2 \, , \forall a, b \in \mathcal{D}, a \neq b \, . \tag{47}$$

*Finally, the correlation coefficient is related to $\gamma$ as:*

$$c_{ab}^{(l)} = \frac{\gamma^{(l)}}{1 + \gamma^{(l)}} \, , \forall a, b \in \mathcal{D}, a \neq b \, . \tag{48}$$

*Proof.* We begin by relating the IGB parameters $\sigma_{\mu^{(l)}}^2$ and $\sigma_{y^{(l)}}^2$ to the signal covariance $q_{ab}^{(l)}$ and variance $q_{aa}^{(l)}$. From Eq. (33), the pre-activation for neuron $i$ at layer $l$ and data point $a$ can be written as

$$y_i^{(l)}(a) = \mu_i^{(l)} + \sigma_{y^{(l)}} \epsilon_i^{(l)}(a), \tag{49}$$

where $\epsilon_i^{(l)}(a)$ are i.i.d. standard Gaussian variables for each neuron and data point. This parametrization ensures

$$\left\langle y_i^{(l)} \right\rangle = \mu_i^{(l)}, \tag{50}$$

$$\left\langle \epsilon_i^{(l)} \right\rangle = 0, \tag{51}$$

Since $\epsilon_i^{(l)}(a)$ is i.i.d. for each neuron, its expectation over the ensemble, $\overline{\epsilon_i^{(l)}(a)}$, can be consistently estimated, in the wide layers limit, by an average over neurons within the layer, i.e.,

$$\overline{\epsilon_i^{(l)}(a)} = \lim_{N \to \infty} \frac{1}{N} \sum_{i=1}^{N} \epsilon_i^{(l)}(a). \tag{52}$$

As a consequence of the central limit theorem, the random variable $r = \frac{1}{N} \sum_{i=1}^{N} \epsilon_i^{(l)}$ is distributed according to the density $p_r(x) = \mathcal{N}(x; 0, 1/N)$ for large $N$, therefore self-averaging to zero in the infinite-width limit. With the help of a little Algebra we can write

$$q_{ab}^{(l)} = \sigma_{\mu^{(l)}}^2 + \sigma_{y^{(l)}}^2 s(a, b) \, ,$$
$$q_{aa}^{(l)} = \sigma_{\mu^{(l)}}^2 + \sigma_{y^{(l)}}^2 s(a, a) \, , \tag{53}$$

where we define

$$s^{(l)}(a, b) \equiv \frac{1}{N} \sum_{i=1}^{N} \epsilon_i^{(l)}(a) \epsilon_i^{(l)}(b). \tag{54}$$

It is easy to prove that for every dataset element $a, b$: $\left| q_{ab}^{(l)} \right| \leq \sqrt{q_{aa}^{(l)} q_{bb}^{(l)}}$; therefore $c_{ab}^{(l)}$ meaningfully defines the correlation coefficient between inputs.

**Lemma E.2.** $\left| c_{ab}^{(l)} \right| \leq 1$, *for every* $a, b \in \mathcal{D}$.

*Proof.* We want to prove that $\left| c_{ab}^{(l)} \right| \leq 1$, which is true if and only if $(q_{ab}^{(l)})^2 \leq q_{aa}^{(l)} q_{ab}^{(l)}$, which is equivalent to

$$
\begin{aligned}
2\sigma_{\mu^{(l)}}^2 \sigma_{y^{(l)}}^2 \mathrm{s}^{(l)}(a,b) + \sigma_{y^{(l)}}^4 (\mathrm{s}^{(l)}(a,b))^2 &\leq \sigma_{\mu^{(l)}}^2 \sigma_{y^{(l)}}^2 \left[ \mathrm{s}^{(l)}(a,a) + \mathrm{s}^{(l)}(b,b) \right] + \\
&+ \sigma_{y^{(l)}}^4 \mathrm{s}^{(l)}(a,a)\mathrm{s}^{(l)}(b,b) \ .
\end{aligned}
\tag{55}
$$

From the Cauchy-Schwarz inequality $\mathrm{s}^{(l)}(a,a)\mathrm{s}^{(l)}(b,b) \geq [\mathrm{s}^{(l)}(a,b)]^2$. Moreover, with a little Algebra we get

$$
\mathrm{s}^{(l)}(a,a) + \mathrm{s}^{(l)}(b,b) - 2\mathrm{s}^{(l)}(a,b) = \frac{1}{N} \sum_{i,j=1}^{N} \left[ \epsilon_i^{(l)}(a) - \epsilon_j^{(l)}(b) \right]^2 \geq 0 \ ,
\tag{56}
$$

which together with the previous Cauchy-Schwarz inequality implies Ineq. (55). $\qquad \square$

Now, we are interested in the distributions of $q_{ab}^{(l)}$ and $q_{aa}^{(l)}$ with respect to the dataset. $\epsilon_i^{(l)}(a)$ and $\epsilon_i^{(l)}(b)$ should be thought as two independent random samples from $\epsilon_i^{(l)}$. Consequently, $\epsilon_i^{(l)}(a)\epsilon_i^{(l)}(b)$ should be thought as the product of two independent standard Gaussian variables, whose mean is zero and variance is one. From a computational point of view, this product is obtained by independently varying the inputs $a$ and $b$. Accordingly, $\epsilon_i^{(l)}(a)\epsilon_i^{(l)}(a)$ is the square of a standard Gaussian, which follows a chi-squared distribution with one degree of freedom. Its mean is one and its variance is two. Therefore, we can fully characterized the variance and covariance in MF as random variables in function of a Gaussian distributed dataset. For large $N$, when $a \neq b$ we have $p_{\mathrm{s}^{(l)}(a,b)}(x) \approx \mathcal{N}(x; 0, 1/N)$, while for $a = b$, $p_{\mathrm{s}^{(l)}(a,a)}(x) \approx \mathcal{N}(x; 1, 2/N)$.

It follows that in the infinite-width limit, the signal variance and covariance are self-averaging with respect to the dataset, i.e $q_{ab}^{(l)} = \left\langle q_{ab}^{(l)} \right\rangle$ and $q_{aa}^{(l)} = \left\langle q_{aa}^{(l)} \right\rangle$. $\qquad \square$

We plot the absolute percentage error for the ReLU of $\gamma^{(l)}$, $q_{aa}^{(l)}$, and $q_{ab}^{(l)}$ as the network sizes increases in Figs. 15, 16, and 17, respectively. Moreover, we report the absolute percentage error of $q_{aa}^{(l)}$, and $q_{ab}^{(l)}$ for Tanh activation (Figs. 18, Fig. 19, respectively).

> **Proposition E.3.** *From a trainability perspective, the optimal initial condition (stable gradients) is not one of neutrality, but rather a state of transient deep prejudice.*

*Proof.* Thm. E.1 establishes the equivalence between MF and IGB in the infinte-width limit. In MF, optimal training conditions are to be found at the edge of chaos (EOC). In the IGB framework, the EOC corresponds to the deep prejudice state, since the asymptotic value of the correlation coefficient is one. Moreover, this prejudiced state is transient since at the EOC, gradients are stable and so the network can absorb the bias rapidly Hayou et al. (2019). $\qquad \square$

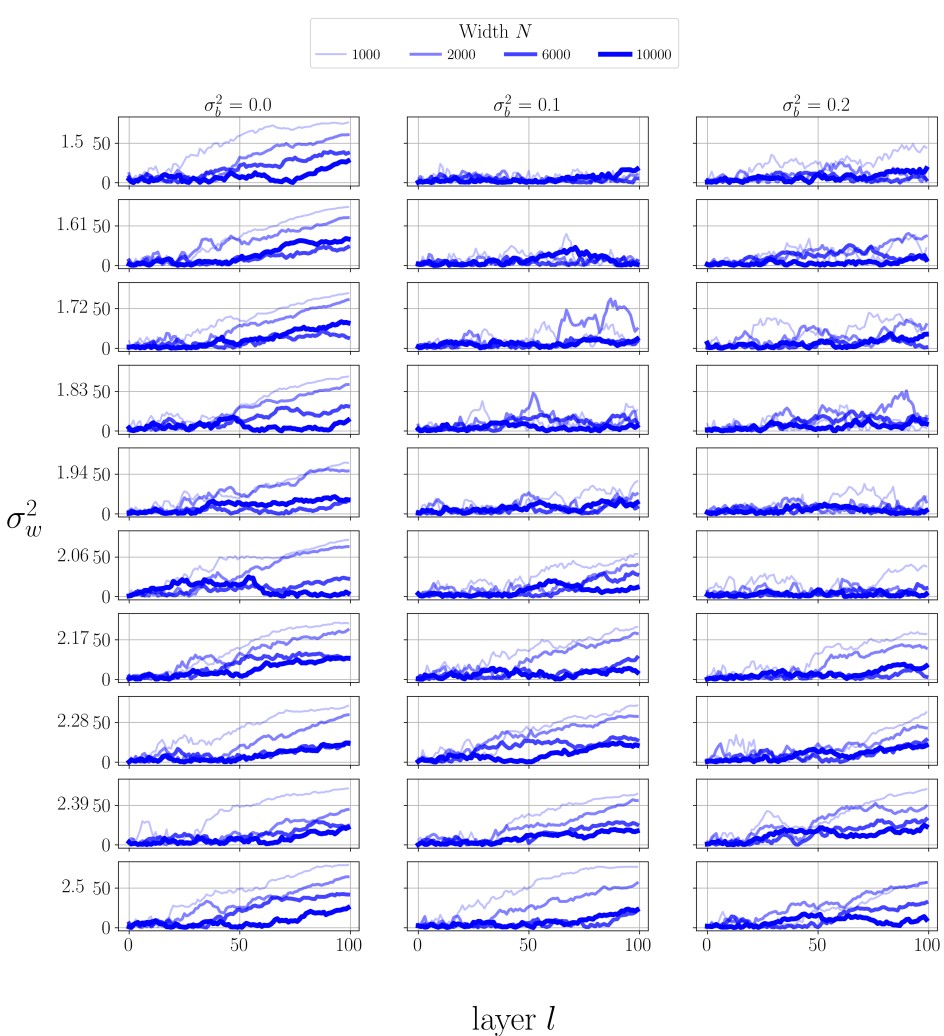

Figure 15: Absolute percentage error of the experimental versus theoretical values of $\gamma^{(l)}$ obtained for ReLU with different values of $\sigma_w^2$ close to the critical point $\sigma_w^2 = 2.0$. The width of the network varies from 1000 to 10000. We observe a reduction of the relative error as the network size increases, corroborating the theoretical curves shown in Fig. 31.

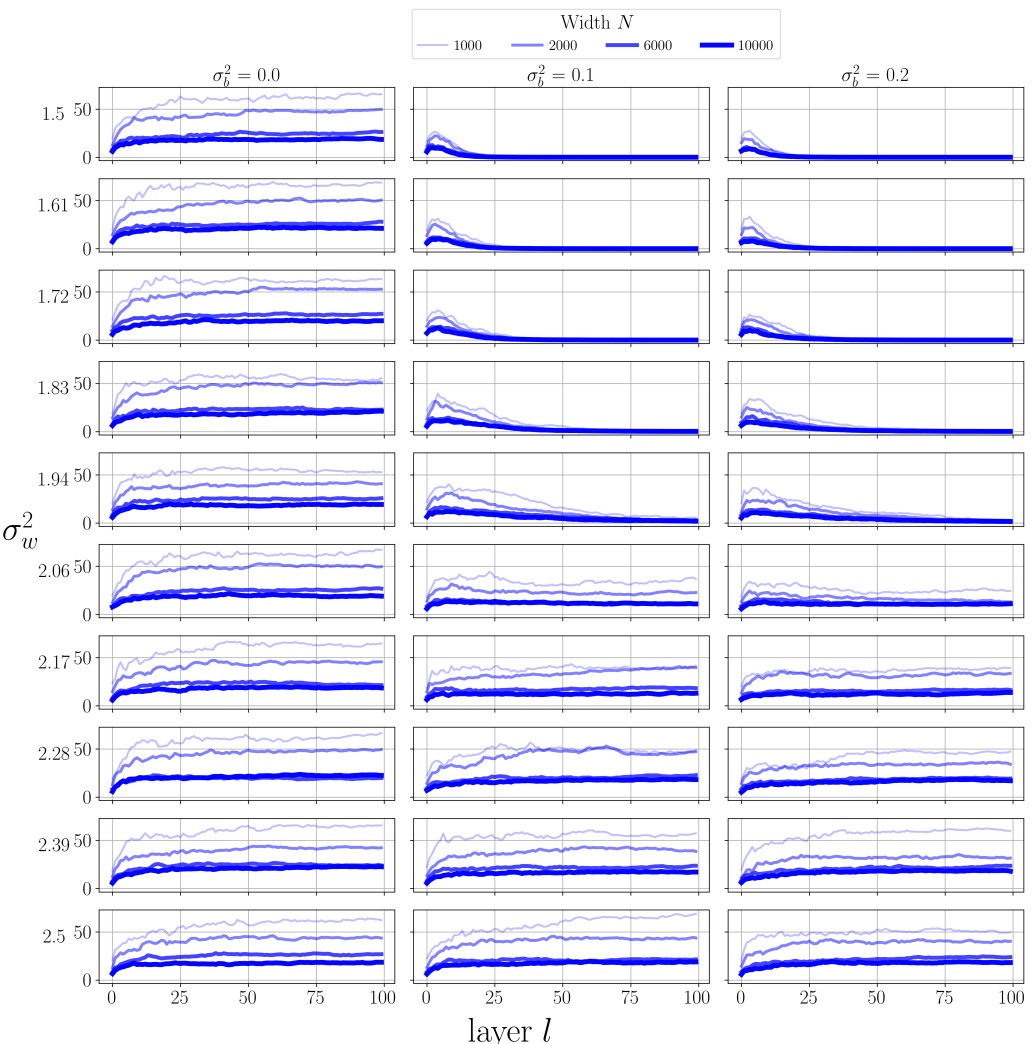

Figure 16: MF 90% confidence interval of the signal variance $q_{aa}^{(l)}$ in percentage of the median for ReLU activation. We compute it for a single MLP with increasing width inputted with 100 random data samples. We observe that the percentage deviation from the median decreases as the network width increases, corroborating the results of Thm. E.1.

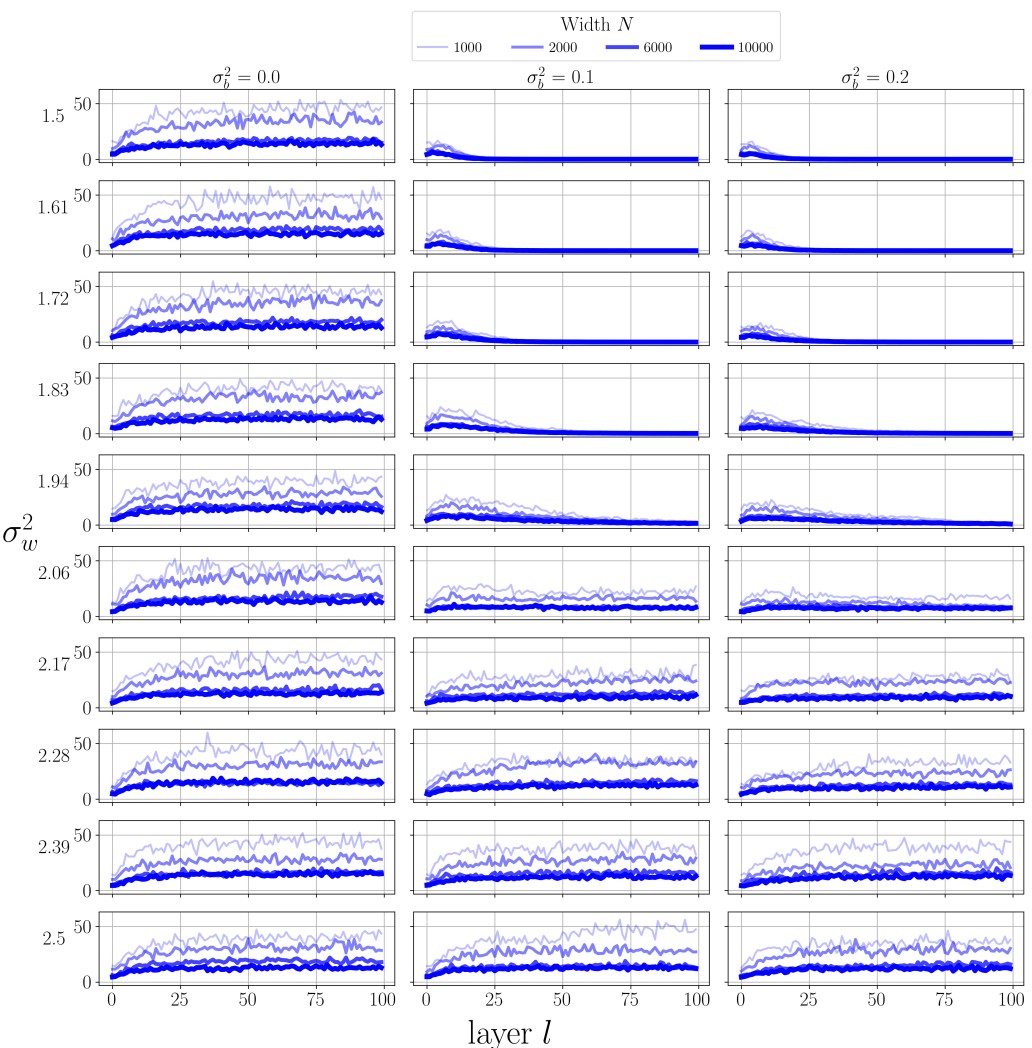

Figure 17: MF 90% confidence interval of the signal covariance $q_{ab}^{(l)}$ in percentage of the median for ReLU activation. We compute it for a single MLP with increasing width inputted with 100 random data samples. We observe that the percentage deviation from the median decreases as the network width increases, corroborating the results of Thm. E.1.

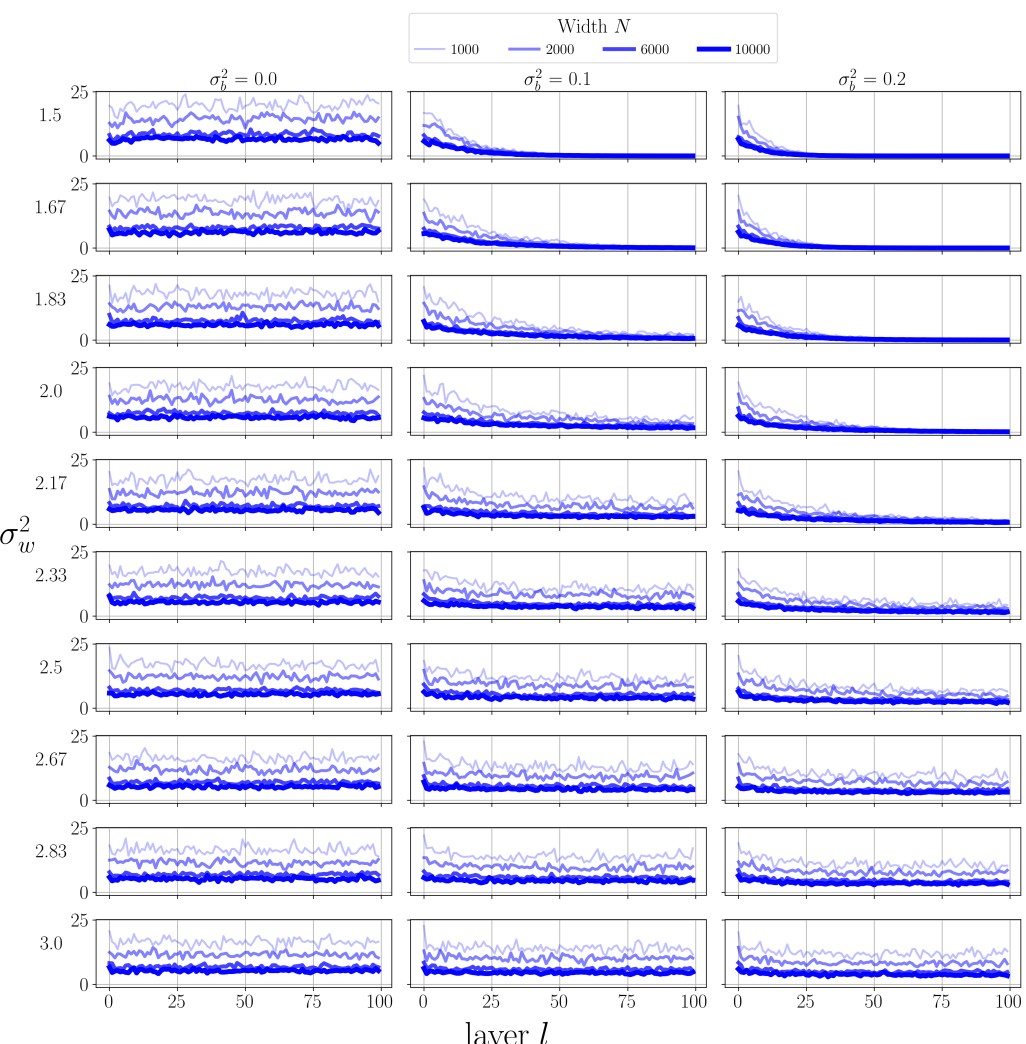

Figure 18: MF 90% confidence interval of the signal variance $q_{aa}^{(l)}$ in percentage of the median for Tanh activation. We compute it for a single MLP with increasing width inputted with 100 random data samples. We observe that the percentage deviation from the median decreases as the network width increases, corroborating the results of Thm. E.1.

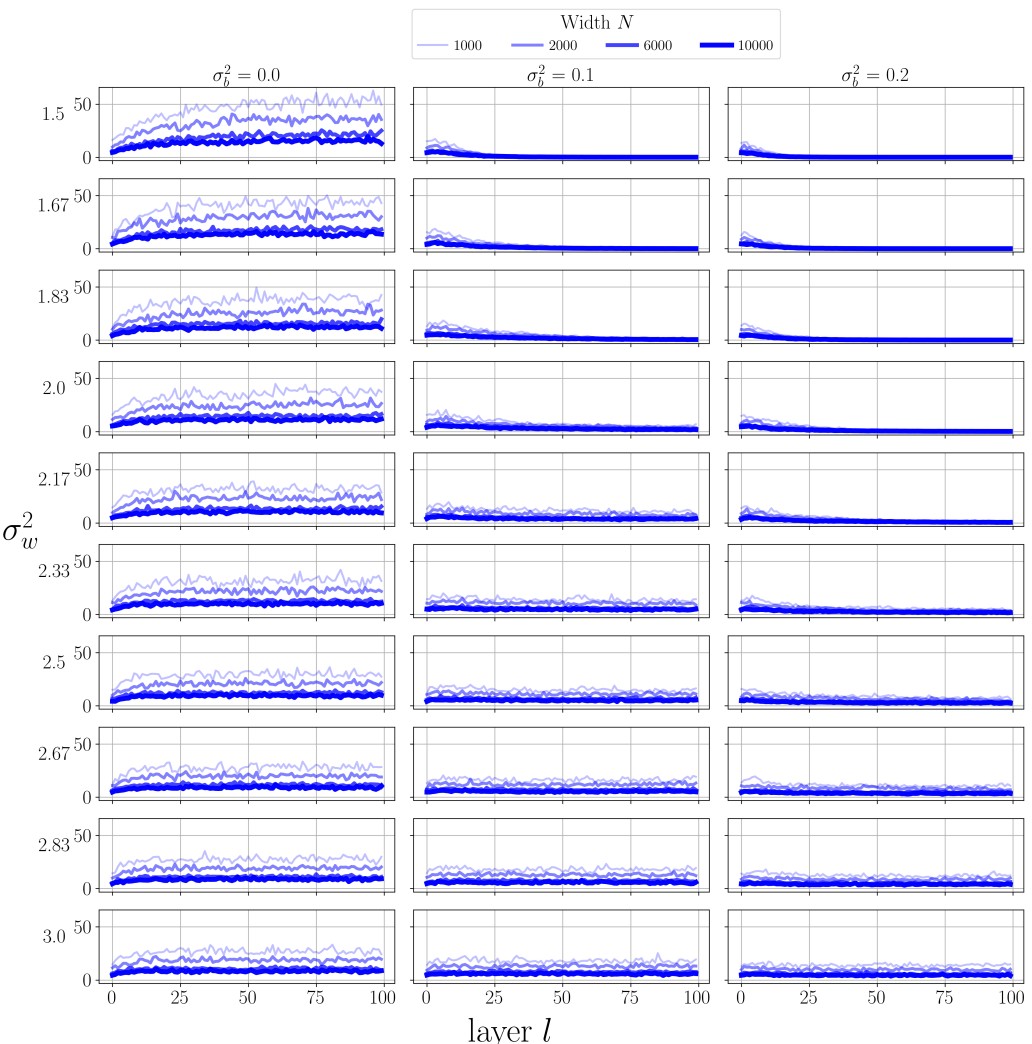

Figure 19: MF $90\%$ confidence interval of the signal covariance $q_{ab}^{(l)}$ in percentage of the median for Tanh activation. We compute it for a single MLP with increasing width inputted with 100 random data samples. We observe that the percentage deviation from the median decreases as the network width increases, corroborating the results of Thm. E.1.

## E.1 MF/IGB EQUIVALENCE FOR PRODUCTION ARCHITECTURES

To validate our theory in further realistic settings, we compute the correlation coefficient with both the IGB and MF theories for two production architectures downloaded from the Timm Python library: a RESNET with 18 layers (`https://huggingface.co/docs/timm/en/models/resnet` - Fig. 20) and a MLP mixer with a 100 layers (`https://huggingface.co/timm/mixer_b16_224.miil_in21k_ft_in1k` - Fig. 21). These figures should be interpreted in the same way as Fig. 2: the equivalence between the two frameworks is demonstrated by the fact that $c_{ab}^l$ has the same value independently of whether it is calculated through using the MF [Eq. (2)] or the IGB approach [Eq. (6)]. We employ the CIFAR10 dataset for these experiments.

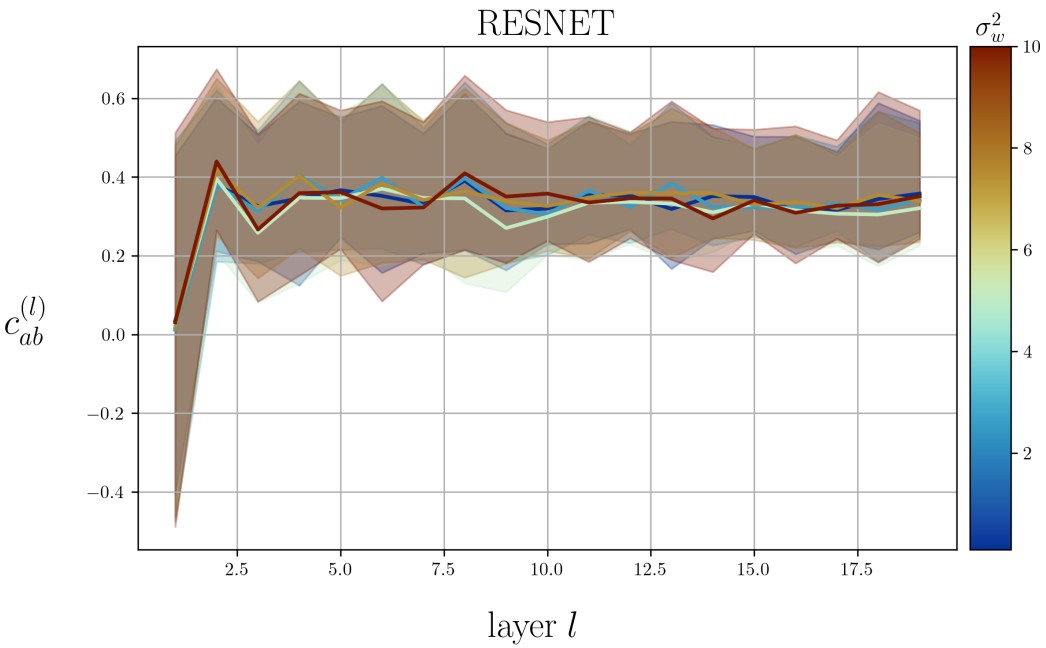

Figure 20: Behaviour the correlation coefficient of a production RESNET. Solid lines are computed using the IGB approach, while shaded areas represent the 90 % central confidence interval computed using the mean-field approach. The IGB solid lines are all within the corresponding MF interval, demonstrating the agreement between the two theories. Moreover, coherently with residual MLPs (see the Appendix B.2), we do not observe distinction in the predicted correlation coefficient at varying $\sigma_w^2$.

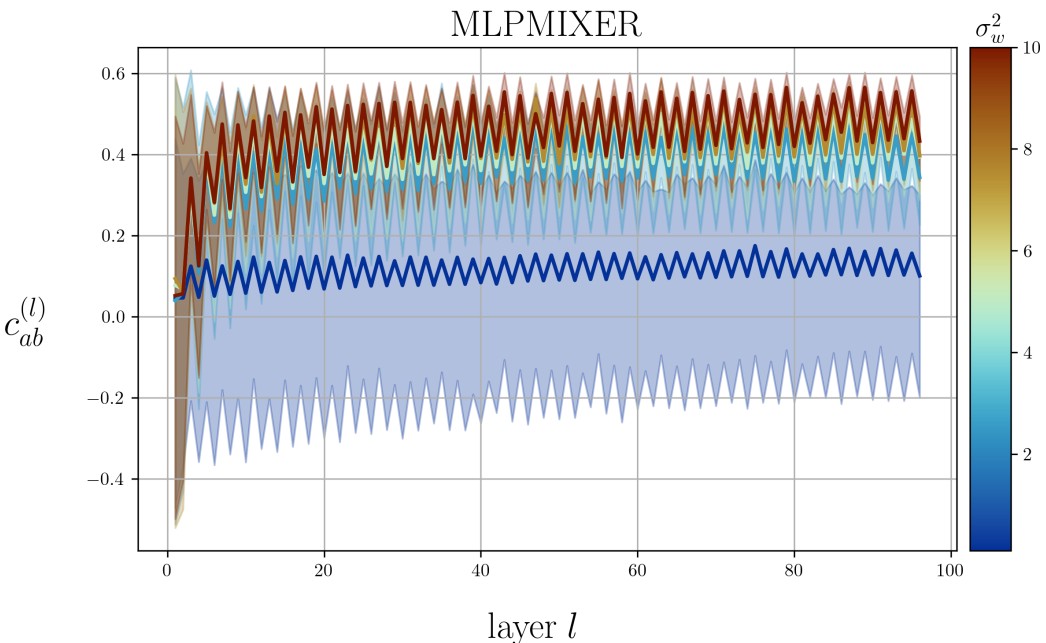

Figure 21: Same as Fig. 20 for a production MLP-MIXER with 100 layers at initialization. The agreement between IGB and MF is good across the entire phase diagram, being stronger as we increase the scaling factor of the weights $\sigma_w^2$.

# F MF/IGB EXTENSION TO MULTI-NODE ACTIVATION FUNCTIONS: THE EFFECT OF POOLING LAYERS

Within the MF literature, common architectural components such as batch normalization Yang et al. (2019) and dropout Schoenholz et al. (2016) have been extensively studied. In this work, however, we investigate the impact of multi-node pooling layers on the phase diagrams — a topic that has received comparatively little attention.

To this end, we leverage the IGB framework, which naturally extends to the general case of multi-node activation functions. We start by deriving recursive equations for a generic activation function $\phi : \mathbb{R}^n \to \mathbb{R}^n$, involving $n$ nodes. Once these general recursive relations are established, the theoretical results from (Hayou et al., 2019) remain applicable. In particular, we can directly employ Algorithm 1 from (Hayou et al., 2019) to compute the EOC.

---

**Lemma F.1.** *Let $f : \mathbb{R}^n \to \mathbb{R}^n$ a generic activation function of $n$ nodes. Then the MF recursive equations read:*

$$q_{aa}^{(l+1)} = \sigma_w^2 \int \prod_{i=1}^{n} \mathcal{D}y_i \, f(\mathbf{u})^2 + \sigma_b^2 \, , \tag{57}$$

$$q_{ab}^{(l+1)} = \sigma_w^2 \int \prod_{i=1}^{n} \mathcal{D}y_i \mathcal{D}y_i' \, f(\mathbf{u})f(\mathbf{u}') + \sigma_b^2 \, , \tag{58}$$

*where*

$$\mathbf{u} \equiv \sqrt{q_{aa}^{(l)}}\mathbf{y} \, , \tag{59}$$

$$\mathbf{u}' \equiv \sqrt{q_{aa}^{(l)}}\left( c_{ab}^{(l)}\mathbf{y} + \sqrt{1 - c_{ab}^{(l)}}\mathbf{y}' \right) \, . \tag{60}$$

---

*Proof.* For a generic function multi variable function $f : \mathbb{R}^n \to \mathbb{R}^{n,*}$ we have

$$
\begin{aligned}
\overline{\langle f(\mathbf{y})^2 \rangle} &= \int \prod_{i=1}^{n} d\mu_i \, \mathcal{N}\left(\mu_i; 0, \sigma_\mu^2\right) \int dy_i \, \mathcal{N}\left(y_i; \mu_i, \sigma_y^2\right) f(\mathbf{y})^2 = \\
&= \int \prod_{i=1}^{n} \frac{dy_i}{\sqrt{2\pi\sigma_y^2}} \int \prod_{i=1}^{n} \frac{d\mu_i}{\sqrt{2\pi\sigma_\mu^2}} \, e^{-\frac{\mu_i^2}{2\sigma_\mu^2} - \frac{(y_i - \mu_i)^2}{2\sigma_y^2}} f(\mathbf{y})^2 = \\
&= \int \prod_{i=1}^{n} \frac{dy_i}{\sqrt{2\pi\sigma_y^2}} \, e^{-\frac{y_i^2}{2q}} \int \prod_{i=1}^{n} \frac{d\mu_i}{\sqrt{2\pi\sigma_\mu^2}} \, e^{-\frac{(\mu_i - \sigma_\mu^2 y_i/q)^2}{2\sigma_y^2 \sigma_\mu^2/q}} f(\mathbf{y})^2 = \\
&= \int \prod_{i=1}^{n} \frac{dy_i}{\sqrt{2\pi q}} \, e^{-\frac{y_i^2}{2q}} f(\mathbf{y})^2 = \int \prod_{i=1}^{n} \mathcal{D}y_i \, f(\sqrt{q}\mathbf{y})^2 \, ,
\end{aligned}
\tag{61}
$$

where in the second equality we swapped the integrals over $y_i$ and $\mu_i$, in the third we completed the square exponent, in the fourth we performed the integration over $\mu_i$, and in the last we re-parametrized the Gaussian integrals. Moreover

---

$^*$In this proof, we omit the $l$-dependency for better readability.

$$\overline{\langle f(\mathbf{y})\rangle}^2 = \int \prod_{i=1}^n d\mu_i \, \mathcal{N}\left(\mu_i; 0, \sigma_\mu^2\right) \int \prod_{i=1}^n dy_i \, \mathcal{N}\left(y_i; \mu_i, \sigma_y^2\right) f(\mathbf{y}) \int \prod_{i=1}^n dy_i' \, \mathcal{N}\left(y_i'; \mu_i, \sigma_y^2\right) f(\mathbf{y}') =$$

$$= \int \prod_{i=1}^n \frac{dy_i}{\sqrt{2\pi\sigma_y^2}} \frac{dy_i'}{\sqrt{2\pi\sigma_y^2}} \, f(\mathbf{y}) f(\mathbf{y}') \int \prod_{i=1}^n \frac{d\mu_i}{\sqrt{2\pi\sigma_\mu^2}} \, e^{-\frac{(y_i-\mu_i)^2+(y_i'-\mu_i)^2}{2\sigma_y^2} - \frac{\mu_i^2}{2\sigma_\mu^2}} =$$

$$= \frac{1}{\sqrt{2\gamma+1}^n} \int \prod_{i=1}^n \frac{dy_i}{\sqrt{2\pi\sigma_y^2}} \frac{dy_i'}{\sqrt{2\pi\sigma_y^2}} \, e^{-\frac{y_i^2+y_i'^2}{2\sigma_y^2} + \frac{\gamma^2}{2\sigma_\mu^2(2\gamma+1)}(y_i+y_i')^2} \, f(\mathbf{y}) f(\mathbf{y}') =$$

$$= \frac{1}{\sqrt{2\gamma+1}^n} \int \prod_{i=1}^n \frac{dy_i}{\sqrt{2\pi\sigma_y^2}} \frac{dy_i'}{\sqrt{2\pi\sigma_y^2}} \, e^{-\frac{1}{2q}\left[\left(\frac{y_i(\gamma+1)-\gamma y_i'}{\sqrt{2\gamma+1}}\right)^2 + y_i'^2\right]} \, f(\mathbf{y}) f(\mathbf{y}') =$$

$$= \int \prod_{i=1}^n \frac{d\tilde{y}_i}{\sqrt{2\pi q}} \frac{d\tilde{y}_i'}{\sqrt{2\pi q}} \, e^{-\frac{\tilde{y}_i^2}{2q} - \frac{\tilde{y}_i'^2}{2q}} \, f(\tilde{y}) f(c_{ab}\tilde{y} + \sqrt{1-c_{ab}^2}\tilde{y}') =$$

$$= \int \prod_{i=1}^n \mathcal{D}y_i \, \mathcal{D}y_i' \, f(\sqrt{q}\mathbf{y}) f(\sqrt{q}(c_{ab}\mathbf{y} + \sqrt{1-c_{ab}^2}\mathbf{y'})) \,,$$

$$\tag{62}$$

where in the second equality we applied the definition of Gaussian measure, in the third we completed the square and integrated over $\mu_i$, in the fourth we completed the squares on $y_i, y_i'$ and change the integration variables to

$$\tilde{\mathbf{y}} = \mathbf{y'} \,, \tag{63}$$

$$\tilde{\mathbf{y}}' = \frac{\gamma+1}{\sqrt{2\gamma^{(l)}+1}}\mathbf{y} - \frac{\gamma}{\sqrt{2\gamma+1}}\mathbf{y'} \,. \tag{64}$$

Finally, we renamed the dummy integration variables and appreciate standard Gaussian integrals. The result follows from the definition of $\gamma^{(l)}$, $c_{ab}^{(l)} = \frac{\gamma^{(l)}}{1+\gamma^{(l)}}$ and $\frac{\sqrt{2\gamma^{(l)}+1}}{\gamma^{(l)}+1} = \sqrt{1-(c_{ab}^{(l)})^2}$. We conclude by recalling Lemma D.3. $\qquad\square$

---

**Lemma F.2.** *Let $f : \mathbb{R}^n \to \mathbb{R}^n$ a generic activation function of $n$ nodes. Then:*

$$\chi_1^{(l)} \equiv \left.\frac{\partial c_{ab}^{(l+1)}}{\partial c_{ab}^{(l)}}\right|_{c=1} = \frac{q^{(l)}}{q^{(l+1)}}\sigma_w^2 \int \prod_{i=1}^n \mathcal{D}y_i \, \left\|\nabla f\left(\sqrt{q^{(l)}}\mathbf{y}\right)\right\|^2 \equiv \frac{q^{(l)}}{q^{(l+1)}}\tilde{\chi}_1^{(l)} \,, \tag{65}$$

$$\alpha^{(l)} \equiv \frac{\partial q^{(l+1)}}{\partial q^{(l)}} = \tilde{\chi}_1^{(l)} + \sigma_w^2 \int \prod_{i=1}^n \mathcal{D}y_i \, f\left(\sqrt{q^{(l)}}\mathbf{y}\right)\Delta f\left(\sqrt{q^{(l)}}\mathbf{y}\right) \,, \tag{66}$$

*where $||\cdot||^2$ is the $\mathrm{L}^2$ norm and $\Delta \equiv \sum_{i=1}^n \partial_i^2$ is the Laplacian operator.*

---

*Proof.* The proof is similar of that of Lemma C.1, where for a generic multi-node function $f : \mathbb{R}^n \to \mathbb{R}^n$, Stein's Lemma reads

$$\int \prod_{i=1}^n \mathcal{D}y_i \, f(\mathbf{y}) \, y_i = \int \prod_{i=1}^n \mathcal{D}y_i \, \partial_i f(\mathbf{y}) \,. \tag{67}$$

$\qquad\square$

We now prove some Lemmas regarding a generic single node activation function $\phi()$ followed by 2-dimensional max- and average- pool layers. This analysis will allow us to draw the phase diagram for ReLU and Tanh enriched with these pooling layers.

### F.1 MAXPOOL

> **Lemma F.3.** *Let $f : \mathbb{R}^2 \to \mathbb{R}^2$ a 2-node activation function that can be written as the composition of $\mathrm{MaxPool} : \mathbb{R}^2 \to \mathbb{R}^2$ with a single-node activation function $\phi : \mathbb{R} \to \mathbb{R}$. Then $\phi$ satisfies the following conditions*
>
> *1. All conditions of Proposition 2 of the main paper of Hayou et al. (2019)*
>
> *2. $\phi(x)$ is either odd or even*
>
> *3. $\phi'(x)$ is either odd or even*
>
> *Then $f = \mathrm{MaxPool} \circ \phi$ exhibit the same EOC of $\phi(x)$.*

To prove this Lemma, we need to compute the following operator, which is defined for any multi-node activation function.

> **Definition F.4** (*$V$ operator*). *Let $f : \mathbb{R}^n \to \mathbb{R}^n$. Then we define the following operator $V$, acting on $f$, for any $x \in \mathbb{R}$ as*
>
> $$V[f](x) \equiv \sigma_w^2 \int \prod_i^n \mathcal{D}y_i f\left(\sqrt{x}\mathbf{y}\right) . \tag{68}$$

Note the the $V$ operator can be used to compute the recursive equation of the variance for a multi-node activation function as $q_{aa}^{(l)} = \sigma_w^2 V[f^2]\left(\sqrt{q_{aa}^{(l)}}\right) + \sigma_b^2$.

> **Lemma F.5** (*$V$ operator for 2-d MaxPool*). *Let $\phi() : \mathbb{R} \to \mathbb{R}$ a generic single-node activation function. Then the $V$ operator of $f = \mathrm{MaxPool} \circ \phi$ can be computed as:*
>
> $$V[f](x) = \sigma_w^2 \int \mathcal{D}y \, \phi\left(\sqrt{x}y\right) \Phi(y) , \tag{69}$$
>
> *where $\Phi(x) = \frac{1}{2}\left[1 + \mathrm{erf}(x)\right]$ is the Gaussian cumulative function.*

*Proof.* From Def. F.4, we have

$$V[f](x) = \sigma_w^2 \int \mathcal{D}y_1 \mathcal{D}y_2 \, \mathrm{Max}\left(\phi\left(\sqrt{x}y_1\right), \phi\left(\sqrt{x}y_2\right)\right) =$$

$$= \sigma_w^2 \int \mathcal{D}y_1 \mathcal{D}y_2 \left[\theta_H(y_1 - y_2)\phi\left(\sqrt{x}y_1\right) + \theta_H(y_2 - y_1)\phi\left(\sqrt{x}y_2\right)\right] = \tag{70}$$

$$= 2\sigma_w^2 \int \mathcal{D}y_1 \phi\left(\sqrt{x}y_1\right) \int_{-\infty}^{y_1} \mathcal{D}y_2 = \sigma_w^2 \int \mathcal{D}y \, \phi\left(\sqrt{x}y\right) \Phi(y) .$$

where in the third equations we exploited the symmetry between $y_1$ and $y_2$, and in the fourth we used the definition of Gaussian cumulative function. Finally, we renamed the dummy integration variable. $\qquad\square$

We can now prove Lemma F.3.

*Proof.* From condition 1, we can use algorithm Algorithm 1 of Hayou et al. (2019) to compute the EOC. Then $f(\mathbf{x})$ exhibits the same EOC of $\phi(x)$ if and only if $V[f^2](x) = V[\phi^2](x)$ and $V[f'^2](x) = V[\phi'^2](x), \forall x \in \mathbb{R}$. The $V$ operator is defined in Def. F.4. It is immediate to verify that condition 1 implies $V[f^2](x) = V[\phi^2](x)$ and condition 2 implies $V[f'^2](x) = V[\phi'^2](x)$. $\quad\square$

We now compute the EOC for ReLU and Tanh enriched with MaxPool layers. In particular, Tanh + MaxPool satisfies the hypothesis of Lemma F.3, so it exhibits the same EOC as Tanh. For ReLU, we first prove the following Lemma.

**Lemma F.6.** *Let $\phi = ReLU$ and $f = MaxPool \circ ReLU$, we have*

$$\alpha = \sigma_w^2 \left( \frac{3\pi + 2}{4\pi} \right) . \tag{71}$$

*The signal variance satisfies the following recursion*

$$q_{aa}^{(l+1)} = \alpha q_{aa}^{(l)} + \sigma_b^2 . \tag{72}$$

*Moreover we can compute*

$$\chi_1^{(l)} = \frac{3\sigma_w^2}{4\alpha} \left( 1 - \frac{\sigma_b^2}{q_{aa}^{(l+1)}} \right) \approx 0.82 \left( 1 - \frac{\sigma_b^2}{q_{aa}^{(l+1)}} \right) < 1 , \forall l > 0 . \tag{73}$$

*Therefore across the entire phase diagram we have*

$$\lim_{l \to \infty} c_{ab}^{(l)} = 1 , \tag{74}$$

*and the convergence rate is exponential.*

*Proof.* In this case

$$\tilde{\chi}_1^{(l)} = \sigma_w^2 \int \mathcal{D}y_1 \mathcal{D}y_2 \, \varphi_X(y_1, y_2) = \frac{3}{4}\sigma_w^2 , \tag{75}$$

where $\varphi_X(y_1, y_2)$ is the *characteristic function* of the set $X \equiv X_1 \cup X_2$, where $X_i \equiv \{y_1, y_2 \in \mathbb{R}^2 | y_i \geq 0\}$. Indeed, it is easy to verify that $X$ is the set of points where $\left\| \nabla f \left( \sqrt{q^{(l)}} \mathbf{y} \right) \right\|^2 = 1$. Let us now compute the second term of Eq. (66); for a standard ReLU, this term is zero, since the second derivative of ReLU is (a) non-zero (distribution) only for $y = 0$, where ReLU is zero. Instead, for ReLU+MaxPool we have:

$$\sigma_w^2 \int \mathcal{D}y_1 \mathcal{D}y_2 \, f \left( \sqrt{q^{(l)}} \mathbf{y} \right) \Delta f \left( \sqrt{q^{(l)}} \mathbf{y} \right) =$$

$$= 2\sigma_w^2 \int \mathcal{D}y_1 \mathcal{D}y_2 \, \mathrm{Max} \left( \sqrt{q^{(l)}} y_1, \sqrt{q^{(l)}} y_2 \right) \theta_H(y_1) \theta_H(y_2) \delta_D \left( \sqrt{q^{(l)}} (y_1 - y_2) \right) = \tag{76}$$

$$= 2\sigma_w^2 \int_0^\infty \frac{dy}{2\pi} e^{-y^2} y = \frac{1}{2\pi} \sigma_w^2 ,$$

where $\delta_D$ is the Dirac-delta and $\theta_H$ is the Heaviside-theta function. Therefore

$$\alpha = \sigma_w^2 \left( \frac{3}{4} + \frac{1}{2\pi} \right) = \sigma_w^2 \left( \frac{3\pi + 2}{4\pi} \right) . \tag{77}$$

It is immediate to verify that this is also the value of the $V$ operator, by using Lemma F.5. Therefore the signal variance satisfies Eq. (80), which, when combined with Eq. (65) yields

$$\chi_1^{(l)} = \frac{3\sigma_w^2}{4\alpha} \left( 1 - \frac{\sigma_b^2}{q_{aa}^{(l+1)}} \right) = \frac{3\pi}{3\pi + 2} \left( 1 - \frac{\sigma_b^2}{q_{aa}^{(l+1)}} \right) < 1 , \forall l > 0 , \tag{78}$$

with $\frac{3\pi}{2+3\pi} \approx 0.82$. The convergence rate is thus always exponential (see also the experimental curves in Fig. 23). $\qquad \square$

**Lemma F.7** (Phase diagram of ReLU+MaxPool). *The phase diagram of ReLU+MaxPool is qualitatively similar to that of ReLU. In particular, the EOC collapses to the singleton $\left( \sigma_w^2 = \frac{4\pi}{3\pi + 2} \approx 1.10, \sigma_b^2 = 0 \right)$, while in general gradients vanish for $\sigma_w^2$ below this threshold and explode above it, independently of $\sigma_b^2$.*

*Proof.* Since the correlation coefficient converges exponentially fast across the whole phase diagram, the signal covariance is rapidly equal to the signal variance and the value of $\alpha = \sigma_w^2 \left( \frac{3\pi + 2}{4\pi} \right)$ dictates where gradients explode or vanish (see Fig. **??**). Across this line, the variance converges only for $\sigma_b^2 = 0$ and so the EOC collapses to this point. □

### F.2 AVERAGE POOLING

For Tanh, we can directly use Algorithm 1 of ref Hayou et al. (2019). For ReLU, we have the following Lemma.

---

**Lemma F.8.** *Let $\phi = ReLU$ and $f = AveragePool \circ ReLU$, we have*

$$\alpha = \sigma_w^2 \left( \frac{\pi + 1}{4\pi} \right) . \tag{79}$$

*The signal variance satisfies the following recursion*

$$q_{aa}^{(l+1)} = \alpha q_{aa}^{(l)} + \sigma_b^2 . \tag{80}$$

*Moreover we can compute*

$$\chi_1^{(l)} = \frac{\sigma_w^2}{4\alpha} \left( 1 - \frac{\sigma_b^2}{q_{aa}^{(l+1)}} \right) \approx 0.76 \left( 1 - \frac{\sigma_b^2}{q_{aa}^{(l+1)}} \right) < 1 \, , \forall l > 0 . \tag{81}$$

*Therefore across the entire phase diagram we have*

$$\lim_{l \to \infty} c_{ab}^{(l)} = 1 \, , \tag{82}$$

*and the convergence rate is exponential.*

---

*Proof.* We compute

$$\tilde{\chi}_1^{(l)} = \sigma_w^2 \int \mathcal{D}y_1 \mathcal{D}y_2 \left\| \nabla f \left( \sqrt{q^{(l)}} \mathbf{y} \right) \right\|^2 = \frac{\sigma_w^2}{4} \, , \tag{83}$$

since

$$\left\| \nabla f \left( \sqrt{q^{(l)}} \mathbf{y} \right) \right\|^2 = 1 \tag{84}$$

in the set $X = \{ y_1, y_2 \in \mathbb{R}^2 | y_1 > 0, y_2 > 0 \}$, which has measure $\frac{1}{4}$, and 0 otherwise. Moreover

$$
\begin{aligned}
& \sigma_w^2 \int \mathcal{D}y_1 \mathcal{D}y_2 \, f \left( \sqrt{q^{(l)}} \mathbf{y} \right) \Delta f \left( \sqrt{q^{(l)}} \mathbf{y} \right) = \\
& = \frac{\sigma_w^2}{4} \int \mathcal{D}y_1 \mathcal{D}y_2 \left[ \phi \left( \sqrt{q_{aa}^{(l)}} y_1 \right) + \phi \left( \sqrt{q_{aa}^{(l)}} y_2 \right) \right] \left[ \delta_D \left( \sqrt{q_{aa}^{(l)}} y_1 \right) + \delta_D \left( \sqrt{q_{aa}^{(l)}} y_2 \right) \right] = \\
& = \frac{\sigma_w^2}{2} \int \mathcal{D}y_1 \mathcal{D}y_2 \, \phi \left( \sqrt{q_{aa}^{(l)}} y_1 \right) \delta_D \left( \sqrt{q_{aa}^{(l)}} y_2 \right) = \frac{\sigma_w^2}{4\pi} .
\end{aligned} \tag{85}
$$

Therefore

$$\alpha = \sigma_w^2 \left( \frac{\pi + 1}{4\pi} \right) , \tag{86}$$

which can be directly obtained by using Eq. (57). Similarly to what done for ReLU+MaxPool, we get

$$\chi_1^{(l)} = \frac{\pi}{\pi + 1} \left( 1 - \frac{\sigma_b^2}{q_{aa}^{(l+1)}} \right) < 1 \, , \forall l > 0 , \tag{87}$$

and therefore we also have that the correlation coefficient converges exponentially to one across the entire phase diagram. □

Finally, we compute the phase diagram for this activation function.

---

**Lemma F.9** (Phase diagram of ReLU+AveragePool). *The phase diagram of ReLU+AveragePool is qualitatively similar to that of ReLU. In particular, the EOC collapses to the singleton* $\left( \sigma_w^2 = \frac{4\pi}{\pi+1} \approx 3.03, \sigma_b^2 = 0 \right)$, *while in general gradients vanish for* $\sigma_w^2$ *below this threshold and explode above it, independently of* $\sigma_b^2$.

---

*Proof.* The proof is similar to that of Lemma F.7. □

### F.3  PHASE DIAGRAMS

In Fig. 22 we report the phase diagram of ReLU and Tanh applied before these pooling layers. In general, the presence of pooling layers has the effect of *shifting* the EOC. MaxPool generally shifts the phase diagram toward lower $\sigma_w^2$ values; this is intuitive, since $\sigma_w^2$ globally scales the recursive equations and MaxPool preserves only the larger signal. Tanh with MaxPool show the same phase diagram as without MaxPool; this holds more generally for symmetric single-node activation functions (Lemma F.3). Conversely, for Average Pool the effect is the opposite and the phase diagram is shifted toward larger values of $\sigma_w^2$.

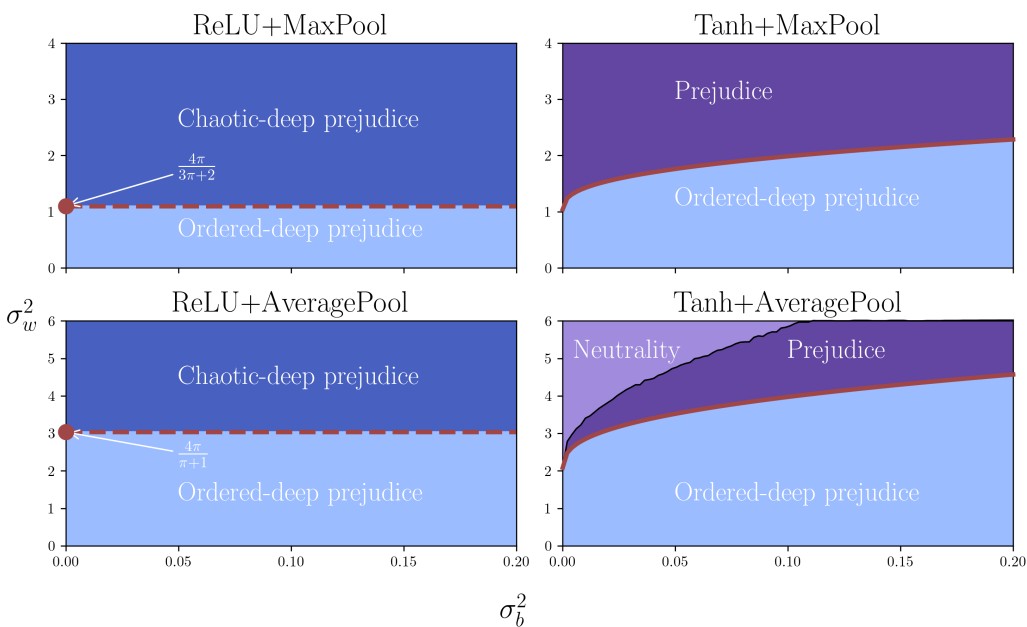

Figure 22: Extensive phase diagrams for ReLU and Tanh enriched with some 2-dimensional pooling layers. These phase diagrams are qualitative equivalent to those without pooling layers, but in general we observe a shift of the EOC and the neutrality/prejudice transition line.

In Fig. 23, we study the behaviour of the correlation coefficient across the phase transition. In particular, we report the convergence with depth for an MLP with ReLU and Tanh activation functions enriched with 2-dimensional average and max pool layers. We can observe the same qualitative behaviour manifested by plain ReLU and Tanh (Fig. 2), but with shifted values of $\sigma_w^2$.

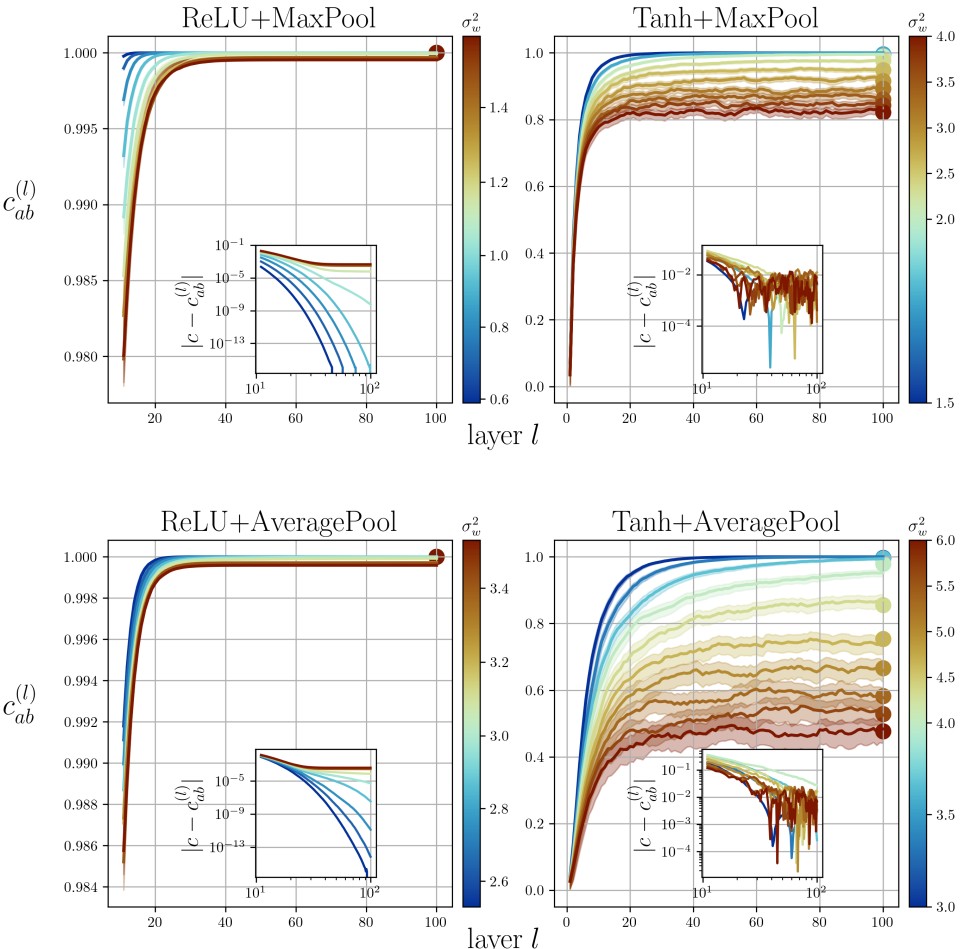

Figure 23: Convergence behaviour the correlation coefficient of ReLU and Tanh with 2-dimensional Max and Average pooling layers for a single MLP with width equal to 10 000 and depth 100. $\sigma_b^2 = 0.1$ and $\sigma_w^2$ varies uniformly from the ordered phase (blue) to the chaotic phase (red). The transition points are $\sigma_w^2 \approx 1.10$ (ReLU+MaxPool), $\sigma_w^2 \approx 3.03$ (ReLU+AveragePool), $\sigma_w^2 \approx 2.00$ (Tanh+MaxPool), and $\sigma_w^2 \approx 3.96$ (Tanh+AveragePool). Scatter points indicate the asymptotic values. The inset plots show the convergence rate for the correlation coefficient to its asymptotic value $c$. Solid lines are computed using the IGB approach, while shaded areas represent the 90 % central confidence interval computed using the MF approach.

# G  GRADIENTS AT INITIALIZATION

Let $\mathcal{L}$ be the generic loss we want to optimize. The gradient compute for a data sample $a$ obeys the following equations:

$$
\begin{aligned}
\frac{\partial \mathcal{L}}{\partial W_{ij}^{(l)}}(a) &= \delta_i^{(l)}(a)\phi\left(y_j^{(l-1)}(a)\right) \ , \\
\delta_i^{(l)}(a) \equiv \frac{\partial \mathcal{L}}{\partial y_i^{(l)}}(a) &= \phi'\left(y_i^{(l)}\right)(a)\sum_{j=1}^{N}\delta_j^{(l+1)}(a)W_{j,i}^{(l+1)} \ .
\end{aligned}
\tag{88}
$$

We define the mean square gradients computed with inputs $a$ and $b$ as:

$$
\tilde{q}_{ab}^{(l)} \equiv \overline{\delta_i^{(l)}(a)\delta_i^{(l)}(b)} \ .
\tag{89}
$$

Practically, $\hat{q}_{ab}^{(l)}$ is computed by propagating two inputs $a$ and $b$ and computing the empirically correlation, over the layer $l$, of $\delta_i^{(l)}(a)$ and $\delta_i^{(l)}(b)$.
By assuming the forward weights to be independent from the backward ones, Hayou et al. (2019) proved that (see their Supplementary Materials)

$$
\tilde{q}_{ab}^{(l)} \approx \tilde{q}_{ab}^{(l+1)}\sigma_w^2 \int \mathcal{D}y\ \mathcal{D}y'\ \phi'(u)\ \phi'(u') \ .
\tag{90}
$$

Therefore, at the critical point $c = 1$, from Eq. (29) gradients satisfy the following recursion

$$
\tilde{q}_{ab}^{(l)} \approx \tilde{q}_{ab}^{(l+1)}\tilde{\chi}_1^{(l)} \ .
\tag{91}
$$

Gradients are thus stable if $\tilde{\chi}_1^{(l)} = 1$.

In the next sections, we report the initialization gradients of the models empirically studied in this work on different datasets. We employ the cross entropy as loss function. For a review of the models and datasets employed we refer to Tab. 2. The value of $\hat{q}_{ab}^{(l)}$ depends on the inputs $a$ and $b$ and becomes a distribution upon imposing a distribution over the input dataset $\mathcal{D}$, though not Gaussian. Thus, in the following we report the median of $\tilde{q}_{ab}^{(l)}$ computed over the dataset. Additionally, we report the same quantity, but computed by restricting the samples to the most favoured and most unfavoured classes.

## G.1  MLP

We report the initialization gradients for a vanilla MLP (Eq. (11)) for both ReLU and Tanh activations on binarized Fashion MNIST (BFMNIST - Fig. 24) and CIFAR10 (Fig. 25). We can observe the gradient vanishing/exploding behaviour very neatly across the phase transition. The same behaviour is present for the most unfavoured class for both datasets and activations. This is expected since the gradient recursion (Eq. (91) does not distinguish between classes. Instead, for the most favoured class of ReLU models, we observe numerically zero gradients of the most favoured class in the chaotic phase. This is due to the fact that the classification loss is numerically zero in the chaotic phase, and so it is in every layer due to Eq. (91). It is easy to explain this by analysing what happens to the cross entropy loss with diverging signals. Indeed, the ReLU chaotic phase is characterized by output signal whose distribution moves further and further from the origin. Therefore, since computing the probability that the output belongs to a certain class involves computing the *exponential* of the output, all the probability mass concentrates to a single class, yielding practically zero classification loss.

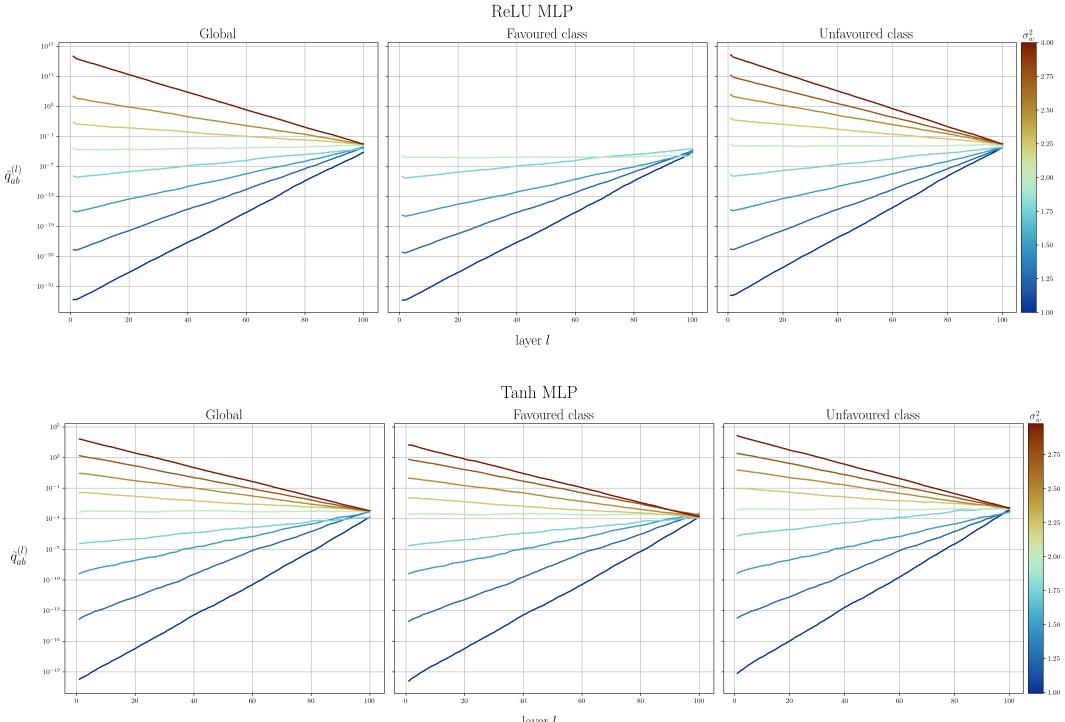

Figure 24: Initialization gradients computed on a batch of binarized Fashion MNIST for a vanilla MLP with ReLU and Tanh activation functions. We set $\sigma_b^2 = 0.1$.

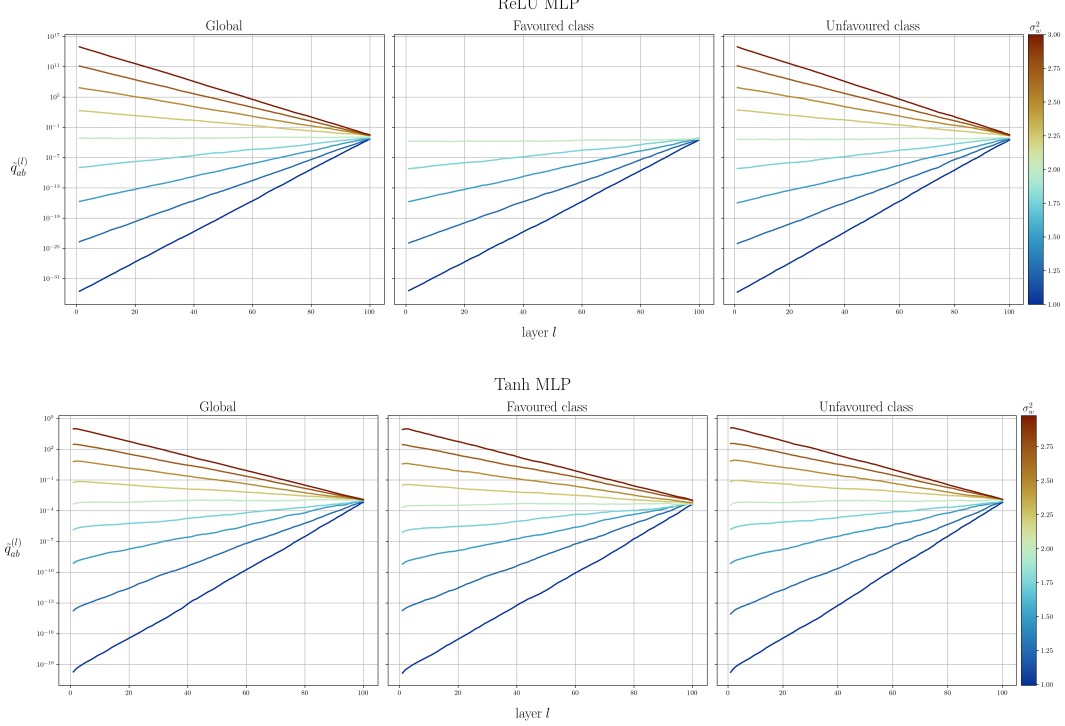

Figure 25: Initialization gradients computed on a batch of CIFAR10 for a vanilla MLP with ReLU and Tanh activation functions. We set $\sigma_b^2 = 0.1$.

## G.2 RESIDUAL MLP

We report the initialization gradients for a Residual MLP (Yang & Schoenholz, 2017) for both ReLU and Tanh activations on binarized CIFAR10 (BCIFAR - Fig. 26) For ReLU (Fig. 26, top), the gradients remain perfectly constant across layers. For Tanh (Fig. 26, bottom), we observe a mild exponential decay—noticeably smaller than in the non-residual MLP (Fig. 25). However, this decay does not impede training nor the model's ability to separate phases (Figs. 10 and 11).

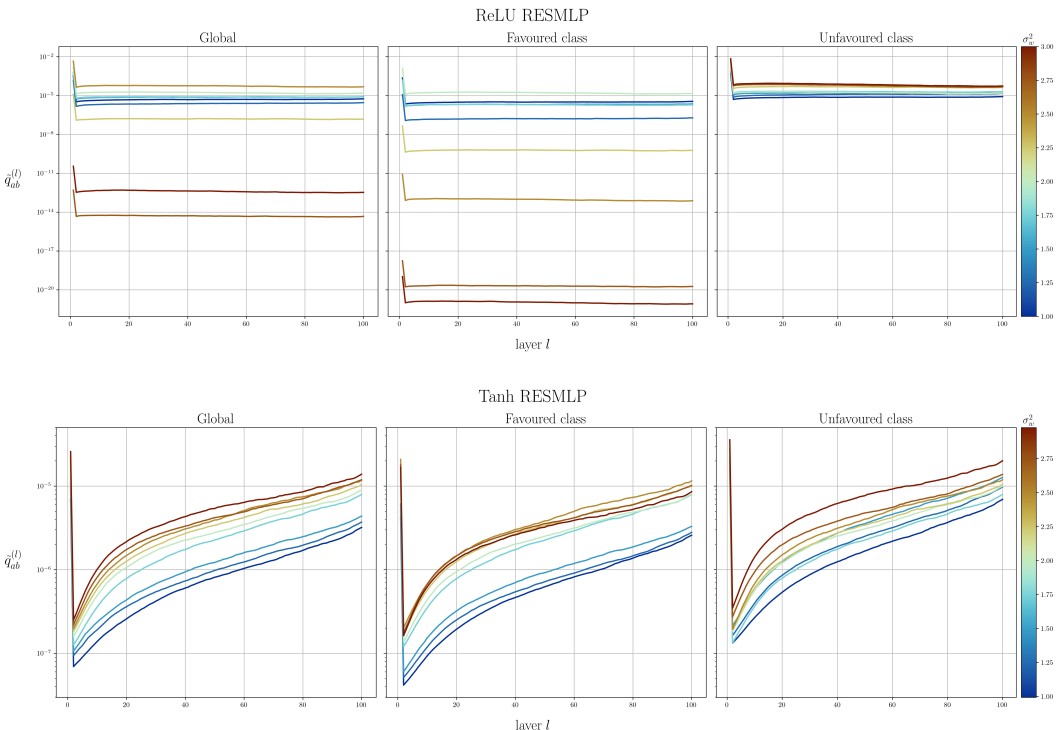

Figure 26: Initialization gradients computed on a batch of binarized CIFAR10 for a vanilla Residual MLP with ReLU and Tanh activation functions. We set $\sigma_b^2 = 0.1$.

## G.3 VANILLA VISION TRANSFORMER

We compute the initialization gradients of a vanilla Vision Transformer (Dosovitskiy et al., 2020) with ReLU activation function on binarized CIFAR10 (BCIFAR, Fig. 27 - top) and CIFAR10 (Fig. 27 - bottom). By removing all batch- and layer-norm layers, we trigger the phase transition observed for instance in the MLP.

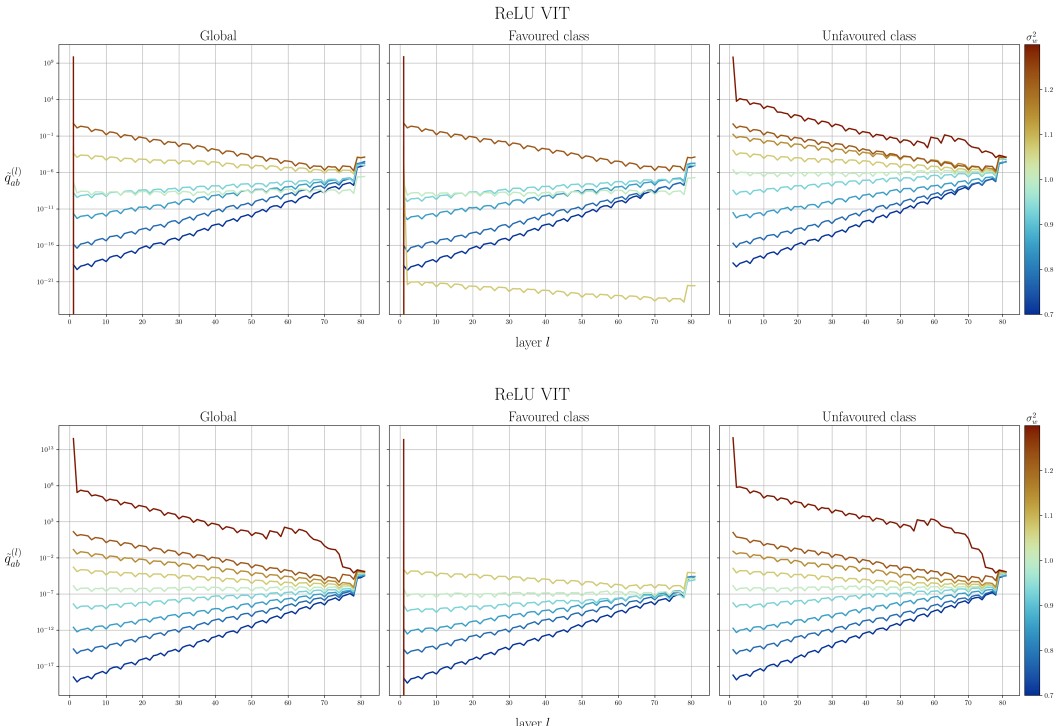

Figure 27: Initialization gradients computed on a batch of binarized CIFAR10 (top) and CIFAR10 (bottom) for a vanilla Vision Transformer (VIT) with ReLU activation function. We set $\sigma_b^2 = 0.1$.

## G.4 LARGE VISION TRANSFORMER

Finally, we compute the gradients on CIFAR100 of a large Vision Tranformer pre-trained on Imagenet. The weights are rescaled as explained in Sec. B.4, while biases are not modified. We observe a similar transition behaviour as in vanilla models (Fig. 27). Due to computational limits, we are able to compute the full gradients only on a batch of size 100 for this model.

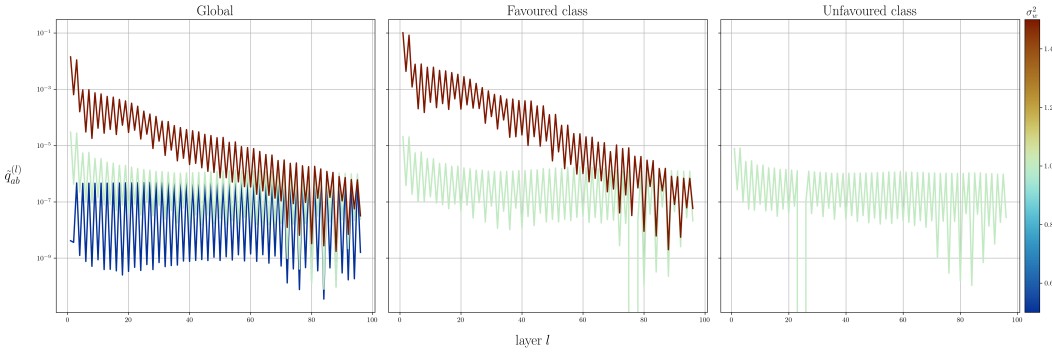

Figure 28: Initialization gradients computed on a batch of CIFAR100 for a large Vision Transformer (VIT) with GeLU activation function. We do not modify the pre-trained biases.

### G.4.1 THE EFFECT OF THE PRE-TRAINING PRIOR ON INITIALIZATION GRADIENTS

One question that might arise naturally is whether the phenomenology analyzed in this paper appears in the same way in pretrained and untrained models. We address this question by comparing the propagation of gradients in a pretrained *vs* an untrained ViT.

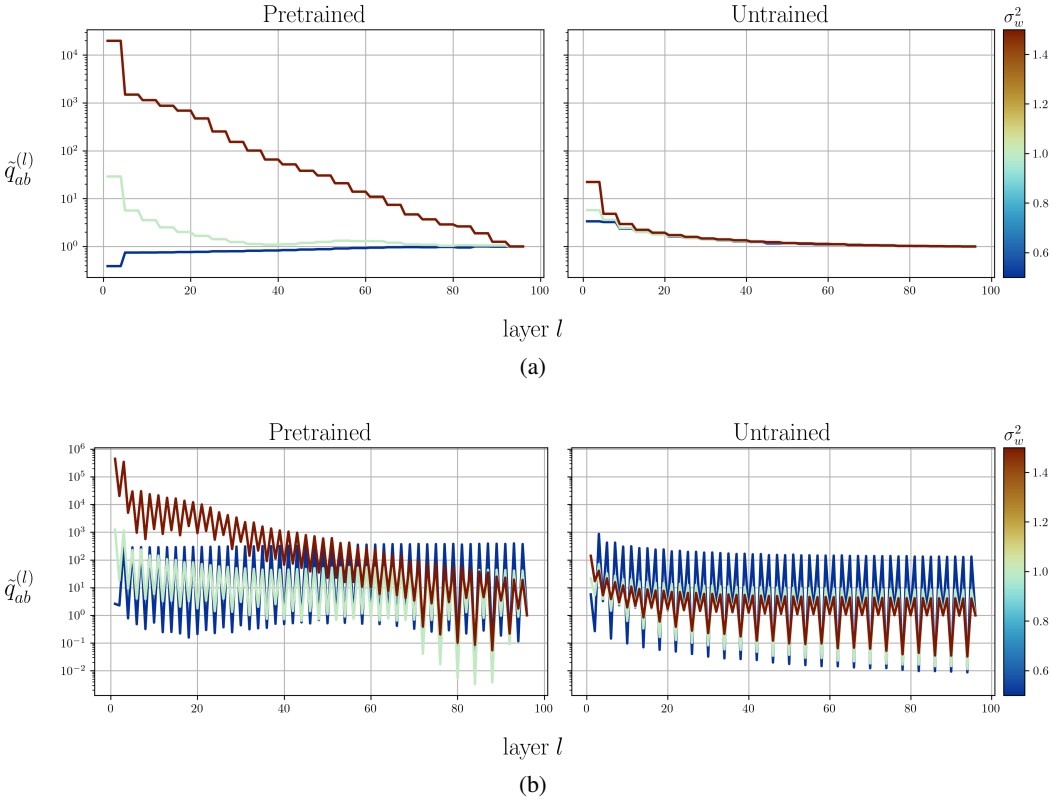

Figure 29: Initialization gradients of the large VIT on CIFAR100, pre-trained and untrained models. In (a) we average the gradients over blocks of four layers to improve clarity, while the original gradients are reported in (b). For easier comparison, curves related to a different value of $\sigma_w^2$ are shifted in such a way to be equal to 1 at the last layer.

We compare the rescaled gradients of the pretrained large ViT with those of same model –this time untrained–, both computed on a batch of CIFAR100 (Fig. 29). Each hidden layer is organized as a block composed of multiple units (or elements). The regular oscillations observed within each layer reveal both the blockwise modular structure and the contribution of individual elements within each block (Fig. 29–b). To more clearly illustrate propagation effects across layers, we additionally report the gradient norms averaged over each layer block (Fig. 29–a). This aggregation suppresses within-layer oscillations and makes the inter-layer propagation behavior more apparent.

As shown in Sec. G.2, we expect that the presence of residual connections suppresses the gradient instability, putting the models closer to an EOC situation. This is indeed what we observe in Fig. 29–b. Surprisingly, however, we do observe gradient exploding in the pretrained netowrk (Fig. 29–a). This suggests that the prior induced by pretrained weights makes gradient stability substantially more fragile than in the untrained case, leading to noticeably stronger gradient explosion under the same range of $\sigma_w^2$.

## H    EXAMPLE OF AN EXPLICIT IGB CALCULATION FOR THE RELU

In this section, we provide an explicit calculation with the IGB approach. In particular, upon computing the explicit recursive relations for $\sigma^2_{y^{(l)}}$ and $\sigma^2_{\mu^{(l)}}$ in case of a ReLU MLP, we are able to prove the following Lemma.

> **Lemma H.1** (Convergence of $c^{(l)}_{ab}$ for ReLU). *The correlation coefficient for the ReLU always converge to one; the convergence rate is exponential for $\sigma^2_b > 0$, $\sigma^2_w < 2$, where $\chi_1 < 1$, and quadratic otherwise ($\chi_1 = 1$).*

*Proof.* Let us consider the ReLU activation function. We want to explicitly compute the objects that appear on the RHS of Eq. (36). The expectation value of $\phi(y)$ over the dataset (*i.e.* the distribution defined in Eq. (33)) is easy to obtain. For better readability, in the next calculations we drop the layer label, since every quantity, if not explicitly declared, refer to layer $l$. We thus have for the linear term:

$$
\langle \phi(y) \rangle = \frac{1}{\sqrt{2\pi\sigma^2_y}} \int_{-\infty}^{\infty} dy \, \max(0, y) e^{-\frac{(y-\mu)^2}{2\sigma^2_y}} = \frac{1}{\sqrt{2\pi\sigma^2_y}} \int_0^{\infty} dy \, y \, e^{-\frac{(y-\mu)^2}{2\sigma^2_y}} =
$$
$$
= \frac{\mu}{2} \left[ \mathrm{erf}\left( \frac{\mu}{\sqrt{2\sigma^2_y}} \right) + 1 \right] + \sqrt{\frac{\sigma^2_y}{2\pi}} e^{-\frac{\mu^2}{2\sigma^2_y}} .
$$
(92)

In the same way we can compute the expectation value $\phi(y)^2$ over data as:

$$
\left\langle \phi(y)^2 \right\rangle = \frac{1}{\sqrt{2\pi V}} \int_0^{\infty} dy \, y^2 e^{-\frac{(y-\mu^2)}{2\sigma^2_y}} = \frac{\mu^2 + \sigma^2_y}{2} \left[ \mathrm{erf}\left( \frac{\mu}{\sqrt{2\sigma^2_y}} \right) + 1 \right] + \mu \sqrt{\frac{\sigma^2_y}{2\pi}} e^{-\frac{\mu^2}{2\sigma^2_y}} .
$$
(93)

We can now compute the expectation values over network ensemble (*i.e.* distributions given by Eq. (34)) for the quadratic term as

$$
\overline{\left\langle \phi(y)^2 \right\rangle} = \overline{\frac{\mu^2 + \sigma^2_y}{2} \left[ \mathrm{erf}\left( \frac{\mu}{\sqrt{2\sigma^2_y}} \right) + 1 \right]} + \overline{\mu \sqrt{\frac{\sigma^2_y}{2\pi}} e^{-\frac{\mu^2}{2\sigma^2_y}}} = \frac{\sigma^2_y + \sigma^2_\mu}{2} = \frac{\sigma^2_y}{2}(\gamma + 1) .
$$
(94)

(95)

For the expectation (over weights and biases) of the square linear term, we get:

$$
\overline{\langle \phi(y) \rangle^2} = \overline{\frac{\mu^2}{4} \left[ \mathrm{erf}\left( \frac{\mu}{\sqrt{2\sigma^2_y}} \right)^2 + 2\,\mathrm{erf}\left( \frac{\mu}{\sqrt{2\sigma^2_y}} \right) + 1 \right]} + \overline{\frac{\sigma^2_y}{2\pi} e^{-\frac{\mu^2}{\sigma^2_y}}} +
$$
$$
+ \overline{\mu \sqrt{\frac{\sigma^2_y}{2\pi}} \left[ \mathrm{erf}\left( \frac{\mu}{\sqrt{2\sigma^2_y}} \right) + 1 \right] e^{-\frac{\mu^2}{2\sigma^2_y}}} = \frac{\sigma^2_\mu}{4} + \frac{1}{4\sqrt{2\pi\sigma^2_\mu}} \int_{-\infty}^{\infty} d\mu \mu^2 \, \mathrm{erf}\left( \frac{\mu}{\sqrt{2\sigma^2_y}} \right)^2 e^{-\frac{\mu^2}{2\sigma^2_\mu}} +
$$
$$
+ \frac{V}{2\pi(\gamma + 1)} \frac{3\gamma + 1}{\sqrt{2\gamma + 1}} = \frac{V}{2} \left( \frac{\gamma}{2} + \frac{1}{\pi(\gamma + 1)} \frac{3\gamma + 1}{\sqrt{2\gamma + 1}} + \frac{I(\gamma)}{\sqrt{\pi\gamma}} \right) .
$$
(96)

where the integral function $I(\gamma) \equiv \int_{-\infty}^{\infty} dx \, x^2 \, \mathrm{erf}(x)^2 \, e^{-x^2/\gamma}$, defined for every $\gamma > 0$, is not trivial. Note that $I(\gamma)$ is smooth in $(0, \infty)$, but it is not defined for $\gamma = 0$. We can thus analytically

prolonged it at $\gamma = 0$ by defining $I(0) \equiv \lim_{\gamma \to 0} I(\gamma) = 0$.

By taking the derivative with respect to $\gamma$, we easily get $I(\gamma) = \gamma^2 \frac{d}{d\gamma} h(\gamma)$, where we introduce the auxiliary integral function $h(\gamma) \equiv \int_{-\infty}^{\infty} dx \ \mathrm{erf}\,(x)^2 \, e^{-x^2/\gamma}$. Moreover, by repeatedly integrating by parts $I(\gamma)$, we have

$$I(\gamma) = -\frac{\gamma}{2} \int_{-\infty}^{\infty} dx \ x \ erf(x)^2 \frac{d}{dx} e^{-x^2/\gamma} = \frac{\gamma}{2} \left( \int_{-\infty}^{\infty} dx \ erf(x)^2 e^{-x^2/\gamma} + \frac{4}{\sqrt{\pi}} \int_{-\infty}^{\infty} dx \ x \ erf(x) e^{-x^2/\gamma - x^2} \right) =$$

$$\overset{\gamma' \equiv \frac{\gamma}{\gamma+1}}{=} \frac{\gamma}{2} h(\gamma) - \frac{\gamma \gamma'}{\sqrt{\pi}} \int_{-\infty}^{\infty} dx \ erf(x) \frac{d}{dx} e^{-x^2/\gamma'} \overset{\gamma'' \equiv \frac{\gamma'}{\gamma'+1}}{=} \frac{\gamma}{2} h(\gamma) + \frac{2\gamma \gamma'}{\pi} \int_{-\infty}^{\infty} e^{-x^2/\gamma''} =$$

$$= \frac{\gamma}{2} h(\gamma) + \frac{2\gamma \gamma'}{\pi} \sqrt{\pi \gamma''} = \frac{\gamma}{2} h(\gamma) + \frac{2\gamma^2}{\sqrt{\pi}} \frac{1}{\gamma+1} \sqrt{\frac{\gamma}{2\gamma+1}} \ . \tag{97}$$

By putting everything together we can write the following differential equation for $h(\gamma)$:

$$\frac{dh(\gamma)}{d\gamma} = \frac{1}{2\gamma} h(\gamma) + \frac{2}{\sqrt{\pi}} \frac{1}{\gamma+1} \sqrt{\frac{\gamma}{2\gamma+1}} \ , \tag{98}$$

whose solution is

$$h(\gamma) = \frac{4\sqrt{\gamma}}{\sqrt{\pi}} \arctan \sqrt{2\gamma+1} - \sqrt{\pi \gamma} \ , \tag{99}$$

where the integration constant has be fixed by analytically computing $h(1) = \frac{\sqrt{\pi}}{3}$. By derivation we thus obtain:

$$\frac{I(\gamma)}{\sqrt{\pi \gamma}} = \frac{2}{\pi} \left( \gamma \arctan \sqrt{2\gamma+1} + \frac{\gamma^2}{(\gamma+1)\sqrt{2\gamma+1}} \right) - \frac{\gamma}{2} \ . \tag{100}$$

By defining

$$g(\gamma) \equiv \frac{\gamma}{2} + \frac{I(\gamma)}{\sqrt{\pi \gamma}} + \frac{1}{\pi \sqrt{2\gamma+1}} \frac{3\gamma+1}{\gamma+1} = \frac{2}{\pi} \gamma \arctan \sqrt{2\gamma+1} + \frac{\sqrt{2\gamma+1}}{\pi} \ , \tag{101}$$

$$f(\gamma) \equiv 1 + \gamma - g(\gamma) \ , \tag{102}$$

we can write also

$$\overline{\langle \phi(y) \rangle^2} = \frac{V}{2} g(\gamma) \ , \tag{103}$$

and

$$\overline{\mathrm{Var}_{\mathcal{D}}(\phi(y))} = \overline{\langle \phi(y)^2 \rangle} - \overline{\langle \phi(y) \rangle^2} = \frac{\sigma^2_{y^{(l)}}}{2} f(\gamma) \ . \tag{104}$$

Therefore we write the recursive relations for $\sigma^2_{y^{(l)}}$ and $\sigma^2_{\mu^{(l)}}$ (by restoring the $l$-dependency):

$$\sigma^2_{y^{(l+1)}} = \frac{\sigma^2_w \sigma^2_{y^{(l)}}}{2} f(\gamma^{(l)}) \tag{105}$$

$$\sigma^2_{\mu^{(l+1)}} = \frac{\sigma^2_w \sigma^2_{\mu^{(l)}}}{2} \frac{g(\gamma^{(l)})}{\gamma^{(l)}} + \sigma^2_b \ . \tag{106}$$

Since $f(\gamma) + g(\gamma) = 1 + \gamma$, by summing together Eq. (105) with Eq. (106) we get a recursion relation for $q^{(l)} = \sigma^2_{y^{(l)}} + \sigma^2_{\mu^{(l)}}$, which has been already discussed by Hayou et al. (2019):

$$q^{(l+1)} = \frac{\sigma^2_w q^{(l)}}{2} + \sigma^2_b \ . \tag{107}$$

In particular, $q^{(l)}$ converges exponentially to zero for $\sigma^2_w < 2$ and diverges exponentially for $\sigma^2_w > 2$, while for $\sigma^2_w = 2$ it is constant when $\sigma^2_b = 0$ and diverges linearly when $\sigma^2_b > 0$. For $\gamma$ can write:

$$\gamma^{(l+1)} = \frac{g(\gamma^{(l)})}{f(\gamma^{(l)})} + \frac{\sigma^2_b}{q^{(l+1)}} \ . \tag{108}$$

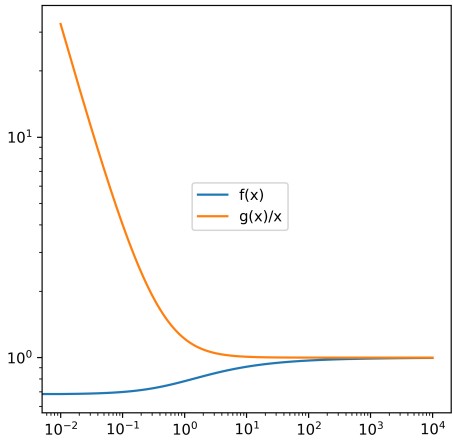

Figure 30: Log-log plot of $g(x)/x$ and $f(x)$ defined in Eq. (101) and Eq. (102), respctively. We observe that they converge to one as $x$ tends toward infinity. Therefore, $g/f$ is asymptotically linear.

The function $(g/f)(\gamma)$ has positive derivative and asymptotically converges to the identity function (see Fig. 30). Therefore, $\gamma$ always diverges with the depth. That is, $c_{ab}^{(l)}$ always converges to one for ReLU, contrary to what claimed by e.g. Schoenholz et al. (2016) and Hayou et al. (2019).

Let us start analyzing the case $\sigma_b^2 = 0$. Interestingly, in such case Eq. (108) does not depend on $\sigma_w^2$. The convergence rate depends on the sub-leading term and to find it we can expand $f$ and $g$ for large $\gamma$. We get

$$g(\gamma) = \gamma + \frac{2\sqrt{2}}{3\pi\sqrt{\gamma}} + O(\gamma^{-3/2}) \,, \tag{109}$$

$$f(\gamma) = 1 - \frac{2\sqrt{2}}{3\pi\sqrt{\gamma}} + O(\gamma^{-3/2}) \,, \tag{110}$$

and thus

$$g/f)(\gamma) = \gamma + \frac{2\sqrt{2}}{3\pi}\sqrt{\gamma} + O(\gamma^{-1}) \,. \tag{111}$$

Therefore $\gamma$ always diverges quadratically when $\sigma_b^2 = 0$. When $\sigma_b^2 > 0$ we have to distinguish the case base on the value $\sigma_w^2$. For $\sigma_w^2 < 2$, $q^{(l)}$ converges exponentially fast to zero, therefore $\gamma^{(l)}$ diverges exponentially. For $\sigma_w^2 > 2$, $q^{(l)}$ diverges, and so the the divergence rate of $\gamma^{(l)}$ is quadratic as in the $\sigma_b^2 = 0$ case. A similar discussion applies for $\sigma_w^2 = 2$.

To summarize, we obtained:

$$\lim_{l\to\infty} \sigma_{y^{(l)}}^2 = \begin{cases} 0 & if\ \sigma_w^2 \le 2, \forall \sigma_b^2 \\ +\infty & if\ \sigma_w^2 > 2, \forall \sigma_b^2 \end{cases} \,, \tag{112}$$

$$\lim_{l\to\infty} \sigma_{\mu^{(l)}}^2 = \begin{cases} +\infty & if\ \sigma_w^2 = 2, \sigma_b^2 > 0 \\ +\infty & if\ \sigma_w^2 > 2, \forall \sigma_b^2 \\ finite & else \end{cases} \,, \tag{113}$$

$$\lim_{l\to\infty} \gamma^{(l)} = +\infty \begin{cases} exponentially & if\ \sigma_w^2 < 2\,, \sigma_b^2 > 0 \\ quadratically & else \end{cases} \,. \tag{114}$$

$\square$

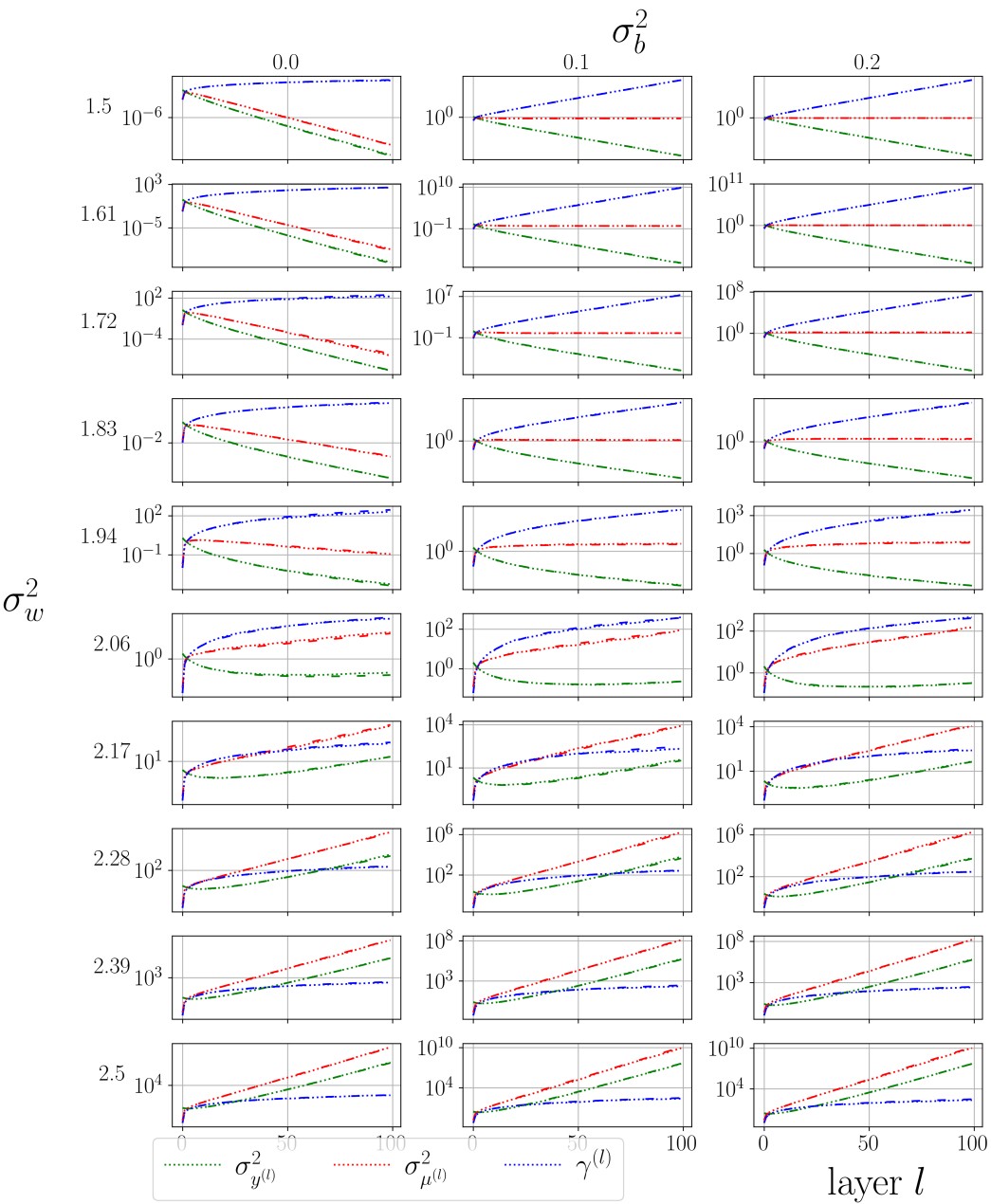

Figure 31: Theoretical (dashed) and experimental (dots) lines obtained for ReLU with different values of $\sigma_w^2$ close to the critical point $\sigma_w^2 = 2.0$. The width of the network is 10000. The initial theoretical values are adjusted to take into account finite datasets effects. We observe good agreement between the theory and the experiments.

