# OpenReview forum: "When Bias Meets Trainability: Connecting Theories of Initialization"
_ICLR.cc/2026/Conference — ICLR 2026 Poster_

### Official Review · Reviewer_YDAS · 2025-10-31

**Soundness:** 2
**Presentation:** 2
**Contribution:** 2
**Rating:** 2
**Confidence:** 3

**Summary:**

Develops a theoretical link between mean-field (MF) trainability and initial guessing bias (IGB) at random initialization. Shows an equivalence between MF quantities and IGB statistics, argues that the edge of chaos corresponds to a state of transient deep prejudice that enables fast learning, and provides supporting experiments on MLPs, a simplified ViT, and a large ImageNet-pretrained ViT fine-tuned on CIFAR-100.

**Strengths:**

- Clear statement of the MF-IGB connection, including a formal mapping between MF covariances and IGB drift ratio, and phase-diagram reinterpretations for bounded and unbounded activations.
- Reproducibility note and code availability claim.
- Empirical plus theoretical curves illustrating $c^{(l)}\rightarrow$1 for ReLU, with different rates in ordered vs. chaotic regimes.

**Weaknesses:**

- Empirical scope is limited relative to broad claims. Experiments focus on binarized Fashion-MNIST, binarized CIFAR-10, CIFAR-10 with simplified MLP/ViT, and a single large ImageNet-pretrained ViT on CIFAR-100.

- Assumptions narrow external validity. Core analysis relies on the infinite-width mean-field regime and an i.i.d. Gaussian input model in the IGB framework.

- Fairness framing without fairness evaluation. The introduction and contributions connect learnability and fairness from initialization onward, but the experiments report only accuracy and maximum class fractions, with no group-attribute metrics.

- Pretraining-bias angle is underdeveloped. For the large ViT, weights are uniformly rescaled to traverse phases, but there is no analysis disentangling ImageNet pretraining priors from weight-scaling dynamics.

- Clarity/calibration: extrapolating MF to practice. Theoretical mapping claims that “best trainability” corresponds to a state of transient deep prejudice (Proposition 4.1), yet validation remains on toy or simplified settings.

**Questions:**

- Representativeness of experiments. Can the empirical section include standard ImageNet-scale or at least non-binarized, multi-class tasks with production architectures (with normalization and residuals) to assess whether the MF–IGB link and the “transient deep prejudice” claim persist beyond toy settings?

- Input-distribution assumptions. How sensitive are the theoretical conclusions to deviating from i.i.d. Gaussian inputs, for example, to real image statistics or correlated features? Any finite-width theory or experiments targeted at non-Gaussian inputs?

- Fairness measurements. Since the paper motivates fairness, could group metrics on a dataset with sensitive attributes be provided to test whether initial “transient deep prejudice” impacts disparities during early training and at convergence?

- Pretraining biases. For the large ViT, is it possible to disentangle the role of ImageNet pretraining from the weight-scaling operation, and measure whether class-prior skew or spurious correlations from pretraining amplify or dampen IGB on the target data?

**Details Of Ethics Concerns:**

The paper states that they are finding related work with LLMs, but is that allowed?
Also, the entire paper appears to be LLM-written. Now I am not certain if they simply rephrased every human-written paragraph using LLMs or made LLMs generate the entire paper.

---

> ### Author Response · Authors · 2025-11-20
> **Answer to YDAS review**
>
> We thank the reviewer for these thoughtful comments. Below we address each point in turn.
> ## 1. Empirical Scope and Representativeness
> We agree that evaluating the MF–IGB connection on realistic architectures strengthens our claims. In the main paper, Thm. 3.1 was first verified on Gaussian-generated data to remain fully aligned with the IGB theoretical assumptions. A central contribution of our work, however, is to show that the key IGB quantities can be expressed in terms of MF correlations, objects that can be computed directly on real data and real architectures. This connection enables us to predict the level of initialization bias without relying on Gaussian inputs.
> As part of this rebuttal, we have computed the MF–IGB curves for standard production architectures such as ResNets and MLPMixers (see Fig. 20 and Fig. 21 in Appendix E.1 of the updated version of the paper), using real structured datasets (analogous to Fig. 2). These additional results show that the MF == IGB equivalence persists in practical, non-toy settings.
> More broadly, all training experiments in the paper are performed on standard benchmark datasets, and quantities such as gradient scaling (Appendix G) are already measured across multiple datasets.
>
>
> ## 2. Infinite-Width and Gaussian Assumptions
> In the IGB framework, i.i.d. Gaussian inputs are used solely to obtain closed-form expressions that isolate how architectural and initialization choices generate bias. The MF framework, on the other hand, does not assume Gaussian data. By establishing an equivalence between IGB and MF predictions in the large-width regime, our work shows that the key IGB quantities can be computed through MF tools even when the data are structured and non-Gaussian. We will clarify this implication more explicitly in the revision.
> The phenomenon we study (initial bias tightly linked to trainability) is not driven by infinite-width artifacts: it appears robustly in finite-width networks, as confirmed by our experiments (see also “On the Infinite-Width Assumption” in this rebuttal). While a comprehensive finite-width theory for output distributions does not yet exist, current approximations are promising; developing such a theory lies beyond the scope of this work, which builds on well-established infinite-width frameworks.
> ## 3. Fairness Framing
> We appreciate the reviewer’s remark. Our work focuses on initialization-induced predictive asymmetries (i.e., skew in the output distribution), not fairness across protected attributes. Since we do not compute group-wise metrics, referencing fairness can be misleading. In the revised manuscript, we will adjust the introduction and terminology to avoid fairness claims and emphasize that our results concern statistical asymmetries at initialization.
> ## 4. Pretraining Bias and Weight Scaling
> We thank the reviewer for the thoughtful observation. Our results are meant to provide insights on a setting that is not included in our theory. While (Francazi et al 2024) showed that in pre-trained models IGB effects are qualitatively similar to the untrained case, here we expanded these observations showing that consistently the similarity extends on gradient stability measures. However we agree that the effects introduced by the pre-training prior on the signal propagation could be not trivial. To delve in this direction and disentangle the IGB effects from correlations coming from the prior, one could repeat the experiments at initialization on the same model with weights randomly initialized. Comparing the slope of the gradient norm in the two cases (keeping the scaling factor with respect to the stable value common in the two cases) would isolate the effects induced by the pre-training prior. Would the reviewer find this a satisfactory test?
> We will incorporate a clearer discussion of these aspects, and add corresponding analyses where feasible, in the final revised version.
>
> ## MF - IGB mapping in practice
> We thank the reviewer for this point. In MF theory, the Edge of Chaos is precisely the regime where gradients remain stable across depth (Poole et al., 2016), and is therefore the region of best trainability for deep architectures. The link to initial prejudice follows directly from Eq. 6: at the edge of chaos the fixed point \(c = 1\) is stable, and Theorem 3.1 shows that this condition corresponds to a regime of deep prejudice (diverging \( \gamma \)). This theoretical connection is derived under mild assumptions (see “Linking MF Predictions to Initialization Bias” in this rebuttal).
> To strengthen the empirical support for this mapping beyond simplified settings, we include in this rebuttal additional experiments on ResNets and MLPMixers (see Fig. 20 and Fig. 21 in Appendix E.1 of the updated version of the paper). These results show the compatibility predicted by the theory, further supporting the practical relevance and applicability of our result.

---

> > ### Author Response · Authors · 2025-11-20
> > **Comment on “Details Of Ethics Concerns”**
> >
> > During submission it is required to indicate the use of LLM and it is possible to use them for certain tasks, among which finding related work. We have not used LLMs to write this paper.

---

> > ### Comment · Reviewer_YDAS · 2025-11-26
> >
> > > "Comparing the slope of the gradient norm in the two cases (keeping the scaling factor with respect to the stable value common in the two cases) would isolate the effects induced by the pre-training prior. Would the reviewer find this a satisfactory test?"
> >
> > > Yes.
> >
> >
> > Most of my concerns have been addressed; the last remaining is the one stated above.
> > I have changed my score to better reflect the current evaluation, without the aforementioned experiment.
> >
> > Best Regards

---

> ### Author Response · Authors · 2025-12-03
> **New results addressing the remaining open point**
>
> We thank the reviewer for the positive response to our reply.
>
> We are glad that the reviewer finds our proposed experiment clarifying. We included the comparison between the untrained and pre-trained model in the new version to address their last doubt. We show the new runs in App. G.4 (Figs. 29).
> Our experiments show reviewer YDAS was right; pretraining does induce a prior which increases the gradient exploding. We added a discussion on this non-trivial observation in App. G.4. We now note that the untrained models are less sensitive to variance rescaling than the pretrained models.
>
> Since we now addressed the remaining point requested by reviewer YDAS, we hope that our paper will benefit a further increase in evaluation.

---

### Official Review · Reviewer_h1rA · 2025-11-01

**Soundness:** 3
**Presentation:** 2
**Contribution:** 3
**Rating:** 4
**Confidence:** 2

**Summary:**

This paper investigates the relationship between initialization bias and trainability of DNNs, reconciling two theoretical perspectives:
- Mean-Field (MF) theory — which studies gradient propagation and identifies the “edge of chaos” (EOC) as the optimal initialization boundary for stable training;
- Initial Guessing Bias (IGB) framework — which describes how untrained networks can exhibit predictive bias (favoring one class) even before training.

The authors establish a formal equivalence between the MF and IGB frameworks, showing that the trainability boundary in MF corresponds to a biased initialization state in IGB. Contrary to the intuitive belief that the most trainable initializations should be neutral, they demonstrate theoretically and empirically that the optimal initialization is systematically biased rather than unbiased — a state they call transient deep prejudice.

**Strengths:**

1. Theoretical contribution: The paper establishes a clean mathematical connection between two previously distinct frameworks (MF and IGB), enriching both perspectives.
2. Insight: It introduces a novel idea: bias at initialization can improve trainability; The new “prejudice-neutrality” phase view offers an intuitive explanation for initialization effects, linking bias to the dynamical stability of gradient flow.

**Weaknesses:**

1. The theory is derived in the infinite-width limit and validated on small- to mid-scale settings. Its applicability to practical, large-scale deep networks (e.g., transformers with normalization and attention) is not demonstrated. Additionally, empirical evaluation focuses on synthetic and small vision datasets. It is of readers' interest to learn results on simple language tasks.
2. Ambiguous practical relevance: While the “transient bias” insight is conceptually interesting, there is no clear recipe for practitioners (e.g., how to initialize weights to achieve the right level of prejudice). It would be helpful to add some executable takeaways.
3. It would be helpful to introduce and compare with alternative trainability-enhancing initializations (e.g., orthogonal, LSUV, scaled ReLU, or NTK-based initializations).

**Questions:**

There can be some terminological confusion: It would be helpful to clarify that “bias,” “prejudice,” and “neutrality” refer to statistical asymmetry / symmetry in prediction space. Otherwise, readers may think about fairness or ethical bias.

---

> ### Author Response · Authors · 2025-11-20
> **Answer to h1rA review**
>
> We thank the reviewer for these constructive observations. Below we address each point raised.
>
> ## 1. Applicability Beyond Small/Mid-Scale Models
> Our theoretical results hold for any architecture obeying Eq. 1, which includes networks composed of stacked modules such as normalization layers, attention blocks, and MLP sublayers. This allows us to investigate complex architectures, including Transformers, by propagating correlations and gradient statistics across these components.
> To empirically support this, we include experiments on a pretrained Vision Transformer (Fig. 4, bottom), which already contains attention, normalization, and residual connections. While training a 300M-parameter model from scratch is computationally infeasible for us, the fine-tuning setting is directly relevant: it demonstrates that the predicted initialization bias persists in large-scale pretrained models and that it interacts with their trainability. We also prepared new experiments on ResNet and MLP mixer, showing that the equivalence between IGB and MF still holds (see Fig. 20 and Fig. 21 in Appendix E.1 of the updated version of the paper). Does the reviewer find these satisfactory to improve the grading of our paper, or would they require more (possibly being mindful of computing expenses and time boundedness of the rebuttal period)?
> Regarding language modeling, IGB theory is intrinsically tied to classification and does not transfer to autoregressive likelihood models in a straightforward manner. For example, how IGB is defined in a model which generates a token according to a probability instead of choosing the argmax is currently an open question. A principled extension would require a dedicated future study.
>
> ## 2. Practical Relevance and Takeaways
> Our analysis yields a concrete guideline: **neutral initializations are not necessarily optimal; transient prejudice can be beneficial when gradients are stable across depth**. In practice, when the Edge-of-Chaos point is unknown (as in most real architectures) one can probe different initialization scales and select those that stabilize gradients across layers (as illustrated in App. G). We will add these practical recommendations to the revised manuscript.
>
> ## 3. Comparison to Alternative Initialization Schemes
> Comparing a broad range of initialization schemes is an interesting direction, but beyond the scope of this work. Our goal is to formally link two theoretical frameworks—MF theory and IGB theory. MF theory already accommodates many initialization parameterizations (standard MF, NTK, maximum update, and the general ABC parametrization), whereas current IGB theory has only been developed for the standard MF parametrization. Our main result (Thm. 3.1), however, is stated at a level of generality that suggests applicability beyond this case; verifying this empirically is a promising avenue for future work.
>
> ## Question: Clarification of Terminology
> We appreciate the opportunity to clarify this point. In our work, the terms **prejudice** refer strictly to statistical asymmetries in the model’s *initial prediction distribution*, not to ethical or societal bias. We will revise the terminology section accordingly and remove fairness-related remarks to avoid any ambiguity (see “Terminology: Prejudice, Bias, and Neutrality” in this rebuttal).

---

### Official Review · Reviewer_cZgs · 2025-11-01

**Soundness:** 3
**Presentation:** 3
**Contribution:** 4
**Rating:** 8
**Confidence:** 2

**Summary:**

The paper links two views of randomly initialized networks: initial guessing bias, which is when untrained models over-predict one class, with the mean field theory describing trainability at initialisation. The core result is that there is a mapping between both views, such that the same initialisation that can yield a good gradient flow also produces the most biases starting predictions. The paper extends this across architectural choices, including an experiment to perturb a pre-trained ViT to demonstrate these effects.

**Strengths:**

1. Clean conceptual claim that ties IGB statistics with network trainability
2. Clear testable claim forhow models which start with maximal bias learn fastest
3. Robust checks beyond toy MLPs to ViTs, both training and perturbing them

**Weaknesses:**

1. The infinite width setting is somewhat limiting on the theory side
2. It's unclear how much the choice of norms matters or whether this result is idiosyncratic
3. The pooling for the 2-dimensional case is similarly limited

**Questions:**

1. Is it possible to link the theory to regimes beyond the infinite width setup?
2. How does this theory work under harder settings e.g. the vanilla MLPs are not the strongest baselines for performance?
3. Is it possible to measure bias level in practice and across training?

---

> ### Author Response · Authors · 2025-11-20
> **Answer to cZgs review**
>
> We are glad for reviewer cZgs’s positive assessment and are glad to address the points they raise.
>
> **W1**: The IGB and MF theories are two established theories which are derived in the infinite-width limit. Since these theories are only defined in this limit, it is there that the connection needs to be drawn. To avoid further repetitions, we point reviewer cZgs to the common post on the top (“Relevance of Mean-field results”), explicitly devoted to explain the relevance of MF.
>
> **W2**: Recent work (https://arxiv.org/abs/2505.11312) addresses the impact of layer and batch normalizations on initialization biases. They show that this is dependent on which norm is placed where. In the specific case of putting batchnorm after the activation, prejudiced initializations become neutral. In most other cases, prejudices persist even with the norm. This is consistent with the mean field theory of batch norm (https://arxiv.org/pdf/1902.08129), which observes gradient exploding regardless of the weights (consistently with a neutral phase). However, as noted in (https://arxiv.org/abs/2505.11312), other normalization schemes give rise to prejudiced phases and the phase transitions reported in our paper. This is also confirmed by our experiments, for example the phase transition is also visible in networks with normalizations (e.g. ViT has layernorm).
>
> **W3**: While we show 1-dimensional pooling, this can be directly extended to 2-dimensional pooling and we do not expect any complication nor qualitative difference.
>
> **Q1**: The infinite-width limit allows us to have a theory which provides exact analytical results for a very wide family of models. In essence, infinite width allows for a theory which is very ample in terms of architecture. Finite-width extensions may be possible, but there are very few theoretical works which can actually provide exact results in the finite-width limit. This means that it would require a huge effort (provided that it is even possible), which goes beyond the scope of our already very long work, and likely one would need to make hypotheses which limit the validity of the theory in other manners.
> On another side, we can address this through numerical experiments, which confirm our results in infinite width.
> For further comments on this, see the common post on the top, on the relevance of MF.
>
> **Q2**: We assume that the question is whether one finds our phenomenology also in realistic networks. We do, see e.g. the Vision Transformer on Fig.13 from section B4, which reaches almost 100% accuracy on cifar100.
> If instead the question is about the theory, although our theory does not cover all architectural elements, MLP modules are present in most. Since these modules are present in every layer, the results on gradient exploding/vanishing/edge of chaos are still valid, but with a more complex pattern (see e.g. Fig.24).
>
> **Q3**: Yes, it is possible to measure the bias level during training. The way we chose to do it is by monitoring how many guesses are assigned to the favored class. This quantity is the Max Freq. shown in Fig.4 and similar ones.

---

### Official Review · Reviewer_2DTJ · 2025-11-03

**Soundness:** 3
**Presentation:** 2
**Contribution:** 2
**Rating:** 4
**Confidence:** 2

**Summary:**

The paper studies initial guessing bias (IGB)—when a network exhibits a bias towards certain classes. It mathematically links IGB to mean-field theory of wide networks, thus connecting learnability properties to IGB. Using this, the authors show that initializing networks to be optimized for trainability and have stable gradients also yields class bias at initialization.

**Strengths:**

Understanding how and why neural networks are biased is an important topic for fairness in AI systems. This paper considers the impact of weight initialization, independent of data, which could help us understand how learning dynamics affect bias. By connecting IGB to established mean-field theory of wide networks, it helps build a framework for this. The authors support their theoretical claims with experiments in several architectures and datasets.

**Weaknesses:**

I found the paper difficult to follow and understand. I think it could have more strongly motivated why the link to MF theory is an important one to make and why the contribution made is important. It was also unclear to me at times what was background on previous work versus a novel contribution of the paper.

“Prejudice” is a more loaded term than “bias” and I find the use of it here somewhat inappropriate for what is being referred to. Bias is used to refer to behavior a network might be more prone to do (e.g., inductive bias) and I think would be more apt.

The paper briefly mentions it, but by assuming the infinite-width limit, the paper focuses on the so-called lazy learning regime. While the lazy learning regime may be valuable to study, it seems like a limitation for a study motivated by fairness to not focus on the rich feature learning regime optimized for in practice. Furthermore, there is substantial literature on feature learning that discusses how different initializations impact learning and representations, which should probably be reviewed and discussed more in this paper.

I think the contribution of the paper is limited. In particular, the main finding concerns how different initializations associated with trainability also reflect different bias. However, although they show that stable initializations are initially biased, this effect goes away with training. It’s unclear whether this specific initialization effect would persist in other settings and actually affect fairness. For example, this result would be strengthened by showing that it persists or is amplified by certain properties of the data.

**Questions:**

What is the difference between “chaotic-deep prejudice” and “(chaotic) prejudice”? Why is it called “deep”?

---

> ### Author Response · Authors · 2025-11-20
> **Answer to 2DTJ review**
>
> ## Presentation of context and contributions
> We thank the reviewer for this helpful feedback. We agree that clarifying the connection to MF theory and more explicitly delineating our contributions will further strengthen the manuscript. In the final revised version, we will incorporate these improvements and add a short, self-contained summary of the MF framework to better contextualize our results (see also the paragraph “The importance of MF theories” and "Linking MF Predictions to Initialization Bias " in this rebuttal).
> ## Bias and Prejudice
> We believe that prejudice is a better term than bias when it comes to untrained models. Prejudice is the bias that the model has before even seeing the data, i.e. an initial guessing bias (see “Terminology: Bias, Prejudice and Neutrality” in this rebuttal). We will make this distinction more clear in the final version of the paper.
>
> ## Infinite width limit
>  While our theoretical analysis relies on the infinite-width limit, our experiments are carried out on finite, practically relevant models. In these settings, we consistently observe the same phase structure and bias/trainability behaviours predicted by the theory. Thus, although a full theoretical treatment of the feature-learning regime is an important direction, our results show that the MF-based insights already transfer meaningfully to real architectures and training setups. We will highlight more this insightful value of the empirical validation and add a discussion about the feature learning regime.
>
> ## Impact of the results
> We thank the reviewer for this comment. A central point to highlight is that while models at the edge of chaos can rapidly absorb their initial bias, this is not the case outside this regime. In the ordered phase, the initial bias is coupled with vanishing gradients and therefore remains persistent throughout training (see Fig.4 or App.B). This has concrete implications beyond final performance. For example, during hyper-parameter tuning, models evaluated through short runs may perform well only on specific subgroups, leading to biased model selection driven purely by improper initialization. We agree that examining how these effects interact with data-induced factors is a valuable direction. Although this lies beyond the scope of our work, our results already provide insights that can support future studies, for instance, by revealing systematic differences in per-class gradient behaviour across phases (see Fig.21 and Fig.22, where we show that gradient exploding can be driven only by a single class). Similar effects can also arise from data properties such as class imbalance, where these per-class gradient discrepancies impact the learning dynamics leading to bias emergence and affecting the convergence speed (Francazi et al 2023). Our results suggest that initialization choices and data composition may generate related gradient-mediated mechanisms. We will clarify these points more explicitly in the final revised version.
>
> ## Questions
>
> In our terminology, **chaotic** refers to the Mean-Field notion of chaos, i.e., a regime in which gradients explode across layers. **Prejudice** denotes a skew in the initial predictions toward one class (see “Terminology: Prejudice, Bias, and Neutrality” in this rebuttal). This skew can vary in magnitude: from mild imbalance among class guesses to the extreme case in which *all* datapoints are assigned to the same class, corresponding to $\gamma \to \infty$ in the IGB framework.
> Francazi et al. (2024) showed that in some settings this skew can *amplify with depth*, driving $\gamma$ toward infinity as the network becomes deeper. We refer to this amplified regime as **deep prejudice**—“deep” indicating that the prejudice intensifies with depth. In contrast, **prejudice** alone refers to the milder case where $0 < \gamma < \infty$, even for architectures of arbitrary depth.
> We appreciate the opportunity to clarify this terminology, and we will use the reviewer’s feedback to present the distinction more cleanly in the revised manuscript.

---

### Author Response · Authors · 2025-11-20
**General comment - relevance of mean-field**

# Relevance of Mean-field results
Our work takes a very influential line of research on initialization (which e.g. enabled easier training of very deep networks), and connects it to a very recent one on initialization biases (from ICML2024). Both theories are obtained in the mean-field limit, so connecting them only makes sense within the mean-field limit, as these theories are not defined outside of it. Furthermore, our results are supported by a wide array of runs in finite-width networks (we now have additional experiments on ResNet and MLP-Mixer (see Fig. 20 and Fig. 21 in Appendix E.1 of the updated version of the paper) and we can provide more if reasonable requests are made soon enough). This reflects the fact that modern architectures are highly overparameterized and practically very wide (at least, wide enough for the phenomenology of our theory).
Additionally, our results provide further practical and theoretical insight:

- Connecting MF to IGB means that we are connecting initialization gradients to biases. Once this connection is made, we are able to unveil and explain non-trivial effects happening in practice. A notable example, which had failed to highlight, is presented in App. G. When gradient exploding occurs, it is not the gradients related to all classes which explode, but only the unfavored ones. E.g. in a binary problem, if the model is initialized in a prejudiced phase, one class’s gradients explode, and the other class’s vanish.

- When hyperparameter (HP) tuning, one performs many short runs and chooses the HPs based on the early time performance of the model. A result of our theory is that, depending on which phase one is, the early time performance could be only representative of a subset of the data, thus harming the quality of hyperparameter tuning.

Since we acknowledge that these points were not clear in the previous version, in the final version of the manuscript, we:

- Improve the clarity and highlight why our results are relevant within a mean-field framework.

- Provide new experiments as suggested by one of the reviewers.

- Highlight how our theory unveils previously unseen behaviors, such as per-class gradient exploding.

## Scope and Generality of MF Theory
Mean-field (MF) theory is an established and widely used analytical framework in modern deep learning, with a long track record of highly cited contributions. Its strength lies in the breadth of its scope: MF provides a unified lens for understanding a wide variety of architectures (convolutional, residual, recurrent, or attention-based) under minimal assumptions such as large width. This generality stands in contrast to many classical theoretical approaches, which often rely on restrictive settings such as shallow networks or synthetic data distributions.

## Practical Impact and Predictive Power
A key reason MF has become influential is its ability to connect the statistical structure of random initializations with the downstream trainability of neural networks. This connection has yielded practical, experimentally validated guidelines for initialization and scaling, enabling faster and more stable training in real-world models. MF predictions, though derived in the infinite-width limit, remain surprisingly accurate at realistic widths: they underpinned the first successful training of very deep CNNs and form the basis of frameworks such as Tensor Programs that allow hyperparameters to transfer reliably across widths and depths. Importantly, MF is not confined to the lazy-training regime—recent work has shown that it can also capture nontrivial feature learning dynamics (e.g.  Yang et al. (ICML 2021)).

---

### Author Response · Authors · 2025-11-20
**General comment - linking MF predictions to initialization bias, terminology and the infinite width assumption**

# Linking MF Predictions to Initialization Bias
## Complementary Frameworks
While MF theory has had substantial impact (see comment "Relevance of Mean-field results"), its strengths also reveal its limitations. MF describes the *average* behavior over random initializations and is largely agnostic to data structure. Consequently, it cannot capture initialization-specific effects or distinguish how different datapoint subgroups are treated at initialization.
IGB fills this gap. Rather than working with ensembles, IGB characterizes the predictive state of a *single* untrained network and identifies when systematic initial biases arise. The two perspectives—ensemble-level averages (MF) and instance-level behavior (IGB)—are therefore naturally complementary.
## Hypotheses Underlying Our Results
Our main results require only that the network outputs at initialization follow a Gaussian distribution, a property that holds broadly for common architectures and initialization schemes. This assumption is simpler and more general than the full IGB analytical setting, which typically involves stronger structural constraints to derive exact output statistics.
Crucially, the quantities linking MF and IGB—such as the correlation $c$ and the induced bias $\gamma$—can be computed using MF tools across a wide variety of architectures, far beyond standard MLPs. This makes the connection between the two frameworks operational in settings where neither MF nor IGB alone would be sufficient: MF provides the required correlations, and IGB translates them into predictions about class-wise initial behavior.

## Main Contributions
Our work establishes a precise relationship between initial guessing bias and trainability. This allows us to augment MF phase diagrams with information about class behavior at initialization and use MF-derived quantities to predict when IGB will appear. Through this link, we also extend both frameworks: for MF, we refine the ReLU phase diagram; for IGB, we incorporate bias weights and address architectures with multi-variable activations (e.g., pooling).
## Implications
The combined perspective reveals when initial bias can be quickly corrected by learning dynamics (near the edge of chaos) and when it persists due to vanishing gradients (in the ordered phase). This has practical consequences. During hyper-parameter tuning, a model evaluated through short runs may seem to perform well only on a subset of the data because persistent initialization bias temporarily favors that subgroup—potentially leading to misleading model selection.




# Terminology: Bias, Prejudice and Neutrality
In this work we analyze how trainability conditions connect to systematic asymmetries in model predictions that arise *at initialization*. Because the term **bias** is heavily overloaded in machine learning—ranging from inductive bias to fairness-related bias—we introduce the term **prejudice** to refer specifically to a statistically skewed assignment of classes *before any data is observed or evaluated*. This usage is purely technical: at initialization the model has no knowledge of the dataset, and any imbalance in class guesses carries no implication of correctness or incorrectness.
Prejudice therefore denotes an **initial skew in predictive tendencies** induced solely by the random initialization scheme and model design, rather than by properties of the data. In contrast, **neutrality** refers to the symmetric case in which all classes are treated equivalently at initialization. Adopting this terminology allows us to clearly separate initialization-driven statistical asymmetries from the other established meanings of “bias” in the literature.
To improve clarity, in the final version of the manuscript, we explain more clearly the terminology and its justification.

# On the Infinite-Width Assumption

 The infinite-width regime is a standard and widely adopted setting in modern deep-learning theory. Most established analytical frameworks, MF theory included, derive asymptotically exact results in this limit, while systematic treatments of *finite-width* networks remain comparatively recent and far less developed.
Importantly, finite-width analyses are not in opposition to infinite-width theory: they typically build upon it, adding finite-size corrections rather than replacing the underlying asymptotic picture. Infinite-width theories therefore serve as the natural foundation for refined finite-width studies.
In our case, this assumption is not introduced by our analysis; it is inherent to the two well-established frameworks we connect (MF and IGB) both of which are defined in the large-width limit. Beyond the theoretical rigor, both frameworks have repeatedly shown strong empirical robustness: their infinite-width predictions align closely with the behavior of realistic finite-size networks. Our experiments confirm this consistency.

---

### Author Response · Authors · 2025-12-03
**General Summary (Part 2/2)**

# How the rebuttal addresses the main concerns
Rather than repeating the full arguments, we point to the relevant rebuttal sections:
- **MF context, scope, and infinite width (2DTJ, cZgs, YDAS).**
    (See _“Relevance of Mean-field results”_ and _“On the Infinite-Width Assumption”_ in the openreview discussion.)

    We clarified why the wide network regime is the natural setting for this work, given that both MF and IGB are defined in the infinite-width regime, and how our finite-width experiments support the relevance of the theory in realistic architectures.  [e.g. see the Introduction]

- **Terminology and fairness framing (2DTJ, h1rA, YDAS).**
   (See _“Terminology: Bias, Prejudice and Neutrality”_ and the answers to 2DTJ, h1rA, YDAS.)

    We refined the terminology (prejudice/neutrality/deep prejudice) and explicitly removed mentions on fairness, restricting the discussion to statistical asymmetries in initial predictions.  [Neutrality vs Prejudice section. Fairness mentions are now absent.]

- **Architectural scope: normalization, pooling, modern models (cZgs, h1rA).**
   (See _“Relevance of Mean-field results”_ and _“Answer to cZgs review”_ and _“Answer to h1rA review”_.)

    We explained how recent work on normalization and initialization bias is consistent with our framework, clarified how pooling extensions work, and discussed how our assumptions cover stacked modules (norm, attention, MLP, residuals) used in modern architectures.  [This was more a curiosity of the reviewer,so we addressed it in the rebuttal without including it in the new version of the manuscript]

- **Empirical scope and pretraining bias (YDAS).**
    We extended experiments to ResNets and MLP-Mixers and performed the additional pretraining vs. random-initialization test requested by reviewer YDAS, as discussed in _“MF–IGB mapping in practice”_ and the _“Official Comment by Reviewer YDAS”_. [App. E.1 and App. G.4.1]

# Why our paper is worth accepting

All points raised by the reviewers have been addressed in the rebuttal and the revised manuscript, including new experiments (ResNet, MLP-Mixer, and the pretraining vs. random-initialization test) and clearer discussions of MF scope, terminology, and practical implications. [See the “Introduction” in the revised version of the manuscript]. Reviewer YDAS — initially the most critical — explicitly acknowledged that their main concerns were resolved and raised their score, with the last requested experiment now completed and included. [App. G.4.1]

In light of the reviews and rebuttal, we believe the paper merits acceptance because it:
1. **Introduces a novel and technically non-trivial link** between two established theoretical frameworks (MF and IGB), under assumptions that make the mapping broadly applicable across architectures. [Thm. 3.1]
2. **Identifies and explains qualitatively new phenomena** in class-wise gradient and prediction behavior, including deep prejudice at the edge of chaos and class-specific gradient exploding in prejudiced phases. [e.g. Fig. 4]
3. **Provides practically relevant insights** for initialization, hyper-parameter selection, and fine tuning, supported by experiments on standard architectures and datasets and summarized into actionable guidelines. [Paragraph “Practical Takeways. ]


Thank you again for your time and for handling this submission under the current constraints.

---

### Author Response · Authors · 2025-12-03
**General summary (Part 1/2)**

Dear Area Chair,

Thanks for handling this paper under the unusual decision process of this year.

We would like to point out that, while three out of four reviewers did not have the chance to take into account our rebuttal, **reviewer YDAS increased their grade** from 2 to 4, **and remained open for a further increase** in case new runs requested were made available (they are now available).

We uploaded a new version of the manuscript which addresses all the concerns raised by the reviewers. In the following, we briefly summarize the core points, referring to the sections of the new version of the manuscript.

# Core contribution in one sentence

We trace a precise connection between two established theories: Mean-Field (MF) initialization theory and Initial Guessing Bias (IGB). We show how MF correlations predict class-wise prejudiced (i.e. initialization-induced skew in the prediction distribution) vs. neutral (i.e. symmetric initial predictions) behavior at initialization and how this is tightly linked to trainability, with empirical confirmation on realistic architectures.

# Key concepts and contributions

- **MF–IGB mapping and extended phase diagram.**
(See rebuttal section _“Linking MF Predictions to Initialization Bias”_ in the openreview discussion.)

    We formally relate MF quantities (e.g., the correlation $c$) to IGB’s bias parameter $\gamma$. This allows one to:

    - predict prejudiced vs. neutral initial behavior directly from MF phase diagrams, and
    - augment MF phase diagrams with class-wise initialization behavior, not just gradient statistics. This reveals new phases.

    We also refine the ReLU MF phase diagram and extend MF theory to multi-input activations such as pooling and IGB theory to include non-null bias weights .


- **Prejudice, deep prejudice, and trainability.**
        (See _“Terminology: Bias, Prejudice and Neutrality”_ and the discussion in _“Relevance of Mean-field results”_ in the openreview discussion.)

  Through the MF–IGB link we show that:

    - at the **edge of chaos**, where gradients remain stable across depth, the network is prejudiced

    - in the **ordered phase**, prejudice couples with vanishing gradients and persists longer during training. [Fig. 5 - Max. Freq. plot]
        This implies that a neutral initialization is not optimal, and that short runs used for hyper-parameter tuning can select models that perform well only on favored subgroups.


- **Class-wise Gradient and Practical Guidance for Bias Mitigation.**
(See _“Relevance of Mean-field results”_ and _“MF–IGB mapping in practice”_ in the openreview discussion.)
The MF–IGB connection reveals that **exploding gradients can be class-specific**: in prejudiced phases, gradients for unfavored classes can explode while those for favored classes do not. [Fig. 4]
It also suggests a practical guideline, especially to avoid persistent bias in short runs (e.g., during hyper-parameter tuning): scan initialization scales to identify regimes where gradients are stable across depth and prejudice is transient rather than persistent.


- **Evidence in realistic settings.**
(See _“Relevance of Mean-field results”_, _“On the Infinite-Width Assumption”_, and the additions described in _“MF–IGB mapping in practice”_ in the openreview discussion.)
    While both MF and IGB are formulated in the infinite-width limit, we empirically verify the predicted phase structure and bias–trainability relationship on:

    - **ResNets** and **MLP-Mixers**, using real data, where the MF == IGB curves closely match theoretical predictions; [App. E.1]

    - a **Vision Transformer** in a fine-tuning setting, where the predicted initialization prejudice persists and interacts with trainability; [Fig. 6 - bottom and App. G.4]

    - an additional **pretraining vs. random initialization** experiment on ViT architectures, isolating the effect of the pretraining prior on gradient stability and prejudice, as suggested by reviewer YDAS. [App. G.4.1]

---

### Meta-Review · Area_Chair_jLKV · 2026-01-09

**Summary:**

The paper establishes a theoretical equivalence between Mean-Field (MF) theory and Initial Guessing Bias (IGB), proposing that the optimal initialization for trainability (the "Edge of Chaos") corresponds to a state of "transient deep prejudice" rather than neutrality. Reviewers recognized the novelty of connecting these two theoretical frameworks. However, the initial reviews highlighted significant concerns regarding the clarity of the presentation, the use of loaded terminology (specifically "prejudice" and "fairness"), the reliance on the infinite-width assumption, and the limited scope of the empirical validation (initially focused on toy models and datasets).

**Reviewer Concerns:**

ADDRESSED: The authors' rebuttal was comprehensive and effectively addressed the majority of the actionable criticisms. The concerns regarding "fairness" framing and "prejudice" terminology were resolved by removing fairness claims and clarifying the statistical definition of prejudice (addressing 2DTJ, h1rA, YDAS). The empirical validation was significantly strengthened; authors included new experiments on ResNets and MLP-Mixers to demonstrate applicability beyond toy models (addressing YDAS, cZgs). Crucially, the authors conducted a specific experiment requested by reviewer YDAS to disentangle pre-training priors from initialization bias in ViTs. The concern regarding practical utility (h1rA) was addressed by providing actionable guidelines for hyperparameter scanning.

OUTSTANDING: The criticism regarding the infinite-width assumption (raised by 2DTJ and cZgs) remains a fundamental limitation of the theoretical framework. While the authors successfully argued that this is the standard regime for MF/IGB theories and provided finite-width empirical support, the theoretical gap regarding the "feature learning" regime (as opposed to lazy learning) raised by 2DTJ cannot be fully resolved within the current scope of the theory, representing an inherent, though defended, limitation.

**Reviewer Scores:**

Reviewer cZgs (Score: 8): This reviewer would likely maintain their score of 8 (Accept). They were already positive about the conceptual contribution, and the rebuttal satisfactorily addressed their specific questions regarding normalization and pooling layers.

Reviewer YDAS (Score: 4): This reviewer would likely have increased their score to a 6. They explicitly stated during the discussion that "Most of my concerns have been addressed" and raised their score from 2 to 4 before the authors provided the final requested experiment. Since the authors successfully provided this experiment (isolating pre-training priors) in their final comment, satisfying the reviewer's "last remaining" condition, a further score increase is well-justified.

Reviewer h1rA (Score: 4): This reviewer would likely have increased their score to a 5. Their primary concerns regarding the lack of practical recipes and the confusion around terminology were directly addressed in the revision, but the reviewer did not engage in the discussion to acknowledge these improvements.

Reviewer 2DTJ (Score: 4): This reviewer might have maintained their score or increased it slightly to a 5. While the terminology issues were resolved, their fundamental reservation regarding the paper's readability and the theoretical focus on the "lazy learning" regime might not have been fully alleviated by the rebuttal arguments alone.

---

### Decision · Program_Chairs · 2026-01-26

Accept (Poster)